# The growth factor EPIREGULIN promotes basal progenitor cell proliferation in the developing neocortex

Paula Cubillos[1], Nora Ditzer [1], Annika Kolodziejczyk [1], Gustav Schwenk[1], Janine Hoffmann [1], Theresa M Schütze[1], Razvan P Derihaci [2,3], Cahit Birdir[2,4], Johannes EM Köllner [5], Andreas Petzold[6], Mihail Sarov [5], Ulrich Martin [7,8], Katherine R Long [9,10], Pauline Wimberger [2,3] & Mareike Albert [1✉]

## Abstract

**Neocortex expansion during evolution is linked to higher numbers of neurons, which are thought to result from increased proliferative capacity and neurogenic potential of basal progenitor cells during development. Here, we show that *EREG*, encoding the growth factor EPIREGULIN, is expressed in the human developing neocortex and in gorilla cerebral organoids, but not in the mouse neocortex. Addition of EPIREGULIN to the mouse neocortex increases proliferation of basal progenitor cells, whereas *EREG* ablation in human cortical organoids reduces proliferation in the subventricular zone. Treatment of cortical organoids with EPIREGULIN promotes a further increase in proliferation of gorilla but not of human basal progenitor cells. EPIREGULIN competes with the epidermal growth factor (EGF) to promote proliferation, and inhibition of the EGF receptor abrogates the EPIREGULIN-mediated increase in basal progenitor cells. Finally, we identify putative cis-regulatory elements that may contribute to the observed inter-species differences in *EREG* expression. Our findings suggest that species-specific regulation of EPIREGULIN expression may contribute to the increased neocortex size of primates by providing a tunable pro-proliferative signal to basal progenitor cells in the subventricular zone.**

**Keywords** Cortical Organoid; Gene Regulation; Human Neurogenesis; Neocortex Evolution; Neural Progenitor Cell
**Subject Categories** Development; Evolution & Ecology; Neuroscience

## Introduction

The neocortex is central to higher cognitive functions. Mammalian brain evolution is associated with an increase in the expansion and folding of the neocortex (Llinares-Benadero and Borrell, 2019; Rakic, 2009; Sousa et al, 2017), yet our understanding of the exact mechanism of this evolutionary expansion and the genomic basis of neocortex evolution is still limited. Inter-species differences in neocortex size are mostly related to the number of neurons, which are generated from neural stem and progenitor cells (NPCs) during development. While the process of neurogenesis is largely conserved across mammals, differences in the proliferative capacity of NPCs, the abundance of distinct NPC types, and the length of the neurogenic period are thought to be key determinants of neuron number (Dehay et al, 2015; Florio and Huttner, 2014; Lui et al, 2011; Taverna et al, 2014; Zhou et al, 2023).

NPCs can be divided into two principal groups: Apical progenitors (APs), mainly apical radial glia (aRG), that reside in the ventricular zone (VZ), and basal progenitor cells (BPs) that reside in the subventricular zone (SVZ). Across mammals, aRG are highly proliferative and can generate more aRG via symmetric proliferative divisions, or BPs and neurons, mostly via multiple rounds of asymmetric self-renewing divisions (Götz and Huttner, 2005; Rakic, 2003). Their position adjacent to the ventricle provides aRG access to the pro-proliferative signals of the cerebrospinal fluid (Lehtinen et al, 2011).

In contrast, BPs are variable in their proliferative potential across mammals. In species with a small and smooth neocortex, such as the mouse, BPs typically divide only once to generate two neurons (Haubensak et al, 2004; Miyata et al, 2004; Noctor et al, 2004), whereas BPs of species with a large and folded neocortex, such as human, can undergo multiple rounds of division (Betizeau et al, 2013; Florio and Huttner, 2014; Lui et al, 2011). Species

[1]Center for Regenerative Therapies TU Dresden, TUD Dresden University of Technology, 01307 Dresden, Germany. [2]Department of Gynecology and Obstetrics, TU Dresden, 01307 Dresden, Germany. [3]National Center for Tumor Diseases, 01307 Dresden, Germany. [4]Center for feto/neonatal Health, TU Dresden, 01307 Dresden, Germany. [5]Max Planck Institute of Molecular Cell Biology and Genetics, 01307 Dresden, Germany. [6]DRESDEN-concept Genome Center, Center for Molecular and Cellular Bioengineering, TUD Dresden University of Technology, 01307 Dresden, Germany. [7]Leibniz Research Laboratories for Biotechnology and Artificial Organs, Department of Cardiothoracic, Transplantation and Vascular Surgery, Hannover Medical School, 30625 Hannover, Germany. [8]REBIRTH-Cluster of Excellence, Hannover, Germany. [9]Centre for Developmental Neurobiology, Institute of Psychiatry, Psychology and Neuroscience, King's College London, London SE1 1UL, United Kingdom. [10]MRC Centre for Neurodevelopmental Disorders, King's College London, London SE1 1UL, United Kingdom. ✉E-mail: mareike.albert@tu-dresden.de

differences are also linked to the types of BPs that are present. In the mouse neocortex (mNcx), basal intermediate progenitor cells (bIPs) are predominant, whereas in species with a large neocortex, basal or outer radial glia (bRG/oRG) are highly abundant (Fietz et al, 2010; Hansen et al, 2010; Reillo et al, 2011). In the context of neocortex expansion, bRG are thought to be important as they are highly proliferative (Betizeau et al, 2013; Borrell and Calegari, 2014; Florio and Huttner, 2014). Overall, the increased proliferative capacity of BPs in species with a large and folded neocortex leads to an expansion of the SVZ, resulting in an inner and outer layer (ISVZ/OSVZ), which represents an additional proliferative niche away from the ventricle and the signals of the cerebrospinal fluid (Fietz et al, 2012; Florio and Huttner, 2014; Lehtinen et al, 2011; Libe-Philippot and Vanderhaeghen, 2021; Pollen et al, 2015). While extracellular matrix (ECM) components are thought to contribute to this niche (Kalebic and Huttner, 2020), few other factors have been functionally explored.

Recent research addressing the genomic basis of human, or generally primate, neocortex expansion has led to the identification of a limited number of human- and primate-specific genes that are implicated in NPC proliferation and/or neocortex expansion and folding (Fiddes et al, 2018; Florio et al, 2015; Florio et al, 2018; Ju et al, 2016; Liu et al, 2017; Pinson et al, 2022; Suzuki et al, 2018; Van Heurck et al, 2022; Zhou et al, 2023). In addition to such novel gene functions that mostly arose from recent segmental duplication events (Dennis and Eichler, 2016), phenotypic evolution is thought to be driven by changes in developmental gene regulatory networks (Davidson and Erwin, 2006). Comparative transcriptomic studies of mouse and primates have provided evidence of primate- or human-specific gene expression changes during cortical development, suggesting that non-coding regulatory modifications contribute to brain evolution (Doan et al, 2018; Espinos et al, 2022; Libe-Philippot and Vanderhaeghen, 2021; Mitchell and Silver, 2018). Specifically, large deletions and duplications in non-coding regions (McLean et al, 2011), human-accelerated regions (HARs) (Hubisz and Pollard, 2014; Whalen et al, 2023), and human ancestor quickly evolved regions (HAQERs) (Mangan et al, 2022) have been reported to involve neurodevelopmental enhancers. HARE5 is an exciting example of an enhancer with species-specific activity in the developing neocortex, linked to *FZD8* encoding a receptor in the canonical Wnt signaling pathway (Boyd et al, 2015). Despite the relevance of such inter-species differences in gene expression arising from changes in gene regulatory networks, few other primates- or human-expressed genes have been functionally explored to date (Espinos et al, 2022; Libe-Philippot and Vanderhaeghen, 2021; Mitchell and Silver, 2018; Pollen et al, 2023).

Here we study EPIREGULIN, a member of the epidermal growth factor (EGF) family of ligands that drive multiple cellular signal transduction pathways, such as ERK and AKT activation, through the ErbB subclass of receptor tyrosine kinases (Abud et al, 2021; Hynes and Lane, 2005). EPIREGULIN is encoded by the *EREG* gene and is first synthesized as a membrane-anchored precursor that is then released from the cell surface by the metalloproteinase ADAM17 (Sahin et al, 2004).

Although *Ereg* knockout mice do not show any overt developmental phenotypes (Lee et al, 2004), Epiregulin has been implicated in the regulation of cell proliferation, differentiation, and cell death in multiple contexts, such as angiogenesis, skin inflammation, ovarian follicle formation, and cancer (Riese and Cullum, 2014). In this study, we have examined the role of EPIREGULIN in NPC proliferation in the context of neocortex evolution by manipulating EPIREGULIN levels in the mNcx, gorilla cortical organoids, and the developing human neocortex (hNcx). We propose that EPIREGULIN secretion by radial glia of the developing hNcx contributes a pro-proliferative signal to the niche of the SVZ that contributes to basal progenitor amplification.

# Results

## *EREG* is expressed in human but not mouse radial glia of the developing neocortex

We previously performed inter-species transcriptome comparisons of sorted NPCs from the developing mNcx and hNcx, resulting in the identification of 207 genes that are expressed in human radial glia (RG) at higher levels than in neurons and that are not expressed in mouse NPCs, despite the presence of an ortholog in the mouse genome (Florio et al, 2015). Of these, 62 genes were marked by repressive histone 3 lysine 27 tri-methylation (H3K27me3) in the mNcx (Albert et al, 2017), potentially contributing to their repression in the mouse.

Among these genes was *EREG*, encoding for the growth factor EPIREGULIN (Fig. 1A), which was undetectable by RNA-seq in the mNcx (Fig. 1B), but expressed in human aRG and bRG at higher levels than in neurons in fetal hNcx at 12/13 and 18/19 gestation weeks (GW) (Figs. 1C and EV1A) (Albert et al, 2017; Florio et al, 2015; Johnson et al, 2015). Moreover, *EREG* was also expressed in aRG of human cerebral organoids (Fig. EV1B) (Camp et al, 2015). The absence of *Ereg* mRNA expression in the mNcx was supported by in situ hybridization data (Fig. EV1C) (Allen Institute for Brain Science, 2004). Expression analysis in embryonic mNcx and fetal hNcx tissue by RT-qPCR (Figs. 1D and EV1D) corroborated the inter-species difference in *EREG* mRNA expression.

Furthermore, in the mNcx, the *Ereg* gene locus showed low enrichment for active chromatin modifications, such as acetylation of H3K27 (H3K27ac) and H3K4me3, and instead was enriched in repressive H3K27me3 (Fig. EV1E). In contrast, the human *EREG* locus was highly enriched in active H3K27ac in the fetal hNcx, in agreement with the differential *EREG* expression between the mNcx and hNcx.

## *EREG* is expressed in primate neural progenitor cells

To address whether the expression of *EREG* in the developing neocortex is a human-specific feature or is also found in other primates, we generated gorilla-induced pluripotent stem cell (iPSC)-derived cerebral organoids and compared *EREG* expression to human iPSC-derived organoids (Figs. 1D and EV1F). Both gorilla and human organoids expressed *EREG* mRNA at similar levels during neurogenesis. Unfortunately, we did not succeed in detecting the Epiregulin protein in organoids (Appendix Fig. S1). However, mining of RNA-seq data from macaque and human NPCs (Kliesmete et al, 2023) revealed similar expression levels of *EREG* mRNA in both species (Fig. EV1G). Moreover, *EREG* may also be expressed in the germinal zones of the ferret, a carnivore with an expanded OSVZ (Fig. EV1H) (de Juan Romero et al, 2015). These data indicate that *EREG* expression is not restricted to

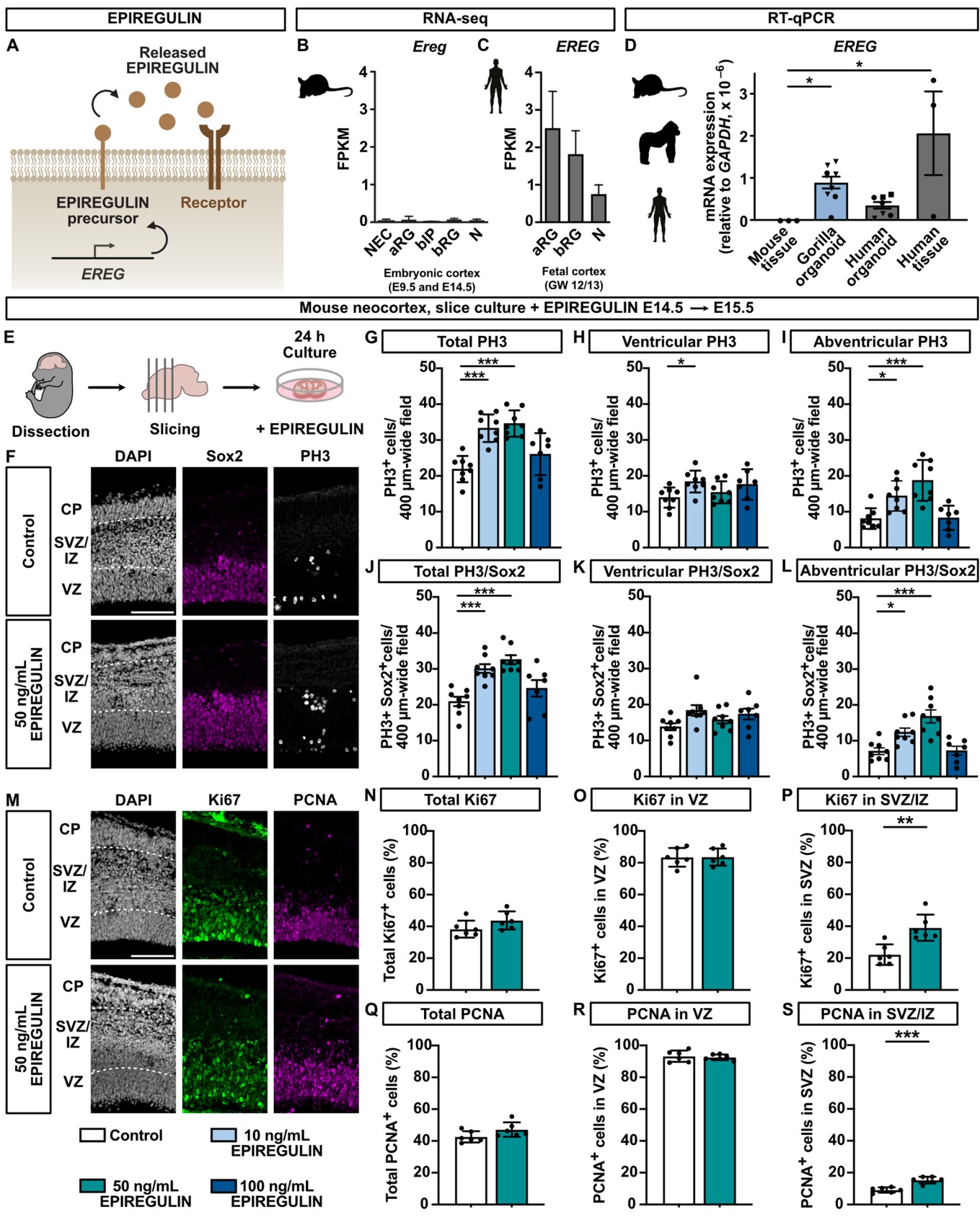

**Figure 1.** *EREG is expressed in the developing hNcx and induces NPC proliferation in the mNcx.*

(A) Schematic illustration of the expression of EPIREGULIN from the *EREG* gene. (B, C) *EREG* mRNA levels in mouse and human NPCs and neurons analyzed by RNA-seq (data from Albert et al, (2017); Florio et al, (2015)). (D) RT-qPCR expression analysis of *EREG* in mNcx (E14.5), gorilla cerebral organoids (week 6–8), human cortical organoids (week 6), and fetal hNcx (GW 12/13), relative to *GAPDH*. (E) Schematic of experimental workflow. Mouse brains (E14.5) were cut into slices and treated with different concentrations of EPIREGULIN for 24 h. (F) DAPI staining and immunofluorescence for Sox2 and PH3 of slices treated with 50 ng/mL EPIREGULIN. (G–L) Quantifications of total, ventricular and abventricular mitotic PH3+ cells and PH3+ Sox2+ double-positive cells of slices treated with different concentrations of EPIREGULIN. (M) DAPI staining and immunofluorescence for Ki67 and PCNA of slices treated with 50 ng/mL EPIREGULIN. (N–P) Quantifications of total Ki67-positive cells, and Ki67-positive cells in the VZ and SVZ/IZ (percentage of DAPI). (Q–S) Quantifications of total PCNA-positive cells, and PCNA-positive cells in the VZ and SVZ/IZ (percentage of DAPI). Data information: Scale bars, 100 μm. Bar graphs represent mean values. Error bars represent SD; (B), of 4–5 tissue samples from different litters; (C), of 2–4 tissue samples from different individuals; (D), of three tissue samples from different individuals and 7–8 organoids (different symbols indicate independent batches); (G–L, N–S), of eight embryos from different litters, with each dot representing the average of three to six images of different sections of the same brain. ***$p < 0.001$, **$p < 0.01$, *$p < 0.05$; one-way ANOVA with Dunnett post hoc test (D, G–L), and Student's *t*-test (N–S). Source data are available online for this figure.

humans, but also seen in two additional gyrencephalic primates, and in the ferret.

Given the differential expression of *EREG* in the developing neocortex of a species with a small and smooth neocortex versus the expanded and highly folded primate neocortex, we speculated that the growth factor EPIREGULIN might contribute to differences in the proliferative capacity of NPCs between species.

## Addition of EPIREGULIN to the developing mNcx increases basal progenitor proliferation

Evidence for a potential role of *EREG* in cell proliferation comes from data on human brain tumors (Appendix Fig. S2) (Bao et al, 2014; Ceccarelli et al, 2016; Gill et al, 2014; Griesinger et al, 2013; Sun et al, 2014; Yan et al, 2012; Zhang et al, 2013). While *EREG* expression is not *per se* increased in glioblastoma, *EREG* expression is higher in high-grade compared to low-grade glioma, and loss of *EREG* correlates with increased survival, supporting a potential role of EPIREGULIN in cell proliferation. Furthermore, Epiregulin was shown to enhance tumorigenicity by activating the ERK/MAPK pathway in a mouse model of glioblastoma (Kohsaka et al, 2014).

To directly test the effect of EPIREGULIN on NPC proliferation, we incubated organotypic slice cultures of embryonic day (E) 14.5 mNcx with or without different concentrations of recombinant EPIREGULIN for 24 h (Fig. 1E–L), followed by the analysis of cell proliferation. Immunofluorescence of phospho-histone 3 (PH3), a marker of mitotic cells, and of the RG marker Sox2 revealed that 10 and 50 ng/mL of EPIREGULIN caused a significant increase in the total number of mitotic cells (Fig. 1F,G). While there was only a small increase in ventricular mitosis (Fig. 1H), abventricular mitosis, defined as more than three nuclei away from the ventricular surface, was strongly increased (Fig. 1I), suggesting that BPs are particularly responsive to the addition of EPIREGU-LIN. Of the three EPIREGULIN concentrations tested, the intermediate concentration of 50 ng/mL resulted in the highest increase in abventricular PH3-positive cells compared to the control condition. This concentration was therefore used for all further experiments.

We next examined additional markers of cell proliferation, specifically Ki67 and proliferating cell nuclear antigen (PCNA) (Fig. 1M–S). The total percentage of both Ki67- and PCNA-expressing cells was comparable between the control and the addition of EPIREGULIN (Fig. 1N,Q). Moreover, there was no change in the percentage of Ki67- and PCNA-positive cells in the VZ, defined based on the alignment of pseudostratified nuclei in

DAPI (Fig. 1O,R). However, both the percentage of Ki67- and PCNA-positive cells was significantly increased in the SVZ/IZ (Fig. 1M,P,S). These data suggest that EPIREGULIN stimulates the proliferation of NPCs in the SVZ at mid-neurogenesis in the mNcx.

## Addition of EPIREGULIN to the developing mNcx increases both major basal progenitor types

Next, we explored which of the distinct NPC types were affected by the addition of EPIREGULIN. First, we performed immunofluorescence analysis of the RG marker Sox2 in organotypic slice cultures with and without EPIREGULIN treatment. The total percentage of Sox2-positive cells was unaffected after the addition of EPIREGULIN (Fig. 2A,B). Likewise, the percentage of Sox2 was unchanged in the VZ; however, there was a significant increase in the percentage of Sox2-positive cells in the SVZ/IZ (Fig. 2C,D). In addition, the percentage of Tbr2-positive bIPs was specifically increased in the SVZ/IZ (Fig. 2E–G), while the overall number of cells did not change (Appendix Fig. S3A–C).

A high number of Sox2-positive bRG is a characteristic of the expanded SVZ in gyrencephalic species, such as humans and ferrets (Fietz et al, 2010; Hansen et al, 2010; Reillo et al, 2011). To further characterize the BP population in the EPIREGULIN-treated mNcx, we performed hemisphere rotation (HERO) cultures at E14.5 (Fig. 2H) (Schenk et al, 2009), followed by slicing into 70 μm thick sections to facilitate the analysis of basal processes, another hallmark of bRG (Fietz et al, 2010; Hansen et al, 2010; Reillo et al, 2011). We first confirmed that the EPIREGULIN-mediated increase in abventricular mitosis was recapitulated in the HERO culture system (Appendix Fig. S3D–I). Using phospho-Vimentin (pVim) to analyze mitotic BPs, we could distinguish BPs without a process, mostly bIPs, and with a process, mostly bRG (Fig. 2I). This analysis corroborated the increase in abventricular mitosis (Fig. 2J) already observed with PH3 (Fig. 1I), and further revealed that mitotic cells both with and without a process were increased upon EPIREGULIN treatment (Fig. 2K,L). Among the basal progenitor cells with a process, both Sox2-positive (Fig. 2M) and Sox2-negative (Fig. 2N) cells were increased.

Taken together, we observed that the addition of recombinant EPIREGULIN to mNcx cultures resulted in an increase in NPC proliferation, preferentially in the SVZ. Both major types of BPs, bIPs characterized by Tbr2 expression and the lack of a process as well as bRG marked by Sox2 and the presence of a process, were increased, suggesting that EPIREGULIN can induce the amplification of different BP types in the mNcx.

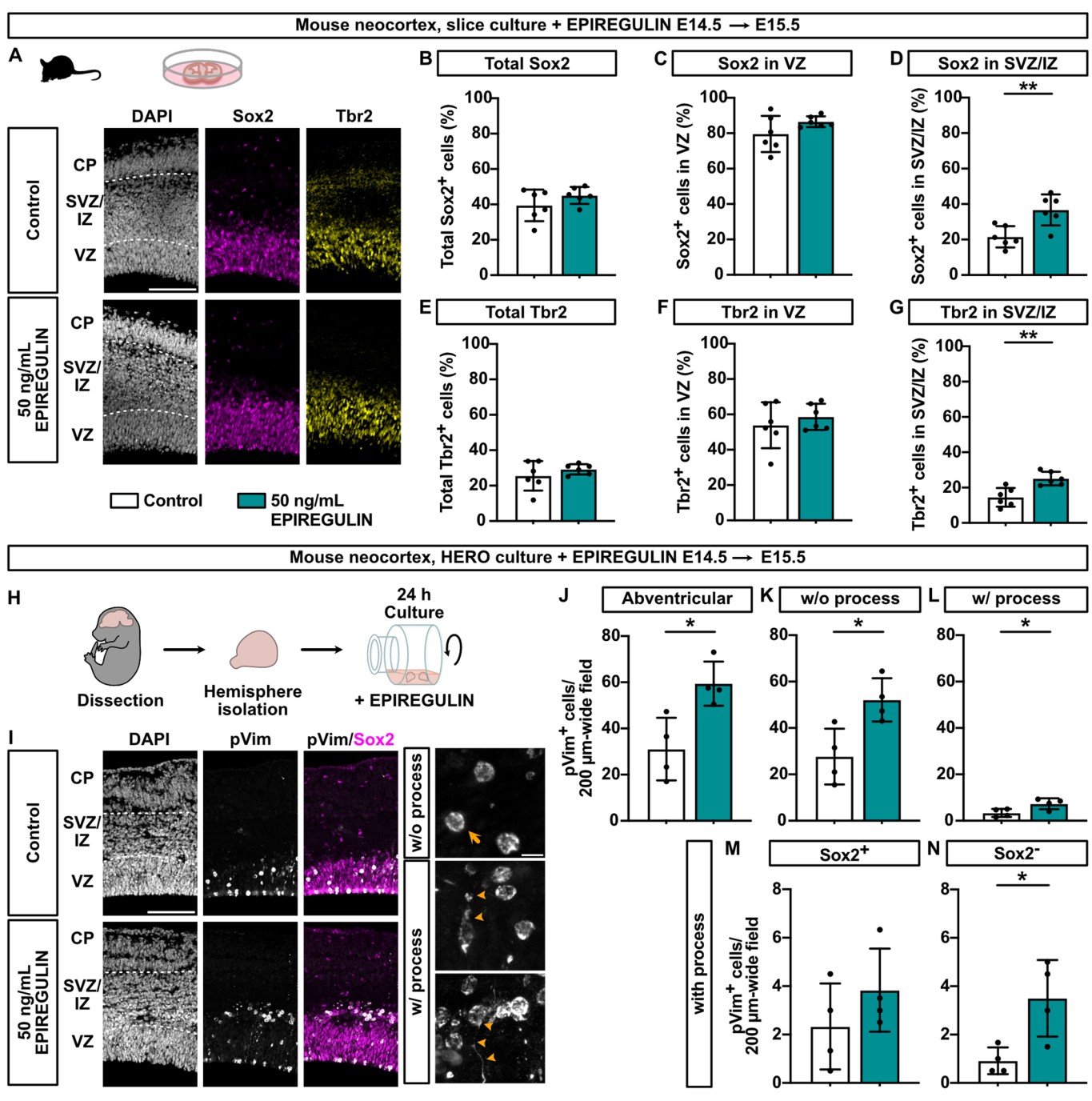

**Figure 2. Addition of EPIREGULIN to the mNcx increases BPs.**

(A) DAPI staining and immunofluorescence for Sox2 and Tbr2 of mNcx slices (E14.5) cultured with 50 ng/mL EPIREGULIN for 24 h. (B–D) Quantifications of the percentage of total Sox2, and Sox2 in the VZ and SVZ/IZ (of DAPI-positive cells). (E–G) Quantifications of the percentage of total Tbr2, and Tbr2 in the VZ, and SVZ/IZ (of DAPI-positive cells). (H) Schematic of experimental workflow. mNcx hemispheres (E14.5) were cultured under rotation (HERO) in the presence of 50 ng/mL EPIREGULIN for 24 h. (I) DAPI staining and immunofluorescence for phospho-Vimentin (pVim) and Sox2 to distinguish mitotic BPs without (top; arrow) or with (bottom; arrowheads) a process. (J–L) Quantifications of abventricular pVim-positive cells, and pVim-positive cells without and with a process. (M, N) Quantifications of Sox2+ and Sox2-cells among pVim-positive cells with a process. Data information: Scale bars, 100 μm (A, I left) and 10 μm (I inset). Bar graphs represent mean values. Error bars represent SD of 4–6 embryos from different litters with each dot representing the average of 3–4 images of different sections of the same brain. **$p < 0.01$, *$p < 0.05$; Student's *t*-test. Source data are available online for this figure.

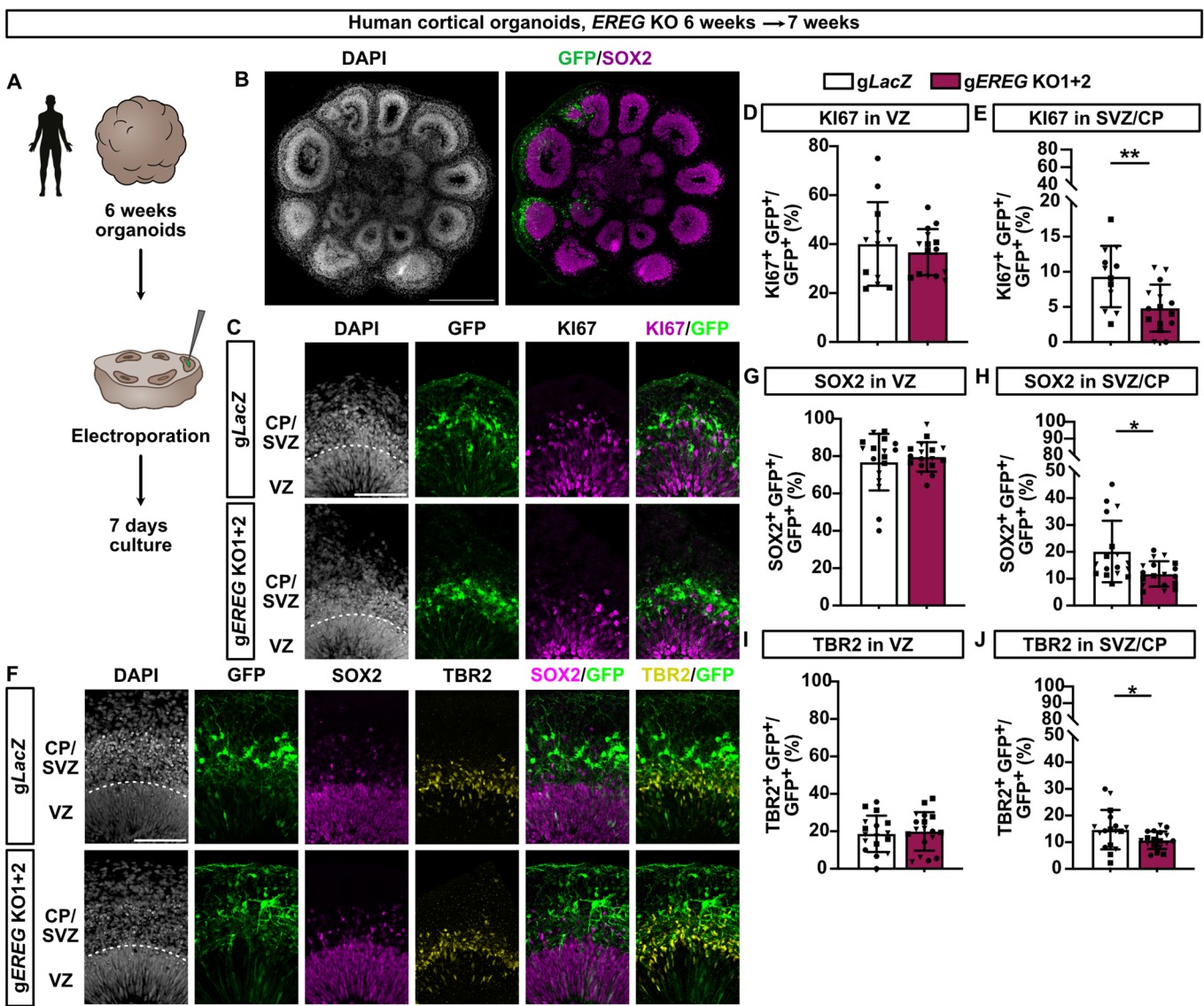

**Figure 3. EPIREGULIN ablation in human cortical organoids reduces BP proliferation.**

(A) Schematic of experimental workflow. Human cortical organoids (6 weeks) derived from the CRTDi004-A iPSC line were electroporated with a plasmid encoding GFP and CRISPR/Cas9 RNP complexes targeting either *LacZ* or *EREG*, and analyzed after 7 days. (B) DAPI staining and immunofluorescence for GFP and SOX2 of an electroporated human cortical organoid. (C) DAPI staining and immunofluorescence for GFP and KI67. (D, E) Quantifications of KI67 in the VZ and SVZ/CP. (F) DAPI staining and immunofluorescence for GFP, SOX2, and TBR2. (G–J) Quantifications of SOX2 and TBR2 in the VZ and SVZ/CP. Data information: Scale bars, 500 µm (B) and 100 µm (C, F). Bar graphs represent mean values. Error bars represent SD of 12–18 organoids from three independent batches (indicated by different symbols). **$p < 0.01$, *$p < 0.05$; one-way ANOVA with Dunnett post hoc test. Source data are available online for this figure.

## EPIREGULIN ablation in human cortical organoids reduces basal progenitor proliferation

We next examined whether endogenously expressed EPIREGULIN is required for BP proliferation in the hNcx. To address this important point, we used sliced human cortical organoids, which were reported to sustain neurogenesis, organoid growth and SVZ expansion over a long time period (Qian et al, 2020). We considered organoids to be a suitable model for this question as *EREG* expression was also observed in human cerebral and cortical organoids in addition to human fetal tissue (Figs. 1D and EV1B). We employed the CRISPR/Cas9 system and injected guide RNAs

(gRNA) in complex with recombinant Cas9 protein (Kalebic et al, 2016) into ventricle-like structures of 6-week cortical organoids followed by electroporation (EP) (Fig. 3A). Co-electroporation of a GFP expression plasmid allowed the identification of targeted cells (Fig. 3B). To disrupt the expression of *EREG*, a set of two gRNAs was used that showed efficient targeting in vitro and in the iPSC line used to generate the cortical organoids (Fig. EV2A–C). Seven days post-EP, immunofluorescence for the proliferation marker KI67 was performed (Fig. 3C–E). This showed a significant reduction in the percentage of KI67-positive GFP-positive cells in the SVZ/CP-like region (Fig. 3E), but not in the VZ (Fig. 3D), compared to the control. Similar results were observed in cortical

organoids from an independent iPSC line (Fig. EV2D–H). This indicates that EPIREGULIN is required for BP, but not AP proliferation in human cortical organoids.

To assess which NPC types were affected by the ablation of *EREG* in the SVZ, we performed immunofluorescence staining for the RG marker SOX2 and the bIP marker TBR2 (Figs. 3F–J and EV2D–H). Both SOX2 and TBR2 were reduced upon *EREG* targeting in the SVZ/CP, but not VZ, whereas the number of CTIP2-positive neurons was not changed in the SVZ/CP (Fig. EV2I–K). Taken together, these data suggest that EPIREGU-LIN contributes to the proliferation of both BP types, bIPs and bRG, in 6-week cortical organoids, in which the SVZ-like area is in the process of expansion.

## Addition of EPIREGULIN to the gorilla, but not human, cortical organoids increases basal progenitor proliferation

Given that exogenous EPIREGULIN promoted BP proliferation in the mNcx, we next asked whether the addition of EPIREGULIN to the hNcx would increase BP proliferation even further. To address this question, we first incubated human fetal cortical tissue of 12/13 GW in a free-floating tissue culture (FFTC) system (Long et al, 2018) with and without 50 ng/mL EPIREGULIN for 24 h (Fig. 4A). Immunofluorescence analysis showed no change in the prolifera-tion marker KI67 nor in the NPC markers SOX2 and TBR2 (Fig. 4B–E). Considering that the cell cycle length of primate NPCs is much longer than in the mNcx (Kornack and Rakic, 1998), we speculated that the 24-h incubation time might not have been sufficient to induce a measurable increase in proliferation. We therefore turned to the cortical organoid system again, which allowed the addition of EPIREGULIN for longer time periods and at different developmental stages. To rule out any issues of EPIREGULIN penetration to the organoid core, we added the growth factor directly after organoid slicing when the ventricle-like structures are open and accessible. However, neither a 4- nor a 10-day incubation with EPIREGULIN were sufficient to increase NPC proliferation or NPC numbers in 6-week or 10-week human cortical organoids (Fig. 4F–M).

In contrast, incubation of 6-week gorilla cortical organoids with the same concentration of EPIREGULIN resulted in a significant increase in the percentage of KI67-positive cells (Fig. 5A–H), specifically in the SVZ (Fig. 5F). In particular, the percentage of SOX2-positive BPs in the SVZ/CP was increased (Fig. 5G), whereas the percentage of TBR2-positive BPs remained largely unchanged (Fig. 5H).

Taken together, treatment with EPIREGULIN did not further increase human BP proliferation in either primary fetal tissue nor in cortical organoid cultures, suggesting that the system is already saturated and that, at least at this concentration, human BPs are not receptive to further stimulation by EPIREGULIN. Unlike the human, gorilla BP proliferation, in particular the percentage of SOX2-positive cells likely representing bRG, could be further stimulated by the addition of EPIREGULIN.

## Addition of EPIREGULIN does not induce major gene expression changes in the mNcx

To investigate the mechanism through which EPIREGULIN promotes NPC proliferation, we first analyzed gene expression

changes upon the addition of EPIREGULIN to the mNcx at E14.5 to identify putative downstream target genes. Since EPIREGULIN promoted changes in the cellular composition of the mNcx (Fig. 2), which would be expected to bias any gene expression analysis, we performed immuno-fluorescent activated cell sorting (FACS) to isolate neural cell populations. Specifically, RG, IPs, and neurons (Fig. EV3A,B) were isolated based on immuno-staining for the nuclear markers Sox2 and Tbr2 (Florio et al, 2015; Schütze et al, 2023), and neurons based on the expression of GFP in the *Tubb3*::GFP mouse reporter line (Attardo et al, 2008). Isolated cell populations were validated by expression of the marker genes *Sox2* and *Prom1* (Prominin 1/CD133) for aRG, *Eomes* (Tbr2) for IPs, and *Dcx* (Doublecortin) and *Tubb3* (Tuj1) for neurons (Fig. EV3C,D). The RNA-seq data confirmed the lack of *Ereg* expression in the mNcx (Fig. EV3D). A comparison of the NPC populations from control and EPIREGULIN-treated mNcx did not reveal any significant gene expression changes (Fig. EV3E). In line with this, principal component analysis showed strong clustering of each of the three cell types, irrespective of EPIREGULIN treatment (Fig. EV3F). Taken together, the RNA-seq analysis of sorted NPC populations did not point to major gene expression changes as the mechanism underlying the EPIREGULIN-induced increase in cell proliferation.

## EPIREGULIN competes with EGF

Secondly, we considered that secreted EPIREGULIN, released upon cleavage by ADAM17 (Fig. 6A), might induce downstream signaling responses, via binding to its target receptors that ultimately affect the balance of NPC proliferation versus differ-entiation. Of note, while EPIREGULIN is one of eleven structurally related members of the EGF family of proteins (Abud et al, 2021), none of the other members showed consistent differences in gene expression between the mNcx and hNcx (Appendix Fig. S4; and Camp et al, (2015)). EPIREGULIN has been reported to bind to the EGF receptor (EGFR) and ERBB4 receptor (Fig. 6A; Appendix Fig. S4) (Hynes and Lane, 2005). Mining RNA-seq data (Florio et al, 2015), we found that *EGFR* and *ERBB4* are both expressed in human aRG, bRG, and neurons (Fig. 6B,C). Interestingly, while *Egfr* expression is similar in the mNcx compared to the hNcx, *ErbB4* levels are very low in mouse RG, but high in bIPs and neurons. At the protein level, EGFR was reported to be expressed at high levels in both the rodent VZ and SVZ throughout cortical development (Eagleson et al, 1996).

To dissect the contribution of different growth factors and receptors, we started by treating mouse neural stem cells (NSCs), derived from the dorsolateral telencephalon at E11.5/E12.5, cultured under proliferative conditions in the presence of 10 ng/mL EGF and 20 ng/mL fibroblast growth factor (FGF), with different concentrations of EPIREGULIN for three days (Fig. 6D–F). This revealed that none of the different EPIREGULIN concentrations resulted in changes in mNSC proliferation in vitro (Fig. 6F). Since EGF and EPIREGULIN are both known to bind to EGFR, we next considered that both ligands may compete for the receptor. We therefore tested the different concentrations of EPIREGULIN on mNSCs grown in a medium lacking EGF (Fig. 6E,G). Upon acute withdrawal of EGF, mNSCs did not proliferate further (Fig. 6G). However, when EPIREGULIN was added, we observed a dose-dependent increase in mNSC

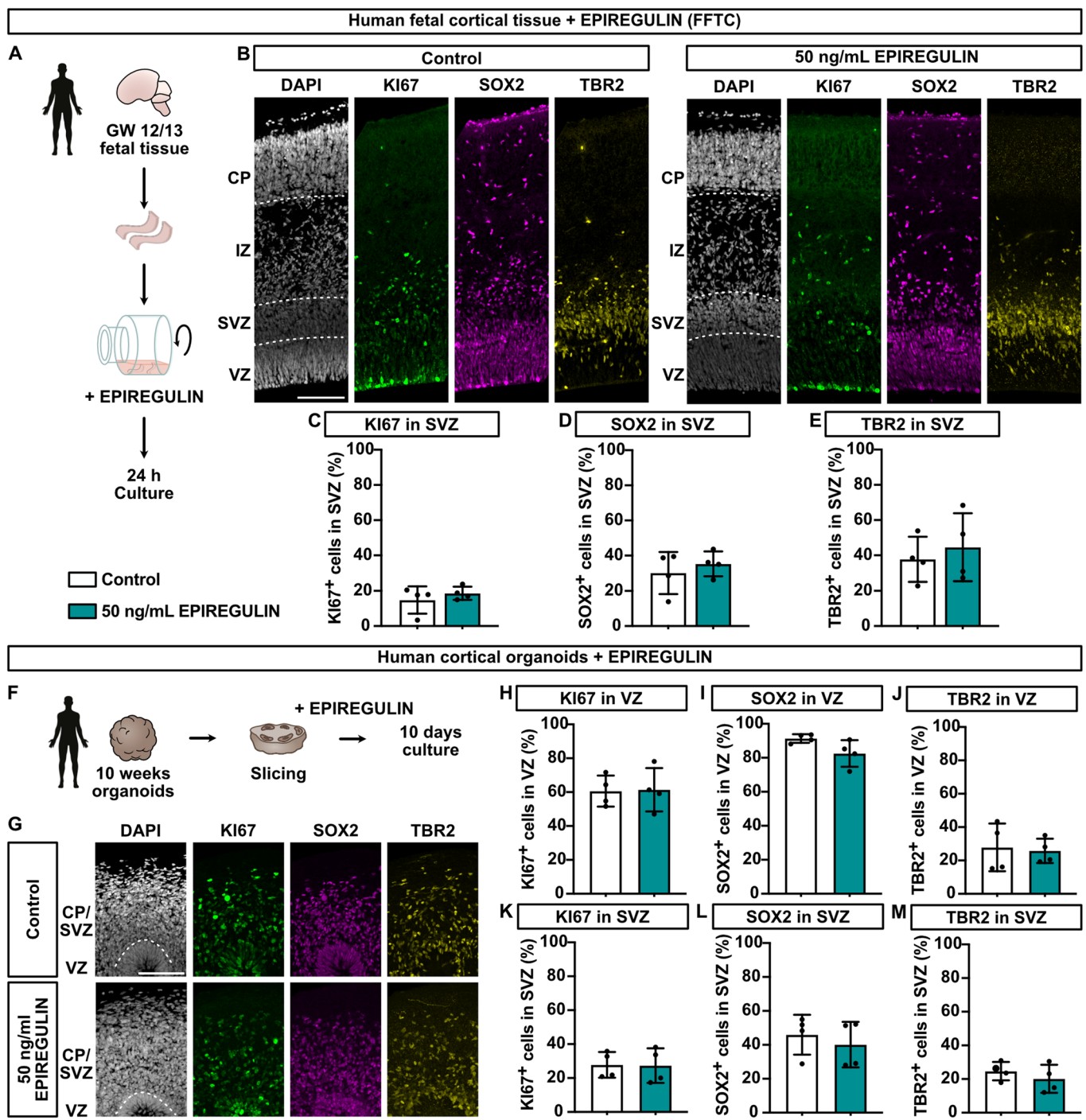

**Figure 4.  Addition of EPIREGULIN to the hNcx does not further induce proliferation.**

(**A**) Schematic of experimental workflow. Human fetal tissue pieces (GW 12/13) were isolated and cultured in a free-floating tissue culture (FFTC) in the presence of 50 ng/mL EPIREGULIN for 24 h. (**B**) DAPI staining and immunofluorescence for KI67, SOX2, and TBR2. (**C–E**) Quantifications of KI67, SOX2, and TBR2 in the SVZ. (**F**) Schematic of experimental workflow. Human sliced cortical organoids (10 weeks) were treated with 50 ng/mL EPIREGULIN for 10 days. (**G**) DAPI staining and immunofluorescence for KI67, SOX2, and TBR2 of human cortical organoids derived from the CRTDi004-A iPSC line treated with EPIREGULIN. (**H–M**) Quantifications of KI67, SOX2, and TBR2 in the VZ and SVZ/CP of human cortical organoids. Data information: Scale bars, 100 μm. Bar graphs represent mean values. Error bars represent the SD of four independent fetal tissue samples (**C–E**) or four cortical organoids (**H–M**); Student's t-test; no statistically significant changes were detected. Source data are available online for this figure.

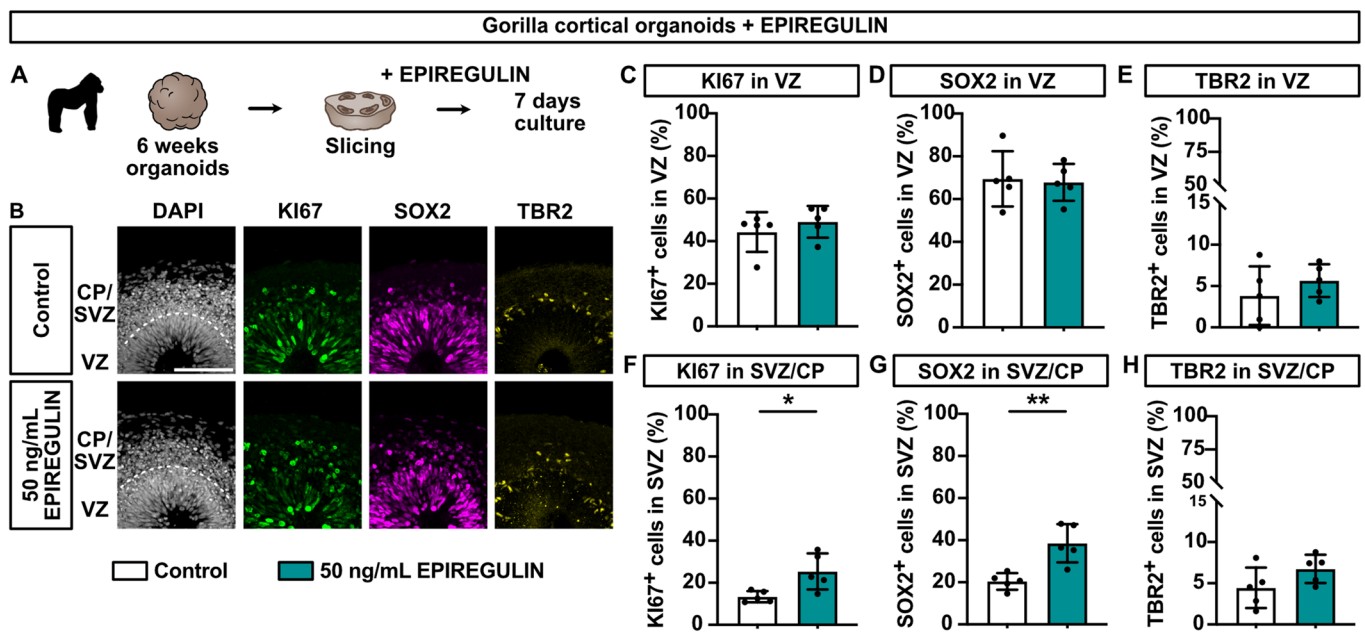

**Figure 5. Addition of EPIREGULIN to gorilla organoids further induces BP proliferation.**

(A) Schematic of experimental workflow. Gorilla-sliced cortical organoids (6 weeks) were treated with 50 ng/mL of EPIREGULIN for 7 days. (B) DAPI staining and immunofluorescence for KI67, SOX2, and TBR2 in gorilla cortical organoids treated with EPIREGULIN. (C–H) Quantifications of KI67, SOX2, and TBR2 in the VZ and SVZ/CP. Data information: Scale bar, 100 µm. Bar graphs represent mean values. Error bars represent SD of five cortical organoids (C–E); **$p < 0.01$, *$p < 0.05$; Student's *t*-test. Source data are available online for this figure.

proliferation with cells cultured with 50 ng/mL EPIREGULIN showing similar growth rates to cells grown under control conditions with 10 ng/mL EGF.

Previously, it was reported that both EGF and FGF elicit the proliferation of freshly isolated mNSC in vitro (Tropepe et al, 1999). Therefore, we repeated the mNSC proliferation assay with cells at passage 0 and with NSCs already adapted to culture conditions (p9) (Fig. EV4A–C). Freshly isolated NSCs (p0) continued to proliferate for three days, independent of EGF and FGF, and could not be stimulated to proliferate further by the addition of EPIREGULIN. This suggests that the freshly isolated mNSCs from E12.5, which mostly represent Sox2-positive apical progenitor cells, maybe in a fully activated proliferative state due to their recent exposure to the pro-proliferative signals of the cerebrospinal fluid (Lehtinen et al, 2011). Cultured mNSCs (p9) adapted to exposure to EGF and FGF completely stopped to proliferate upon withdrawal of EGF, which could be rescued by the addition of EPIREGULIN (Fig. EV4A–C).

Thus, EPIREGULIN can induce the proliferation of mNSC in vitro, and this ability to stimulate proliferation is attenuated by the presence of EGF.

### EPIREGULIN promotes proliferation via the EGF receptor

Next, we aimed to investigate which receptors are required for the EPIREGULIN-induced increase in BP proliferation in the mNcx. For this, we selected two inhibitors that were reported to block signaling via the two major EPIREGULIN receptors: AG-1478, an EGFR tyrosine kinase inhibitor (Martin et al, 2017) and Dacomitinib, an irreversible pan-ErbB receptor tyrosine kinase

inhibitor targeting EGFR and ErbB4 (Xu et al, 2017) (Fig. 6A). We then performed HERO cultures with only the solvent (ethanol; control), solvent and 50 ng/mL EPIREGULIN, or EPIREGULIN and 1 µM of either of the two inhibitors (Fig. 6H,I). Higher concentrations of the inhibitors were found to disrupt tissue integrity (Fig. EV4D). Of note, in the presence of ethanol, the EPIREGULIN-induced increase in abventricular PH3 was smaller (compare Figs. 1I and 6I), but still significant. Inhibition of EGFR alone by AG-1478 resulted in a complete loss of the EPIREGULIN-induced increase in abventricular mitosis (Fig. 6I), whereas ventricular mitosis was unaffected by the inhibitors, at the concentrations used. Likewise, the pan-ErbB inhibitor Dacomitinib abrogated the increase in PH3. Treatment of human cortical organoids with the same inhibitors did not result in significant changes in proliferation after 7 days of treatment (Appendix Fig. S5), in line with reports of variable effects of *Egfr* knockout in the mNcx (Tropepe et al, 1999).

Taken together, these results suggest that EGFR is the key mediator of the EPIREGULIN-mediated increase in BP amplification in the mNcx, whereas ErbB4 appears to not play a major role. Moreover, the presence of different epidermal growth factors modulates the cellular outcome.

### Identification of putative *EREG* enhancer regions in the human genome

Finally, we aimed to dissect the mechanism of differential *EREG* expression across species. Epigenetic modifications, such as histone methylation, contribute to the regulation of gene expression during neocortex development (Bölicke and Albert, 2022; Hirabayashi and

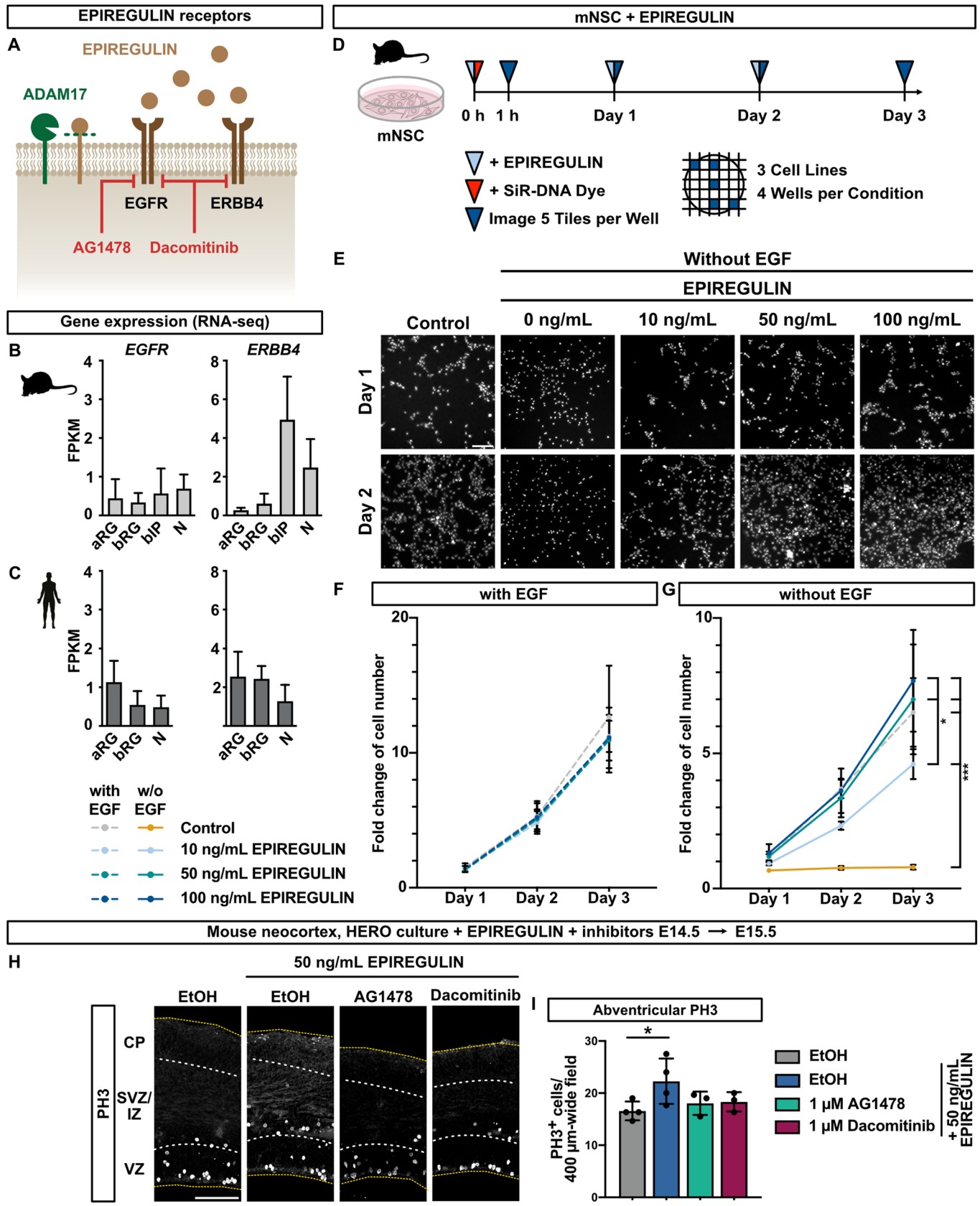

© The Author(s)

**Figure 6. EPIREGULIN mediates BP proliferation via EGFR-signaling.**

(A) Schematic of the EPIREGULIN receptors and inhibitors. (B, C) Expression of EPIREGULIN receptor genes in mNcx and hNcx analyzed by RNA-seq (data from Florio et al, (2015)). (D) Schematic illustration of the experimental workflow. Mouse NSC cultures were treated with different concentrations of EPIREGULIN for 3 days. The SiR-DNA dye was added to the culture to allow imaging of live cells on days 1, 2, and 3. (E) Images of SiR-DNA stained mNSC following treatment with EPIREGULIN in culture medium containing FGF but lacking EGF. The control mNSCs were cultured in a medium with EGF and FGF. (F, G) Quantification of SiR-DNA-positive cells on days 1, 2, and 3 following EPIREGULIN treatment, either with or without EGF, shown as fold change relative to 1 h. (H) Immunofluorescence for PH3 of mNcx slices treated with 50 ng/mL EPIREGULIN and receptor inhibitors for 24 h. (I) Quantifications of abventricular mitotic PH3-positive cells. Data information: Scale bars, 100 μm. Bar and line graphs represent mean values. Error bars represent SD; (B) of 4–5 tissue samples from different litters; (C) of 2–4 tissue samples from different individuals; (F, G) of three different mNSC lines, with each replicate representing the average of five images; (I) of 3–4 embryos from different litters, with each dot representing the average of three images of different sections of the same brain. $*p < 0.05$, $***p < 0.001$; (F, G) two-way ANOVA; (I) one-way ANOVA with Dunnett post hoc test. Source data are available online for this figure.

Gotoh, 2010; Hoffmann and Albert, 2021). In the mNcx, the *Ereg* locus was marked by repressive H3K27me3 and devoid of active modifications, such as H3K4me3 and H3K27ac (Fig. EV1E). To test whether the H3K27me3 marking was actively involved in repressing the *Ereg* gene in the mNcx, we established a CRISPR/Cas9-based epigenome editing system (Albert et al, 2017; Kearns et al, 2015) to remove the H3K27me3 by targeting the catalytic jumonji domain of the histone demethylase KDMB6 (referred to as JMJC_6B) (Agger et al, 2007) to sites up- and downstream of the *Ereg* transcription start site (TSS) (Fig. EV5A,B). To perform the epigenome editing, mNSCs were employed, as they recapitulate the H3K27me3 patterns observed in aRG of the mNcx (Fig. EV5C). Epigenome editing in the mNSCs using dCas9-JMJC_6B resulted in a reduction of H3K27me3 at the *Ereg* locus, but not at the unrelated *Hoxb5* and *Eomes* genes (Fig. EV5D,E). However, this reduction in repressive epigenetic modifications alone did not result in an upregulation of *Ereg* gene expression (Fig. EV5F,G), suggesting that the mouse *Ereg* locus is in a fully repressed rather than poised state.

Given the contribution of gene regulatory networks to evolutionary changes in development (Davidson and Erwin, 2006), we next aimed to identify cis-regulatory elements (CREs) that may contribute to the expression of *EREG* in the hNcx. To this end, we examined ATAC-seq and H3K27ac ChIP-seq data of the fetal hNcx (de la Torre-Ubieta et al, 2018; Reilly et al, 2015) and found 11 regions of open chromatin within 100 kb up- and downstream of the *EREG* gene (excluding TSS regions) that may represent putative active enhancers (Fig. 7A). Two additional genes are located within this genomic region, *AREG* and *EPGN*, which also encode for growth factors known to bind to EGFR (Appendix Fig. S4A). However, based on chromatin (Fig. 7A) and RNA-seq (Appendix Fig. S4B, and Camp et al, (2015)) data, they are likely not expressed in the fetal hNcx. Using LiftOver conversion of the human genomic coordinates to the mouse genome, we identified the orthologous mouse genomic regions. Interestingly, of the 11 putative *EREG* CREs, only one shows a small ATAC-seq peak in the mNcx, whereas the other orthologous regions lack H3K27ac or ATAC-seq peaks in the mNcx (Fig. 7B) (Gorkin et al, 2020). Whereas the putative CRE1 is not conserved in the mouse, the other ten putative CREs are at least partially conserved in the mouse, yet show higher sequence divergence compared to primate species (Fig. 7C; Appendix Fig. S6).

To test whether these putative enhancer regions display enhancer activity, we cloned two of the human CRE regions (216 bp) and their orthologous mouse sequences, respectively, into a plasmid containing a minimal promoter upstream of a fluorescent reporter (mScarlet). These plasmids were then electroporated into

human cortical organoids, along with a control plasmid expressing GFP, and enhancer activity was observed after 3 days (Fig. 7D). Indeed, human CRE6 was able to drive reporter gene expression in human cortical organoids (Fig. 7E). Comparing hCRE6 and mCRE6, higher levels of mScarlet were observed for hCRE6 compared to the orthologous mouse sequence (Fig. 7F,G). Human CRE9 showed low mScarlet expression in the VZ/SVZ, yet, in this case, the orthologous mouse sequence showed even higher activity (Fig. 7F,G). The enhancer activity assay therefore confirms that at least one of the human putative CREs identified based on their open chromatin configuration has the potential to enhance gene expression from a reporter plasmid. For the endogenous locus that is embedded in chromatin, the epigenetic state of the locus may further contribute to tuning the activity of the CRE.

Thus, based on the open chromatin status, which correlates with gene expression in the hNcx, and closed chromatin correlating with repression in the mNcx, respectively, these 11 genomic regions may represent putative enhancers contributing to differential *EREG* expression in the developing neocortex of different mammalian species.

## Discussion

Here, we show that the growth factor EPIREGULIN provides a pro-proliferative signal in the human SVZ niche that is received by BPs and promotes their amplification. EPIREGULIN is expressed in the apical and basal radial glia of the human neocortex, where it is required for basal progenitor proliferation. EPIREGULIN is also expressed in macaque and gorilla, both gyrencephalic primates, but is not detectable in the mouse, a species with a small lissencephalic neocortex. In the mNcx, the addition of EPIREGULIN promotes BP amplification, of both bIPs and bRG. This may be of evolutionary relevance as BPs have high self-renewing capacities, are highly abundant in species with a large gyrencephalic neocortex, and are associated with increased neocortex size, neuron number, and cortical folding (Dehay et al, 2015; Florio and Huttner, 2014; Llinares-Benadero and Borrell, 2019; Lui et al, 2011; Zhou et al, 2023). Human mutations affecting BP generation and amplification often result in microcephaly (Baala et al, 2007; Johnson et al, 2018). Further addition of EPIREGULIN to the hNcx did not increase BP proliferation to even higher levels, suggesting that the system may have been saturated, either at the level of receptors (see below) or that maximum BP proliferation had already been reached. As the addition of EPIREGULIN to the gorilla Ncx resulted in increased BP proliferation, this suggests that

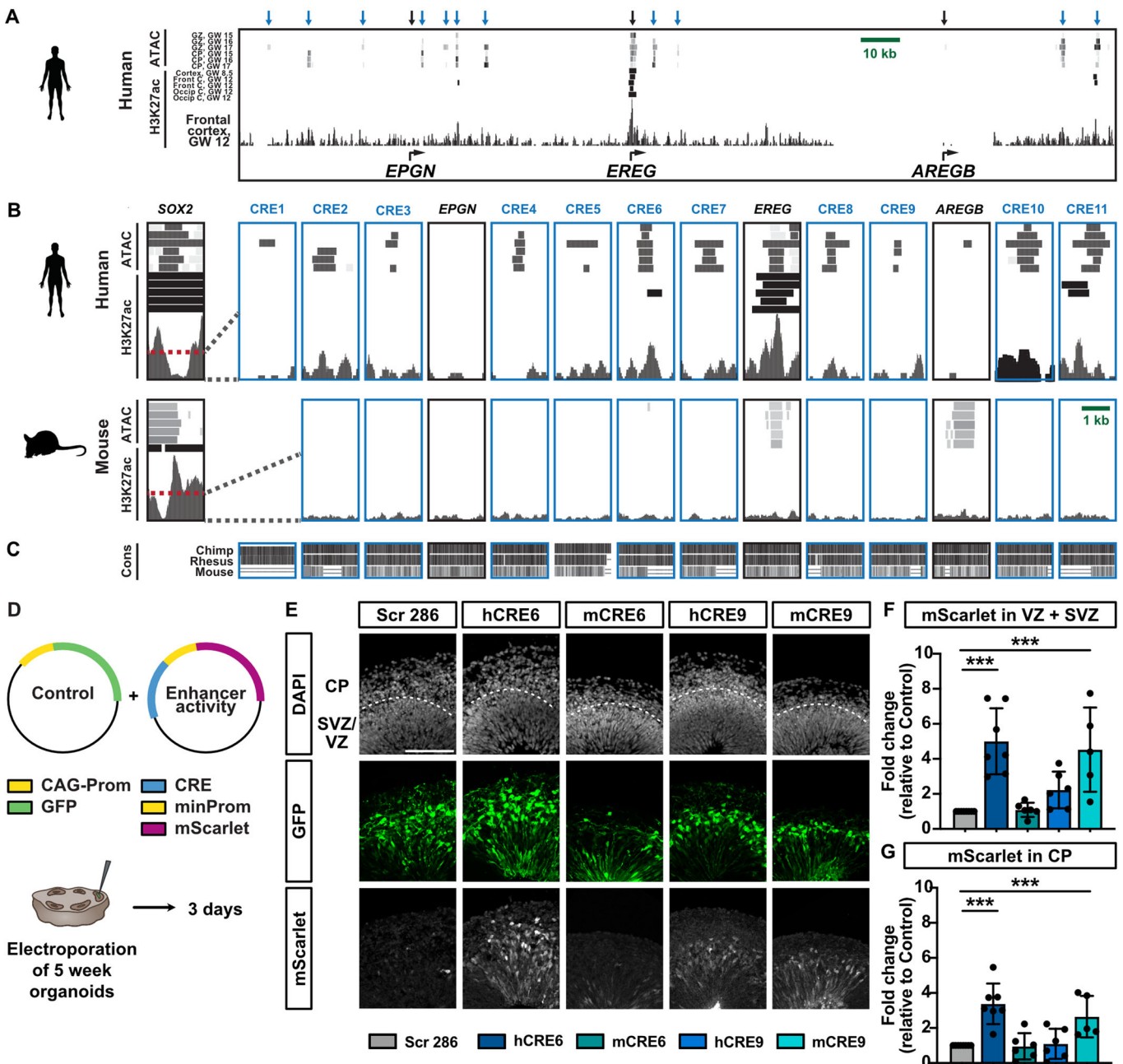

**Figure 7. Putative enhancer regions for *EREG* differential gene expression.**

(A) ATAC-seq (de la Torre-Ubieta et al, 2018) and H3K27ac ChIP-seq (Reilly et al, 2015) peaks around the *EREG* gene (±100 kb) in the hNcx at given ages. Black arrows, TSS; blue arrows, putative CREs. (B) Zoom in on the TSS and CRE regions indicated by arrows in (A), relative to open chromatin marks at the *SOX2* gene, for humans (top) and mouse (bottom, (Gorkin et al, 2020)). Window size, 2 kb. (C) Evolutionary conservation from 100 vertebrate species (Blanchette et al, 2004) for the indicated species. (D) Schematic of experimental workflow. Human sliced cortical organoids (6 weeks) were electroporated with a plasmid encoding GFP and a plasmid with a putative CRE upstream of a minimal promotor driving mScarlet expression, and analyzed after 3 days. (E) DAPI staining and immunofluorescence for GFP and mScarlet of an electroporated human cortical organoid to test for enhancer activity of mouse and human CRE6 and CRE9. (F, G) Quantification of mScarlet signal intensity in the VZ + SVZ and CP, relative to Scr control. Data information: Scale bar, 100 μm. Bar graphs represent mean values. Error bars represent the SD of six organoids from two batches. ***p < 0.001; one-way ANOVA with Dunnett post hoc test. Source data are available online for this figure.

the impact of EPIREGULIN on BP amplification depends on the prevailing EPIREGULIN concentration in a given species and/or on the species-specific context, as reported for other genes (Van Heurck et al, 2022), and may provide a mechanism to tune progenitor amplification across species.

Based on transcriptomic studies, it was proposed that bRG maintains the OSVZ as a proliferative stem cell niche by locally producing growth factors and ECM to activate self-renewal pathways (Fietz et al, 2010; Kalebic and Huttner, 2020; Pollen et al, 2015). In particular, the germinal zones, including the mVZ,

hVZ, hISVZ, and hOSVZ, but not the mSVZ, preferentially express genes related to ECM formation (Fietz et al, 2010; Florio et al, 2015; Pollen et al, 2015). Indeed, targeted activation of the ECM receptor integrin αvβ3 on BPs in the mNcx promoted BP expansion (Stenzel et al, 2014). Moreover, conditional expression of Sox9, a transcription factor that induces expression of ECM components, resulted in cell-non-autonomous stimulation of BP proliferation linked to increased neuron production (Guven et al, 2020). In addition to proliferation, specific ECM components with higher expression in human than mouse NPCs were shown to cause folding of the hNcx (Long et al, 2018). Here, we provide functional evidence that a growth factor expressed in primate bRG promotes BP amplification in the SVZ of the mNcx and contributes to BP proliferation in the hNcx. In gyrencephalic species, BPs were shown to have a higher number of processes enabling them to receive extrinsic pro-proliferative signals linked to an increased BP proliferative capacity (Kalebic et al, 2019). Our data suggests that EPIREGULIN provides such a pro-proliferative signal in the human SVZ niche and stimulates BPs amplification.

Moreover, transcriptional signatures of bRG have been reported to be enriched in cells from human primary glioblastoma, suggesting that developmental programs are reactivated in tumor cells (Bhaduri et al, 2020; Pollen et al, 2015). Indeed, *EREG* expression is higher in high- compared to low-grade glioma and loss of *EREG* correlates with increased survival. This is in line with the induction of NPC proliferation by EPIREGULIN described here, even though EPIREGULIN may also contribute to other aspects of tumor progression, such as angiogenesis or inflammation (Riese and Cullum, 2014).

Mechanistically, the addition of EPIREGULIN did not appear to alter transcriptional programs but rather resulted in altered cell type proportions, as has also been observed for other factors that regulate brain size, such as abnormal spindle-like microcephaly-associated (*ASPM*), the most common recessive gene associated with human primary microcephaly (Johnson et al, 2018). Both the mNSC proliferation assay in the presence/absence of EGF and the inhibition of EGFR suggests that EPIREGULIN mediates NPC proliferation, resulting in altered cell type proportions, via EGFR-mediated signaling. Among the main signaling cascades activated by EGFR ligand binding are the Ras-Raf-MEK-ERK, STAT, and PI3K-AKT-mTOR pathways, all of which are important for the regulation of NPC proliferation, differentiation, and progenitor pool maintenance (Kalebic and Huttner, 2020; Pollen et al, 2015; Romano and Bucci, 2020). In contrast to EGF, which represents a high-affinity ligand reported to bind cell-surface EGFR with an apparent Kd of 0.1–1 nM, EPIREGULIN represents a low-affinity ligand with 10- to 100-fold weaker binding. Interestingly, EGF was reported to promote transient EGFR activation resulting in transient ERK activation, whereas the low-affinity ligand EPIR-EGULIN was counterintuitively shown to promote both sustained EGFR activation and sustained ERK activation (Freed et al, 2017). The distinct biological responses were independent of any effects on other ErbB family receptors, which is in line with our finding that both specific EGFR inhibition and pan-ErbB inhibition resulted in a similar loss of the EPIREGULIN-mediated increase in basal mitosis. Thus, ErbB4 does not appear to play a major role in the EPIREGULIN response in the mNcx, despite its high expression in BPs.

The stimulation of BP amplification by EPIREGULIN was dose-dependent, reaching a maximum of 50 ng/mL in the mNcx. At 100 ng/mL, EPIREGULIN did not induce any increase in proliferation in the mNcx. Similarly, human BPs, in contrast to gorilla BPs, were not receptive to further stimulation of proliferation by EPIREGULIN. This suggests that levels of individual growth factors and their combinations present in the tissue dictate cellular behavior. Moreover, different desensitization strategies have been reported for ErbB receptors and their distinct ligands, which may further attenuate the response to high levels of those ligands (Yamamoto et al, 2014).

Research has only begun to unravel the genomic basis for neocortex evolution. Recently, a handful of human-, hominid- and primate-specific genes have emerged as potential adaptive evolutionary innovations implicated in neocortex evolution (Florio et al, 2017; Libe-Philippot and Vanderhaeghen, 2021; Pollen et al, 2023). However, species-specific differences are likely to extend beyond novel genes, and there is compelling evidence emerging that non-coding regions of the genome contribute to evolutionarily relevant inter-species differences in gene expression (Doan et al, 2018; Espinos et al, 2022; Silver, 2016). Among the few examples of regulatory changes that were functionally investigated is *HARE5*, a human-accelerated regulatory enhancer that drives earlier and more robust brain expression of *FZD8*, a WNT receptor, resulting in accelerated cell cycle and enlarged brains (Boyd et al, 2015). Moreover, the deletion of a 15-base pair region in a regulatory element of *GPR56* selectively influences NPC proliferation and neocortex folding, potentially implicated in neocortex evolution (Bae et al, 2014). Here, we have identified CRE regions adjacent to human *EREG* that are marked by open chromatin in the human fetal cortex, suggesting that they may represent putative active enhancers. Indeed, one of the hCRE regions that we tested in enhancer activity assays was able to promote significant reporter gene expression. The homologous mouse regions could be identified for 10 of the 11 regions, yet show higher sequence divergence compared to other primates and lack signs of active chromatin. Unlike the microRNA *miR-3607*, which has been linked to reduced progenitor amplification in rodents and was secondarily lost during evolution (Chinnappa et al, 2022), most putative *EREG* CREs are more likely to represent primate-specific innovations based on their conservation, rather than a secondary loss in rodents. Of the mCREs that we tested in enhancer activity assays, one showed reduced activity compared to humans, whereas the other sequence showed even higher activity. Given that the endogenous CRE regions are embedded in chromatin, local epigenetic modifications and chromatin compaction may further contribute to regulating the accessibility of factors and, thus, the activity of the regions. Evolutionary changes in enhancer activity have previously been implicated in the regulation of conserved developmental pathways during cortical evolution (Reilly et al, 2015). Overall, these potential CREs represent interesting candidate enhancers that may contribute to inter-species differences in *EREG* expression.

Taken together, we propose that EPIREGULIN presents a pro-proliferative signal for BP cells in primates to sustain a proliferative niche in the subventricular zone of the developing neocortex, which may have important evolutionary significance as a tunable mechanism to control neocortex size across mammalian species.

# Methods

### Reagents and tools table

| Reagent/Resource | Reference or source | Identifier or catalog number |
|---|---|---|
| **Experimental Models** | | |
| Subcloning Efficiency™ DH5α competent cells | Invitrogen | Cat. # 18265017 |
| ElectroMAX Stbl4 Competent Cells | Thermo Fisher | Cat. # 11635018 |
| GW 12/13 human fetal brain tissue | This study | N/A |
| Human: CRTDi004-A iPSC | (Völkner et al, 2022) | https://hpscreg.eu/cell-line/CRTDi004-A |
| Human: HPSI0114i-kolf_2 iPSC | Welcome Trust Sanger Institute (HipSci) | RRID:CVCL_AE29 |
| Gorilla: GC1 iPSC | (Wunderlich et al, 2014) | N/A |
| Mouse: NSC | This paper | N/A |
| Mouse: C57BL/6 J | Charles River | RRID:IMSR; JAX:000664 |
| Mouse: *Tubb3*::GFP | (Attardo et al, 2008) | N/A |
| **Recombinant DNA** | | |
| pCAG-GFP | (Florio et al, 2015) | N/A |
| pCas9-JMJC_6B-T2A-EGFP-gRNAs | This paper | N/A |
| pCAG-Ereg | This paper | N/A |
| Candidate cis-regulatory elements | This paper | Table EV3 |
| **Antibodies** | | |
| Goat anti-Sox2 (1:200) | R&D Systems AF2018 | RRID:AB_355110 |
| Rabbit anti-Ki67 (1:500) | Abcam ab15580 | RRID:AB_443209 |
| Mouse anti-PCNA (1:500) | Millipore CBL407 | RRID:AB_93501 |
| Mouse anti-PhVim (1:200) | Abcam ab22651 | RRID:AB_447222 |
| Rat anti-PH3 (1:500) | Abcam ab10543 | RRID:AB_2295065 |
| Rat anti-Ctip2 (1:250) | Abcam ab18465 | RRID:AB_2064130 |
| Rabbit anti-TBR2 (1:250) | Abcam ab23345 | RRID:AB_778267 |
| Chicken anti-GFP (1:2,000)_ | Abcam ab13970 | RRID:AB_300798 |
| Rat anti-RFP (1:500) | proteintech | RRID:AB_2336064 |
| Mouse anti-Tuj1 (1:300) | Abcam ab78078 | RRID:AB_2256751 |
| Rabbit anti-EREG (1:5,000) | Abcam ab233512 | |
| Rabbit anti-EPIREGULIN (1:5,000; 1:10,000) | Cell Signaling #12048 | RRID:AB_2797808 |
| Rabbit anti-Histone H3K27me3 (1:50) | Cell Signaling 9733 | RRID:AB_2616029 |
| Donkey anti-Rat IgG, Alexa Fluor 488 conjugated (1:1,000) | Invitrogen A-21208 | RRID:AB_141709 |
| Donkey anti-Rat IgG, Alexa Fluor 647 conjugated (1:1,000) | Invitrogen A-78947 | RRID:AB_2910635 |
| Goat anti-Rabbit IgG, Alexa Fluor 405 conjugated (1:1,000) | Invitrogen A-31556 | RRID:AB_221605 |
| Donkey anti-Rabbit IgG, Alexa Fluor 488 conjugated (1:1,000) | Invitrogen A-21206 | RRID:AB_2535792 |

| Reagent/Resource | Reference or source | Identifier or catalog number |
|---|---|---|
| Donkey anti-Rabbit IgG, Alexa Fluor 647 conjugated (1:1000) | Invitrogen A-31573 | RRID:AB_2536183 |
| Donkey anti-Goat IgG, Alexa Fluor 555 conjugated (1:1,000) | Invitrogen A-21432 | RRID:AB_2535853 |
| Donkey anti-Mouse IgG, Alexa Fluor 488 conjugated (1:1,000) | Invitrogen A-21202 | RRID:AB_141607 |
| Donkey anti-Mouse IgG, Alexa Fluor 555 conjugated (1:1,000) | Invitrogen A-31570 | RRID:AB_2536180 |
| Donkey anti-Mouse IgG, Alexa Fluor 647 conjugated (1:1,000) | Invitrogen A-31571 | RRID:AB_162542 |
| Donkey anti-Chicken IgG, Alexa Fluor 488 conjugated (1:1,000) | Jackson Immuno Research 703-545-155 | RRID:AB_2340375 |
| Goat anti-rabbit IgG H&L HRP (1:3,000) | Abcam ab6721 | RRID:AB_955447 |
| Goat anti-mouse IgG H&L HRP (1:10,000) | Abcam ab205719 | RRID:AB_2755049 |
| PE Mouse Anti-Sox2 (1:40) | BD Biosciences 562195 | RRID:AB_10895118 |
| V450 Mouse anti-Sox2 (1:40) | BD Biosciences 561610 | RRID:AB_10712763 |
| **Oligonucleotides** | | |
| gRNAs | This paper | Table EV1 |
| Primers | This paper | Table EV2 |
| Oligo(dT)18 Primer | Thermo Scientific | Cat. # SO132 |
| **Chemicals, Enzymes and other Reagents** | | |
| Alt-R S.p. HiFi Cas9 Nuclease V3 | Integrated DNA Technologies | Cat. # 1081060 |
| Alt-R CRISPR-Cas9 tracrRNA | Integrated DNA Technologies | Cat. # 1072533 |
| Recombinant Human EPIREGULIN Protein 1195-EP (without carrier) | R&D Systems | Cat. # 1195-EP-025/CF |
| Dimethyl sulfoxide (DMSO) | Sigma Aldrich | Cat. # D2650-100ML |
| AG-1478 (Synonyms: Tyrphostin AG-1478; NSC 693255) | BIOZOL Diagnostica Vertrieb GmbH | Cat. # MCE-HY-13524 |
| Dacomitinib | BIOZOL Diagnostica Vertrieb GmbH | Cat. # MCE-HY-13272 |
| Rat serum | Charles River Laboratories Japan | Crl:CD(SD) |
| Cellmatrix type I-A | Kyowa chemical products | Cat. # 631-00651 |
| DMEM/F12, HEPES | Gibco | Cat. # 31330095 |
| Poly-D-Lysine | Invitrogen | Cat. # A3890401 |
| Laminin from Engelbreth–Holm–Swarm murine sarcoma basement membrane | Sigma Aldrich | Cat. # L2020 |
| Bovine serum albumin (BSA) | Sigma Aldrich | Cat. # A2153 |
| Fetal bovine serum (FBS) | Sigma Aldrich | Cat. # F7524 |
| Heparin sodium salt from porcine intestinal mucosa | Sigma Aldrich | Cat. # H4784 |
| MEM Non-essential amino acids solution (100X) | Gibco | Cat. # 11140050 |
| GlutaMAX™ Supplement | Gibco | Cat. # 35050038 |
| Human EGF Recombinant Protein | Gibco | Cat. # PHG0311 |

| Reagent/Resource | Reference or source | Identifier or catalog number |
|---|---|---|
| Human FGF-basic (FGF-2/bFGF) (aa 10-155) Recombinant Protein | Gibco | Cat. # PHG0021 |
| B-27™ Supplement (50x), serum-free | Gibco | Cat. # 17504044 |
| N-2 Supplement (100X) | Gibco | Cat. # 17502048 |
| Collagenase type IV | Gibco | Cat. # 17104019 |
| TrypLE™ Express Enzyme (1X), no phenol red | Gibco | Cat. # 12604021 |
| ReLeSR™ | Stem Cell Technologies | Cat. # 05873 |
| Rock inhibitor, Y-27632 (2HCl) | Stem Cell Technologies | Cat. # 72308 |
| mTeSR1 Complete Kit | Stem Cell Technologies | Cat. # 85850 |
| StemFlex™ Medium | Gibco | Cat. # A3349401 |
| Matrigel hESC-Qualified Matrix, LDEV-free | Corning | Cat. # 354277 |
| STEMdiff™ Cerebral Organoid Kit | Stem Cell Technologies | Cat. # 8570 |
| STEMdiff™ Cerebral Organoid Maturation Kit | Stem Cell Technologies | Cat. # 8571 |
| Amphotericin B (Fungizone) | Gibco | Cat. # 11520496 |
| KnockOut™ Serum Replacement | Gibco | Cat. # 10828010 |
| Dorsomorphine | Stem Cell Technologies | Cat. # 72102 |
| A83-01 | Stem Cell Technologies | Cat. # 72022 |
| CHIR-99021 | STEMCELL Technologies | Cat. # 72052 |
| SB-431542 | Stem Cell Technologies | Cat. # 72232 |
| Insulin Solution, Human recombinant | Sigma Aldrich | Cat. # I9278 |
| Matrigel Growth Factor Reduced (GFR) Basement Membrane Matrix, LDEV-free | Corning | Cat. # 354230 |
| Recombinant Human/Murine/Rat BDNF | PeproTech | Cat. # 450-02 |
| Recombinant Human GDNF | PeproTech | Cat. # 450-10 |
| Ascorbic acid | Sigma Aldrich | Cat. # 1043003 |
| Dibutyryl-cAMP | Stem Cell Technologies | Cat. # 73882 |
| UltraPure™ low melting point agarose | Invitrogen | Cat. # 16520100 |
| 5-Brom-2'-deoxyuridine (BrdU) | Sigma Aldrich | Cat. # B5002 |
| Saponin | Sigma Aldrich | Cat. # 47036 |
| Thermo Scientific™ Halt™ Protease-Inhibitor-Cocktail (100x) | Thermo Scientific | Cat. # 10516495 |
| Superscript III Reverse Transcriptase | Invitrogen | Cat. # 18080044 |
| LightCycler 480 SYBR Green I Master | Roche | Cat. # 4887352001 |
| TURBO DNA-free™ Kit | Invitrogen | Cat. # AM1907 |
| Maxima H Minus Reverse Transcriptase (200 U/μl) | Thermo Scientific | Cat. # EP0753 |
| UltraPure™ BSA (50 mg/mL) | Invitrogen | Cat. # AM2618 |
| Collagen I-based hydrogel (Cellmatrix type I-A) | Kyowa chemical products | Cat. # 631-0065 |
| DAPI | Roche | Cat. # 10236276001 |

| Reagent/Resource | Reference or source | Identifier or catalog number |
|---|---|---|
| SiR-DNA kit | Spirochrome | Cat. # SC007 |
| Lipofectamine 2000 | Thermo Scientific | Cat. # 11668019 |
| Pierce™ Detergent-Compatible Bradford Assay Kit | Thermo Scientific | Cat. # 23246 |
| PVDF Transfer Membranes, 0.45 μm | Thermo Scientific | Cat. # 88518 |
| SuperSignal™ West Pico PLUS Chemiluminescent Substrate | Thermo Scientific | Cat. # 34579 |
| **Software and Algorithms** | | |
| Fiji/ImageJ | Fiji/ImageJ | https://imagej.nih.gov/ij/ |
| Prism (8.4.3) | GraphPad software | N/A |
| Geneious Prime® (2019.2.1) | Biomatters Ltd. | N/A |
| Adobe Illustrator (27.2) | Adobe | N/A |
| Affinity Photo + Designer (1.10.5.1342) | Serif Ltd. | N/A |
| FACSDiva (8.0.2) | BD Biosciences | N/A |
| FACSChorus | BD Biosciences | N/A |
| FastQC (v0.11.6) | Babraham Bioinformatics | https://www.bioinformatics.babraham.ac.uk/projects/fastqc/ |
| **Other** | | |
| Plasmid Maxi Kit | QIAGEN | Cat. # 12163 |
| GeneJET Plasmid-Midiprep-Kit | Thermo Scientific | Cat. # K0481 |
| P3 Primary Cell 4D-Nucleofector™ X Kit L (mNSC) | Lonza | Cat. # V4XP-3024 |
| P3 Primary Cell 4D-Nucleofector™ X Kit S (hiPSC) | Lonza | Cat. # V4XP-3032 |
| RNeasy Mini Kit | QIAGEN | Cat. # 74104 |
| QIAquick Gel Extraction Kit | QIAGEN | Cat. # 28106 |
| MACS Neural Tissue Dissociation Kit (P) | Miltenyi Biotec | Cat. # 130-092-628 |
| Quick-DNA/RNA Microprep Plus Kit | Zymo Research | Cat. # D7005 |
| Quick-RNA FFPE MiniPrep kit | Zymo Research | Cat. # R1008 |
| Low Cell Number Chromatin Immunoprecipitation kit | Diagenode | Cat. # C01010072 |
| KAPA Library Quantification Kits - Complete kit | Roche | Cat. # 07960298001 |
| LowCell ChIP kit | Diagenode | Cat. # C01010072 |
| microTUBE AFA Fiber Pre-Slit Snap-Cap 6x16mm | Covaris | Cat. # 520045 |
| DNA LoBind microcentrifuge tubes, 1.5 mL | Eppendorf | Cat. # 0030108051 |
| 35 mm glass bottom dish, No. 1.5 coverslip, 14 mm glass diameter, uncoated | MatTek Cooperation | Cat. # P35G-1.5-14-C |
| Phylogenetic pictures | N/A | https://www.phylopic.org/ |
| Raw and analyzed RNA-seq data | This paper | NCBI GEO: GSE228007 |
| RNA-seq | (Florio et al, 2015) | NCBI GEO: GSE65000 |
| RNA-seq; ChIP-seq (H3K4me3, H3K27me3) | (Albert et al, 2017) | NCBI GEO: GSE90695 |
| ATAC-seq | (de la Torre-Ubieta et al, 2018) | NCBI GEO: GSE95023 |
| ChIP-seq (H3K27ac) | (Reilly et al, 2015) | NCBI GEO: GSE63649 |

## Methods and protocols

### Mice

All experimental procedures were conducted in agreement with the German Animal Welfare Legislation after approval by the Landesdirektion Sachsen (licenses DD24.1-5131/476/8; 25-5131/496/19). Animals were kept on a 12-h/12-h light/dark cycle with food and water ad libitum. All mice were wildtype mice from the inbred C57BL/6 J strain, except for the isolation of neural cell populations for RNA-seq, where the transgenic *Tubb3*::GFP line (Attardo et al, 2008) was used to identify neurons based on GFP expression. Embryonic day (E) 0.5 was set as noon on the day on which the vaginal plug was observed. All experiments were performed in the dorsolateral telencephalon of mouse embryos, at a medial position along the rostro-caudal axis. The developmental time point E14.5 of experimental procedures corresponds to a mid-neurogenic stage, when the production of upper-layer neurons has started. The sex of embryos was not determined, as it is not likely to be of relevance to the results obtained in the present study. Mouse NSCs were isolated from E11.5/12.5 embryos as previously described (Schmitz et al, 2011; Schütze et al, 2023).

### Human fetal brain tissue

Human fetal brain tissue was obtained from the Department of Gynecology and Obstetrics, University Clinic Carl Gustav Carus of the Technische Universität Dresden, following elective pregnancy termination and informed written maternal consents, and with approval of the local University Hospital Ethical Review Committee (IRB00001473; IORG0001076; ethical approval number EK 355092018), in accordance with the Declaration of Helsinki. The age of fetuses ranged from gestation week (GW) 10 to 13, as assessed by ultrasound measurements of crown-rump length and other standard criteria of developmental stage determination. These developmental time points correspond to an early/mid-neurogenic stage, when the OSVZ expands, and the production of upper-layer neurons starts. Due to the protection of data privacy, the sex of the human fetuses, from which the hNcx tissue was obtained, cannot be reported. The sex of the human fetuses is not likely to be of relevance to the results obtained in the present study. The fetal hNcx tissue samples used in this study reported no health disorders. Fetal human brain tissue was dissected in Tyrode's solution and used immediately (within 1 h) for further culture or fixation.

### Human and primate-induced pluripotent stem cell lines

All experiments involving hiPSCs were performed in accordance with the ethical standards of the institutional and/or national research committee, as well as with the 1964 Helsinki Declaration, and approved by the University Hospital Ethical Review Committee (IRB00001473; IORG0001076; ethical approval number SR-EK-456092021). Human cortical organoids were generated using the previously generated human induced pluripotent stem cell (iPSC) lines CRTDi004-A (Völkner et al, 2022) and HPSI0114i-kolf_2 (Welcome Trust Sanger Institute, HipSci), derived from healthy donors. The human iPSC lines were maintained on Matrigel-coated (Corning, 354277) culture dishes in mTeSR1 and passaged using TrypLE Express enzyme (CRTDi004-A) or ReLeSR (HPSI0114i-kolf_2) (Schütze et al, 2023).

Gorilla organoids were created using the previously generated gorilla iPSC line GC1 (Wunderlich et al, 2014). Matrigel-coated culture dishes and StemFlex medium was used to maintain the line, and passaging was performed using ReLeSR.

### Ex vivo experiments on mNcx

Mouse organotypic slice cultures were performed from E14.5 brains, as previously described (Wong et al, 2014). The meninges were surgically removed from the brains, brains were embedded in 3% low melting point agarose (Invitrogen, 16520100) and cut on a vibratome into 250-μm-thick slices. Slices were embedded into a collagen I-based hydrogel and placed in 35 mm Petri dishes with a 14 mm microwell. Slice culture medium, either without (control) or with 10, 50, or 100 ng/mL human recombinant EPIREGULIN, dissolved in PBS, were added to the slices. Slice culture medium (SCM) contained Neurobasal medium supplemented with 10% rat serum, 20 μM glutamine, 1X penicillin/streptomycin, 1X N-2 and 1X B-27 supplement, and 0.1 M HEPES/NaOH, pH 7.2. Slices were cultured for 24 h in a humidified incubation chamber supplied with 40% $O_2$, 5% $CO_2$, and 55% $N_2$ at 37 °C.

Mouse brain hemispheres were cultured under rotation conditions (HERO), as previously described (Schenk et al, 2009). Briefly, E14.5 brains were dissected in cold PBS. Before separating the hemispheres, the meninges were removed. Hemispheres were then transferred into a rotating flask with 1.5 mL of SCM. SCM was either without (control) or with 50 ng/mL human recombinant EPIREGULIN. For the inhibitor experiments, 1 μM of AG-1478 or Dacomitinib were added additionally. HERO culture was performed for 24 h in a Whole Embryo Culture System (Nakayama, 10-0310) in a humidified atmosphere of 40% $O_2$, 5% $CO_2$, and 55% $N_2$ at 37 °C.

### Mouse NSC culture

Mouse NSCs were plated at a density of $10^5$ cells/mm² on previously coated six-well plates (poly-D-lysine and laminin) and grown in NSC medium (1:1 mixture of DMEM/F12 and Neurobasal medium, supplemented with 10 and 20 ng/mL of epidermal and fibroblast growth factors, respectively, (EGF), 1X N-2 and 1X B-27 supplements, 1X penicillin/streptomycin, 1X Sodium-Pyruvate, 1X GlutaMAX, 1X MEM-NEAA, 4 mg/mL Heparin, 5 mM HEPES, 0.01 mM 2-mercaptoethanol, and 100 mg/L BSA. NSCs are passaged using trypsin-EDTA and soybean trypsin inhibitor and kept in standard conditions at 37 °C and 5% $CO_2$.

Mouse NSCs were isolated as previously described (Schmitz et al, 2011). Briefly, the E11.5 cerebral cortex was dissected and treated with trypsin-EDTA at 37 °C for 20 min. A soybean trypsin inhibitor was added, and cells were dissociated mechanically. Fresh mNSC were plated and cultured as described above. For the proliferation assay, freshly isolated (passage 0) and p9 mNSCs were plated in 24 well plates in conditions with and without EGF and EPIREGULIN. Proliferation was analyzed every day for 3 days.

### Ex vivo experiments on hNcx

Fetal hNcx tissue was cultured under free-floating tissue culture (FFTC) conditions, as previously described (Long et al, 2018). Fetal hNcx tissue was placed into 1.5 mL of human SCM containing Neurobasal medium supplemented with 10% knockout serum replacement, 20 μM glutamine, 1X penicillin/streptomycin, N-2 and B-27 supplements, and 0.1 M HEPES/NaOH at pH 7.2, either without (control) or with 50 ng/mL human recombinant EPIREGULIN, and cultured for 24 h under rotation conditions in a humidified atmosphere of 40% $O_2$, 5% $CO_2$, and 55% $N_2$ at 37 °C.

### Generation of human and gorilla organoids

Human and gorilla cortical organoids were generated according to the sliced neocortical organoid (SNO) method, as previously

described in detail (Qian et al, 2020; Schütze et al, 2023). Briefly, hiPSC colonies of about 1.5 mm in size were detached with collagenase and transferred to ultra-low attachment six-well plates (Corning, 3471), containing forebrain medium 1, to form 3D aggregates. On day 7, embryoid bodies (EBs) were embedded in Matrigel (Corning, 354230) with about 20 EBs per Matrigel cookie. From day 7 to 14, EBs were cultured in forebrain maturation medium 2. At day 14, organoids were mechanically released from the Matrigel cookie and further cultured in six-well ultra-low attachment plates in forebrain medium 3 on an orbital shaker at 100 rpm. From day 35, 1% v/v Matrigel was added to forebrain medium 3. At 6 weeks, organoids were embedded in 3% low melting point agarose and cut on a vibratome into 500-μm-thick slices, supporting an enhanced supply of oxygen and nutrients throughout the organoids. Slicing of organoids was repeated at 10 weeks. From day 72, organoids were cultured in forebrain medium 4. For treatments, 50 ng/mL EPIREGULIN were added directly after slicing of organoids, either at 6 weeks or 10 weeks, to avoid issues with penetration. EPIREGULIN was added every other day for a period of 4 or 7 days (6-week organoids) or 10 days (10-week organoids). For the inhibitor experiments, 1 or 10 μM of AG-1478 or Dacomitinib were added directly after slicing at 6 weeks. Organoids were treated for 7 days and fresh inhibitors were added with every media change.

Additionally, gorilla cerebral organoids were generated according to the cerebral organoid method, using the STEMdiff Cerebral Organoid Kit, based on (Lancaster and Knoblich, 2014; Lancaster et al, 2013).

### Guide RNA design and validation

For CRISPR/Cas9-mediated targeting of the human *EREG* gene, the guide RNAs (gRNAs; see Table EV1) were designed using Geneious. The previously published gRNA for *LacZ* (Kalebic et al, 2016; Platt et al, 2014) was used as a control. gRNAs were purchased as custom crRNAs and assembled into functional gRNA duplexes with the tracrRNA according to the manufacturer's instructions. Ribonucleoprotein (RNP) complexes were assembled using 24 μM gRNA and 8 μM recombinant Cas9 protein from *Streptococcus pyogenes* (1:3 ratio Cas9:gRNA). The gRNAs gEREG KO1 and gEREG KO2 were assembled as separate RNPs, and then used as a 1:1 mix.

Targeting of the gRNAs was validated using two different systems. Firstly, in vitro cleavage of target DNA was carried out. In brief, to create the template, PCR was performed with 2x NEBNext High-Fidelity PCR Master Mix (New England Biolabs, M0541S) and 10 μM primers (Table EV2) on genomic DNA isolated from CRTDi004-A iPSC under the following cycling conditions: initial denaturation at 98 °C for 30 s, 40 cycles [98 °C 10 s, 58/60 °C 30 s, 72 °C 1 min] and final elongation at 72 °C for 5 min. PCR products were analyzed via gel electrophoresis on a 1.5% agarose gel, cut from the gel, and extracted using the QIAquick gel extraction kit. Templates were combined with the generated RNP complexes, and incubated for 60 min at 37 °C, the digestion stopped with 2 mg/mL Proteinase K (56 °C, 10 min), and the final cleavage products were analyzed on a 2% agarose gel.

Secondly, gRNAs were validated in CRTDi004-A iPSCs. For this, 160,000 cells per condition were nucleofected with the RNPs and 0.1 μg/μl of a pCAG-GFP plasmid using the 4D-Nucleofector by Lonza (CB150 pulse, P3 Primary Cell 4D-Nucleofector X Kit S;

Lonza, V4XP-3032). Nucleofected cells were plated on Matrigel-coated six-well plates and cultured in mTeSR1 with 1X penicillin/streptomycin. After 3 days in culture, cells were detached using TrypLE Express Enzyme, resuspended in 300 μl of 2% FBS in PBS with 1000 ng/μl DAPI, and processed by FACS on a Melody Cell Sorter (BD Biosciences) using the BD FACSChorus software. Roughly 10,000–60,000 GFP-positive cells per condition were collected in 2% FBS in PBS, and DNA extraction was performed. PCR templates were generated as described above using Seq-Primers (Table EV2), gel-purified, and sent for Sanger sequencing (Microsynth) using the forward primers (Table EV2). The sequencing results were analyzed using Geneious Prime (pairwise, global alignment with free end gaps, 93% similarity cost matrix).

### CRISPR/Cas9-mediated gene knockout in human cortical organoids

Electroporation of the RNP complexes for CRISPR/Cas9-mediated targeting of *EREG* was carried out 2 days after the first slicing of human cortical organoids at week 6 of organoid culture. Organoids that did not display ventricle structures after slicing were discarded. Organoids were transferred to a 6 cm ultra-low-attachment dish (Eppendorf, 30701011) containing Tyrode's solution (Sigma, T2145). Using a glass microcapillary (Sutter Instrument, BF120-69-10), 0.2 to 0.5 μl of the RNP/plasmid (pCAG-GFP) mix at a final concentration of 24 μM RNP and 0.1 μg/μl plasmid diluted in 0.1% Fast Green Solution (in water) were injected into areas depicting ventricular morphology. Injections were carried out using a microinjector (World Precision Instruments, SYS-PV820) in a continuous setting. Up to five ventricles were injected per organoid. A total of 7-8 organoids were processed per replicate. After injection, organoids were transferred into an electroporation chamber containing Tyrode's solution and electroporated with five pulses applied at 38 V for 50 ms each at intervals of 1 s (Harvard Bioscience, BTX ECM 830). Subsequently, the electroporated organoids were moved back into the culture medium and incubated for 7 days before fixation with 4% PFA.

### Immunohistochemistry

Tissue was fixed in 4% PFA in 120 mM phosphate buffer pH 7.4 for 24 h at 4 °C, transferred to 30% sucrose for 24 h, embedded in OCT compound (Sakura Finetek, 4583) with 15% sucrose, and cut into 12 μm (human tissue), 20 μm (mouse tissue), or 25–30 μm (organoids) cryosections. For analysis of radial processes, fixed tissue was cut into 70 μm vibratome sections.

Immunofluorescence was performed as previously described (Florio et al, 2015). Antigen retrieval for 1 h with 10 mM citrate buffer pH 6.0 at 70 °C in a water bath was followed by a wash with PBS, quenching for 30 min in 0.1 M glycine in PBS and blocking for 30 min in blocking solution (10% horse serum and 0.1% triton in PBS; or glycine 0.2%, 300 mM NaCl and 0.3% triton in PBS for phospho-vimentin staining) at room temperature. Primary antibodies were incubated in a blocking buffer overnight at 4 °C. Subsequently, sections were washed three times in PBS, incubated with secondary antibodies (1:1000) and DAPI (1:1000) in a blocking solution for 1–2 h at room temperature, and washed again three times in PBS before mounting on microscopy slides with Mowiol.

Most images were acquired using either a Zeiss LSM 980 confocal microscope with a 20x objective using 0.83-μm or 0.50-μm thick optical sections or with a 40x objective with 0.34-μm optical

sections. Few images were acquired either with a Keyence BZ-X800E fluorescence microscope with a 10x objective or with a Zeiss ApoTome2 fluorescence microscope with a 20x objective and 1.5-μm thick optical sections. Images are shown as maximum intensity projections of 11–15 optical sections. When images were taken as tile scans, the stitching of tiles was performed using the ZEN software.

### Cell counting

All samples were blinded before the acquisition of the data. Quantification was performed using Fiji, processed using Excel (Microsoft), and plotted using Prism (Graphpad Software). Image quantifications were performed either manually or using the Fiji plugin StarDist (Schmidt et al, 2018). Shortly, images were analyzed as maximum intensity projections. Images were cropped to either 400 μm (PH3) or 100 μm (Sox2, Tbr2, Ki67, and PCNA) wide pictures, followed by a crop of the different cortical layers (VZ, SVZ/IZ, CP). Channels were split and the StarDist 2D plugin was performed using the versatile (fluorescent nuclei) model. If necessary, segmentation was corrected manually. Abventricular PH3 was defined as more than three nuclei away from the ventricular surface using DAPI as a reference. Slice culture samples that did not show an intact apical surface (based on pan-cadherin staining) were excluded from the analysis. Quantification of electroporated samples was performed by segmenting the GFP cells either with StarDist or manually, and then counting double-positive cells using the cell counter tool or fluorescence intensity measures in Fiji. For the analysis of electroporated cells, only ventricles towards the outside of the organoid were analyzed, as ventricles at the organoid core may suffer from reduced oxygen and nutrient availability (Qian et al, 2020). The proliferation assay in mNSC was quantified using the BZ-X800 analyzer software or manually (p0 vs p9 comparison).

### Protein expression analysis by Western blotting

Proteins were isolated from two to three human cortical organoids at 6 weeks or from N2a cells that overexpressed human *EREG* for 24 h. Briefly, N2a cells were plated in six-well plates and grown in N2a medium containing DMEM/F12, 10% FBS, and 1X penicillin/streptomycin. Cells were transfected with a pCAG-empty or pCAG-*EREG* plasmid, together with pCAG-GFP plasmid, using lipofectamine 2000 according to the manufacturer's instructions. Protein concentration was measured using the Pierce detergent-compatible Bradford assay kit (Thermo Scientific, #23246). Subsequently, 40 μg of protein were resolved on a 10% SDS Polyacrylamide gel and transferred to a PVDF transfer membrane (Thermo Scientific, #88518). Membranes were blocked for 1 h at room temperature with 5% skim milk in PBS with 0.05% tween, and then incubated with primary antibodies overnight at 4 °C. Secondary antibodies were incubated for 1 h at room temperature. Antibody signal was detected using the SuperSignal West Pico plus kit (Thermo Fisher Scientific, #34579).

### Gene expression analysis by RT-qPCR

Gene expression analysis was performed as previously described (Albert et al, 2017). For RT-qPCR, RNA was isolated either from 10,000 cells (after FACS) or from one to two whole organoids using the RNeasy Mini kit, cDNA was synthesized using random hexamers and Superscript III Reverse Transcriptase (Invitrogen) and qPCR

performed with LightCycler 480 SYBR Green I Master (Roche) on a LightCycler 480 (Roche). For each sample, three technical replicates were run. Gene expression data was normalized based on the housekeeping gene *Gapdh*. Primers are listed in Table EV2.

### Isolation of neural cell populations for RNA-seq

Isolation of aRG, bIPs, and neurons has been previously described (Florio et al, 2015; Molyneaux et al, 2015; Schütze et al, 2023). After HERO culture of E14.5 cerebral hemispheres from *Tubb3*::GFP mice (Attardo et al, 2008) with or without 50 ng/mL EPIREGULIN for 24 h, the dorsolateral neocortex was dissected, and a single cell suspension was produced from two to three hemispheres per condition using the MACS Neural Tissue Dissociation kit containing papain (Miltenyi Biotec, 130-092-628). Cells were fixed in 1% formaldehyde for exactly 10 min on a rotating wheel (10 rpm) at room temperature, the fixative was quenched by the addition of 0.2 M glycine for 5 min, cells were washed in 1% BSA with centrifugation for 5 min at 500×*g*, permeabilized for 10 min at 4 °C with 0.1% saponin (Sigma Aldrich, 47036) and 0.2% BSA (Sigma Aldrich, A2153) in Tyrode's solution and subsequently spun down for 3 min at 500×*g*. At least 3 million cells were stained with antibodies against Sox2 (Mouse anti-Sox2-PE, 1:40) and Tbr2 (Rabbit anti-TBR2/Eomes, 1:250), washed twice in permeabilization buffer and once in 0.5% BSA. This was followed by staining with an anti-rabbit secondary Alexa405 (1:1000). The cells were resuspended to 1–3 million cells/100 μL in Tyrode's solution, passed through a 35-μm filter and immediately processed by FACS on a five-laser FACS Aria II sorter using the FACSDiva software. For each cell type and condition, 250,000 cells were collected in 1.5 mL DNA LoBind tubes containing lysis buffer. RNA was extracted using the Quick-RNA FFPE MiniPrep kit. Biological replicates ($n = 3$) were collected from three pregnant female mice that were processed and sorted independently on different days.

### RNA-seq library preparation

Transcriptome libraries were prepared using the SmartSeq 2 protocol (Picelli et al, 2013). Isolated total RNA from an equivalent of 25,000 cells was denatured for 3 min at 72 °C in 4 μL hypotonic buffer (0.2% Triton-X 100) in the presence of 2.5 mM dNTP, 100 nM dT-primer and 4 U RNase Inhibitor (Promega, N2611). Reverse transcription was performed at 42 °C for 90 min after filling up to 10 μl with RT buffer mix for a final concentration of 1X superscript II buffer (Invitrogen), 1 M betaine, 5 mM DTT, 6 mM MgCl2, 1 μM TSO-primer (Table EV2), 9 U RNase inhibitor, and 90 U Superscript II. The reverse transcriptase was inactivated at 70 °C for 15 min. For subsequent PCR amplification of the cDNA the optimal PCR cycle number was determined with an aliquot of 1 μL unpurified cDNA in a 10 μL qPCR containing 1X Kapa HiFi Hotstart Readymix (Roche), 1X SybrGreen, and 0.2 μM UP-primer (Table EV2). The residual 9 μL cDNA were subsequently amplified using Kapa HiFi HotStart Readymix (Roche) at a 1X concentration together with 250 nM UP-primer (Table EV2) under the following cycling conditions: initial denaturation at 98 °C for 3 min, 12 cycles [98 °C 20 s, 67 °C 15 s, 72 °C 6 min] and final elongation at 72 °C for 5 min. Amplified cDNA was purified using 1X volume of Sera-Mag SpeedBeads (GE Healthcare) resuspended in a buffer consisting of 10 mM Tris, 20 mM EDTA, 18.5% (w/v) PEG 8000, and 2 M sodium chloride solution. The cDNA quality and concentration were determined using a Fragment Analyzer (Agilent).

For library preparation, 2 µl amplified cDNA was tagmented in 1X Tagmentation Buffer using 0.8 µl bead-linked transposome (Illumina DNA Prep, (M) Tagmentation, Illumina) at 55 °C for 15 min in a total volume of 4 µL. The reaction was stopped by adding 1 µl of 0.1% SDS (37 °C, 15 min). Magnetic beads were bound to a magnet, the supernatant was removed, beads were resuspended in 14 µL indexing PCR Mix containing 1X KAPA Hifi HotStart ReadyMix (Roche) and 700 nM unique dual indexing primers (i5 and i7), and subjected to a PCR (72 °C 3 min, 98 °C 30 s, 12 cycles [98 °C 10 s, 63 °C 20 s, 72 °C 1 min], and 72 °C 5 min). Libraries were purified with 0.9X volume Sera-Mag SpeedBeads, followed by a double size selection with 0.6X and 0.9X volume of beads. Sequencing was performed after quantification using a Fragment Analyzer on an Illumina Novaseq 6000 with an average sequencing depth of 60 million fragments.

### RNA-seq data analysis

Basic quality control of the sequence data was done with FastQC (v0.11.6). Adapters (nextera:CTGTCTCTTATA) and polyA/T tails were trimmed with cutadapt (v2.6) (Martin, 2011), and pairs where one or both reads were shorter than 35 bp were removed. Trimmed reads were aligned to the mouse reference genome GRCm39 using the aligner gsnap (v2020-12-16) (Wu and Nacu, 2010) with Ensembl 104 mouse splice sites as support. Uniquely mapped reads were compared based on their overlap to Ensembl 104 human gene annotations using featureCounts (v2.0.1) (Liao et al, 2014) to create a table of fragments per mouse gene and sample. All raw fragments were normalized based on library size and tested for the R package DESeq2 (v1.30.1). Principal component analysis was computed to explore the correlation between biological replicates and different libraries based on the top 500 genes with the highest variance. To identify genes differentially expressed, counts were fitted to the negative binomial distribution. Wald test of DESeq2 was used to test genes between conditions. $P$ values were corrected for multiple testing with the Independent Hypothesis Weighting package (IHW 1.18.0) (Ignatiadis and Huber, 2021; Ignatiadis et al, 2016). Genes with a maximum of 10% false discovery rate ($p$adj ≤0.1) were considered as significantly differentially expressed. TPM values for quantifying gene expression levels within samples were calculated with kallisto (v0.46.1) (Bray et al, 2016) based on an index of Ensembl 104 mouse transcript sequences, and TPMs were summarized at the gene level.

### Histone methylation analysis by ChIP-qPCR

ChIP was performed as previously described (Albert et al, 2017). For ChIP-qPCR, 100,000 cells were sorted into 1.5 mL tubes, filled up to 500 µL with PBS, and transferred into DNA LoBind tubes. Cells were fixed in 1% formaldehyde while shaking at 700 rpm for 10 min at room temperature. The reaction was stopped by the addition of 80 µL 1.25 M glycine and incubated at 350 rpm for 5 min at RT. To facilitate pelleting of cells, 60 µL fetal bovine serum was added (Adli and Bernstein, 2011), and cells were centrifuged at 1600×$g$ for 5 min at 4 °C. Cells were washed twice with 1 mL PBS containing 10% FBS and then resuspended in 80 µL cold lysis buffer containing 1% SDS, 50 mM Tris-HCl (pH 8.1), 100 mM NaCl, 5 mM EDTA, and protease inhibitors and incubated on ice for 5 min with intermittent brief vortexing. The samples were sonicated on a Covaris S2 in microTUBES with AFA Snap-cap for 3 min with 2% duty cycle, intensity of 3200 cycles per burst using frequency

sweeping as power mode and continuous degassing at 4 °C. ChIP was performed using the LowCell ChIP kit according to manufacturer's instructions. For each ChIP, 1 µL of anti-H3K27me3 (Cell Signaling, 9733) was used. For the qPCR, the LightCycler 480 SYBR Green I Master and the LightCycler 480 from Roche were used. Primers are listed in Table EV2.

### Constructs for epigenome editing

The constructs for epigenome editing were based on the plasmid pSpCas9n(BB)-2A-GFP (Addgene #48140) (Ran et al, 2013) for expression of Cas9 and GFP, linked by a T2A site, from a CAGS promoter. The original vector was modified by converting the Cas9 nickase into a non-cutting dCas9. Additionally, a BsaI-flanked ccdB cassette between the U6 promoter and the guide RNA scaffold was introduced, together with two BbsI sites between the C-terminus of dCas9 and the NLS sequence. The coding sequence of the catalytic domain of human KDM6B (1025–1680 aa) was obtained from the MSCV_JMJD3 plasmid (Addgene #21212) (Sen et al, 2008). BsaI sites in the coding sequence were inactivated by the introduction of silent mutations through PCR amplification of KDM6B fragments and reassembly downstream of dCas9. The resulting plasmid backbone pCAGS_dCas9-KDM6B-L_GFP_ccdB featured sequences for a dCas9-JMJC_6B fusion protein, and GFP expressed from a CAGS promoter as well as a U6 promoter followed by a BsaI-ccdB cassette for efficient gRNA cloning.

Four gRNAs targeting sequences upstream and downstream of the mouse *Ereg* transcription start site were selected using Geneious Prime software (see Table EV1 for gRNA sequences). Two pairs consisting of an upstream and a downstream gRNA expressed from the human U6 promotor and fused to a gRNA scaffold were cloned into the dCas9-JMJC_6B-T2A-EGFP plasmid. Sequences encoding target-specific gRNAs were cloned by Golden Gate cloning using BsaI. The ccdB cassette was excised and replaced by a cassette featuring the spacer and scaffold for the first gRNA, followed by a second U6 promoter and a second spacer. CcdB removal allowed counter-selection. The previously published gRNA targeting *LacZ* was used as a control (Albert et al, 2017; Kalebic et al, 2016).

### Epigenome editing in mNSC

For epigenome editing, mouse NSC were transfected with the dCas9-JMJC_6B-T2A-EGFP plasmid encoding a pair of gRNAs targeting the *Ereg* locus. Two to three million NSCs were resuspended in 100 µL P3 primary solution (Lonza) containing 10 µg of the plasmid DNA and transfected using the program DS-112 of the 4D-nucleofector (Lonza). NSCs were harvested 48 h post-nucleofection, resuspended in Tyrode's solution containing 0.5% BSA and GFP-positive cells were collected by FACS. Cells were then further processed for RT-qPCR and ChIP-qPCR.

### Enhancer activity assay

The constructs for examining enhancer activity of human candidate regulatory regions and their mouse orthologues were generated as previously described (Noack et al, 2022), and a detailed protocol can be found at https://www.protocols.io/view/mpra-plasmid-pool-preparation-bxchpit6/. Briefly, 312-bp single-stranded oligonucleotides containing candidate regulatory elements or scrambled control sequences and overhangs for Gibson assembly were synthesized (Twist Bioscience) (all sequences are listed in Table EV3). All inserts were cloned into the pMPRA1 Addgene plasmid

backbone (Addgene #49349) (Melnikov et al, 2014) via Gibson assembly and transformed into ElectroMAX Stbl4 Competent Cells (Thermo Fisher, 11635018) using 1.8 kV, 25 μF, and 200 Ω. The resulting plasmid was digested with KpnI/EcoRI and ligated with a minimal promoter and the mScarlet-I (kindly provided by the Bonev lab). The purified ligation product was heat-transformed in *E. coli*. The final plasmids were purified using the GeneJET Plasmid-Midiprep-Kit (Thermo Scientific, K0481). All primers used are listed in Table EV2.

The plasmids were co-electroporated with a CAG-GFP plasmid (ratio of pMPRA1:CAG-GFP is 3:1) into human cortical organoids at week 5 as described above. Three days post electroporation, the organoids were fixed, and immunohistochemistry was performed for GFP and for mScarlet using the anti-RFP antibody, as described above.

## Statistical analysis

Sample sizes are reported in each figure legend. Sample sizes were estimated based on previous literature (Albert et al, 2017; Long et al, 2018; Qian et al, 2020). All statistical analysis was performed using Prism (GraphPad Software). The normal distribution of datasets was tested by the Shapiro–Wilk or Kolmogorov–Smirnov test. The tests used included Student's *t*-test and one-way ANOVA with Dunnett post hoc test, as indicated in the figure legend for each quantification. Significant changes are indicated by stars for each graph and described in the figure legends.

## Data availability

RNA-seq data have been deposited with the Gene Expression Omnibus under accession code GSE228007.

## Peer review information

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

## Acknowledgements

We are grateful to the facilities of the CRTD and Dresden Concept partners for the outstanding support provided, notably K. Neumann and her team at the Stem Cell Engineering Facility, H. Hartmann and her team at the Light Microscopy Facility, A. Gompf and her team of the Flow Cytometry Facility, the teams for animal husbandry and histology, the Dresden Concept Genome Center lab team for technical support and the MPI-CBG Genome Engineering Facility. We thank all members of the Albert laboratory for their help and discussions. We thank R. Haase for advice on image analysis, F. Noack for advice on the enhancer activity analysis, and the Bonev lab for kindly providing the construct for the enhancer activity assay. We acknowledge the Welcome Trust Sanger Institute (HipSci) for providing the HPSI0114i-kolf_2 iPSC line. MA acknowledges funding from the Center for Regenerative Therapies TU Dresden, the DFG (Emmy Noether, AL 2231/1-1), the Schram Foundation, and ERA-NET Neuron (MEPIcephaly; Federal Ministry of Education and Research (BMBF), 01EW2208). The content of this publication is the responsibility of the authors. Moreover, this project was funded by the Federal Ministry of Education and Research (BMBF) and the Free State of Saxony as part of the Excellence Strategy of the federal and state governments, in the framework of transCampus. KL is supported by MRC funding (MR/S025065/1).

## Author contributions

**Paula Cubillos**: Conceptualization; Formal analysis; Investigation; Visualization; Writing—review and editing. **Nora Ditzer**: Resources; Formal analysis; Investigation; Visualization; Writing—review and editing. **Annika Kolodziejczyk**: Resources; Investigation; Writing—review and editing. **Gustav Schwenk**: Investigation; Writing—review and editing. **Janine Hoffmann**: Resources; Investigation; Writing—review and editing. **Theresa M Schütze**: Investigation; Writing—review and editing. **Razvan P Derihaci**: Resources. **Cahit Birdir**: Resources. **Johannes EM Köllner**: Resources. **Andreas Petzold**: Formal analysis. **Mihail Sarov**: Resources. **Ulrich Martin**: Resources. **Katherine R Long**: Resources; Funding acquisition; Writing—review and editing. **Pauline Wimberger**: Resources. **Mareike Albert**: Conceptualization; Supervision; Funding acquisition; Investigation; Writing—original draft; Writing—review and editing.

## Funding

## Disclosure and competing interests statement

The authors declare no competing interests.

# Expanded View Figures

**Figure EV1.  Expression of *EREG* in neural progenitor cells of different species.**

(A, B) *EREG* mRNA levels in human fetal Ncx tissue and cerebral organoids analyzed by RNA-seq (data from Camp et al, (2015); Johnson et al, (2015)). (C) In situ hybridization data for *Sox2* and *Ereg* of E13.5 and E15.5 mouse neocortex, obtained from the Allen Brain Atlas (Allen Institute for Brain Science, 2004). (D) Immunofluorescence for SOX2, TBR2, and TUJ1 of mNcx and hNcx tissue. (E) H3K4me3, H3K27me3, and H3K27ac ChIP-seq signal around the *EREG* transcription start site (±1 kb) in mouse proliferative aRG, forebrain, and cortex (top) and of H3K27ac in the human cortex (bottom) (data from Albert and Huttner (2018); Gorkin et al, (2020); Reilly et al, (2015)). (F) Immunofluorescence for SOX2, TBR2, and TUJ1 of gorilla cerebral and human cortical organoids. (G) *EREG* mRNA levels in macaque and human NPCs analyzed by RNA-seq (data from Kliesmete et al, (2023)). (H) *EREG* mRNA levels in the ferret Ncx analyzed by microarray (data from de Juan Romero et al, (2015)). Data information: Scale bars, 100 μm. Bar graphs represent mean values. Error bars represent SD; G, of three samples; H, of six micro-dissected tissue samples.

▶

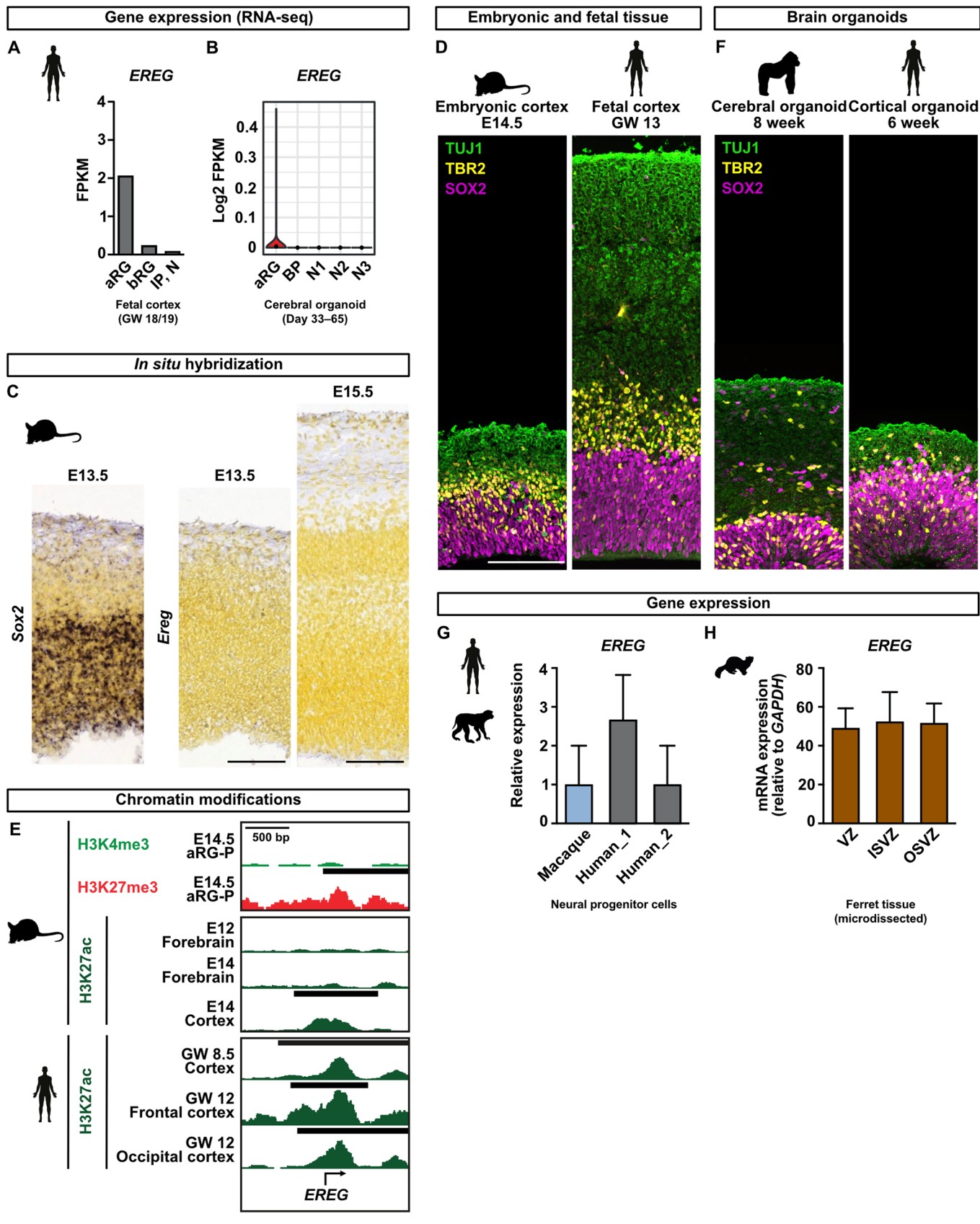

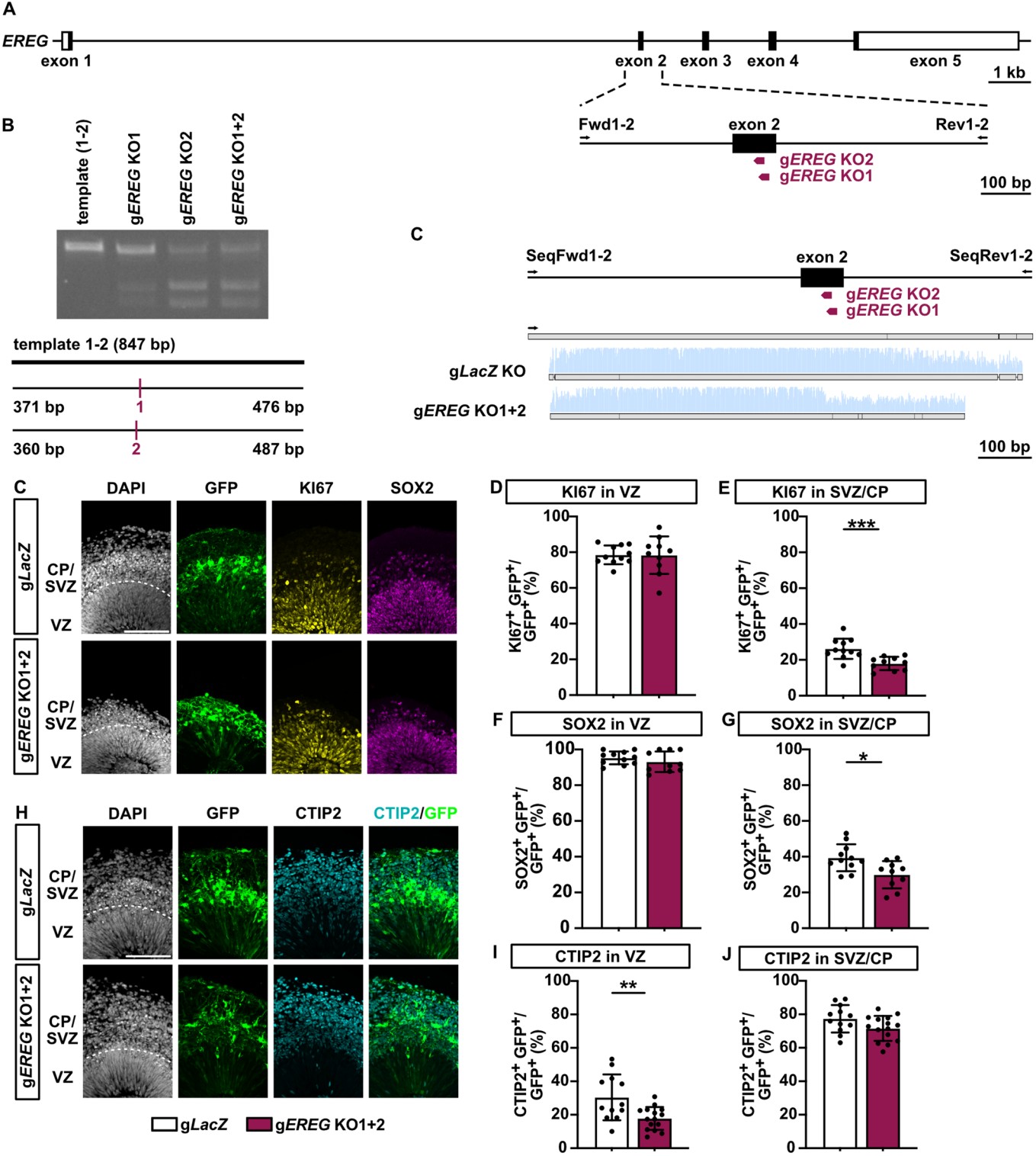

◀ **Figure EV2.  Validation of *EREG* gRNA function.**

(**A**) Schematic illustration of the human *EREG* gene locus. The location of the guide RNAs for CRISPR/Cas9-mediated ablation of EPIREGULIN expression (g*EREG* KO1 + 2) is shown, as well as the location of primer binding sites (Fwd, forward; Rev, reverse) for the generation of DNA templates for in vitro gRNA efficiency testing. (**B**) Guide RNA efficiencies were tested in vitro. The effects of the g*EREG* KO1 + 2 RNAs to direct Cas9-mediated cutting of PCR templates was analyzed by agarose gel electrophoresis. Schemes of the sizes of PCR templates, guide RNA binding sites, and expected sizes of cut fragments are indicated below. (**C**) CRISPR/Cas9-mediated targeting of *EREG* was confirmed in the CRTDi004-A iPSC line by electroporation of Cas9/gRNA ribonucleoprotein complexes together with a GFP plasmid, followed by FACS of GFP-positive cells, PCR amplification of the target region and Sanger sequencing. The sequencing results are shown for g*EREG* KO1 + 2. (**D**) DAPI staining and immunofluorescence for GFP, KI67, and SOX2 of an electroporated human cortical organoid derived from the HPSI0114i-kolf_2 iPSC line. (**E–H**) Quantifications of KI67 and SOX2 in the VZ and SVZ/CP. (**I**) DAPI staining and immunofluorescence for GFP and CTIP2 of an electroporated human cortical organoid from the CRTDi004-A iPSC line. (**J, K**) Quantifications of CTIP2 in the VZ and SVZ/CP. Data information: Scale bar, 100 μm. Bar graphs represent mean values. Error bars represent SD of 12–15 organoids from three batches. **$p < 0.01$; Student's *t*-test.

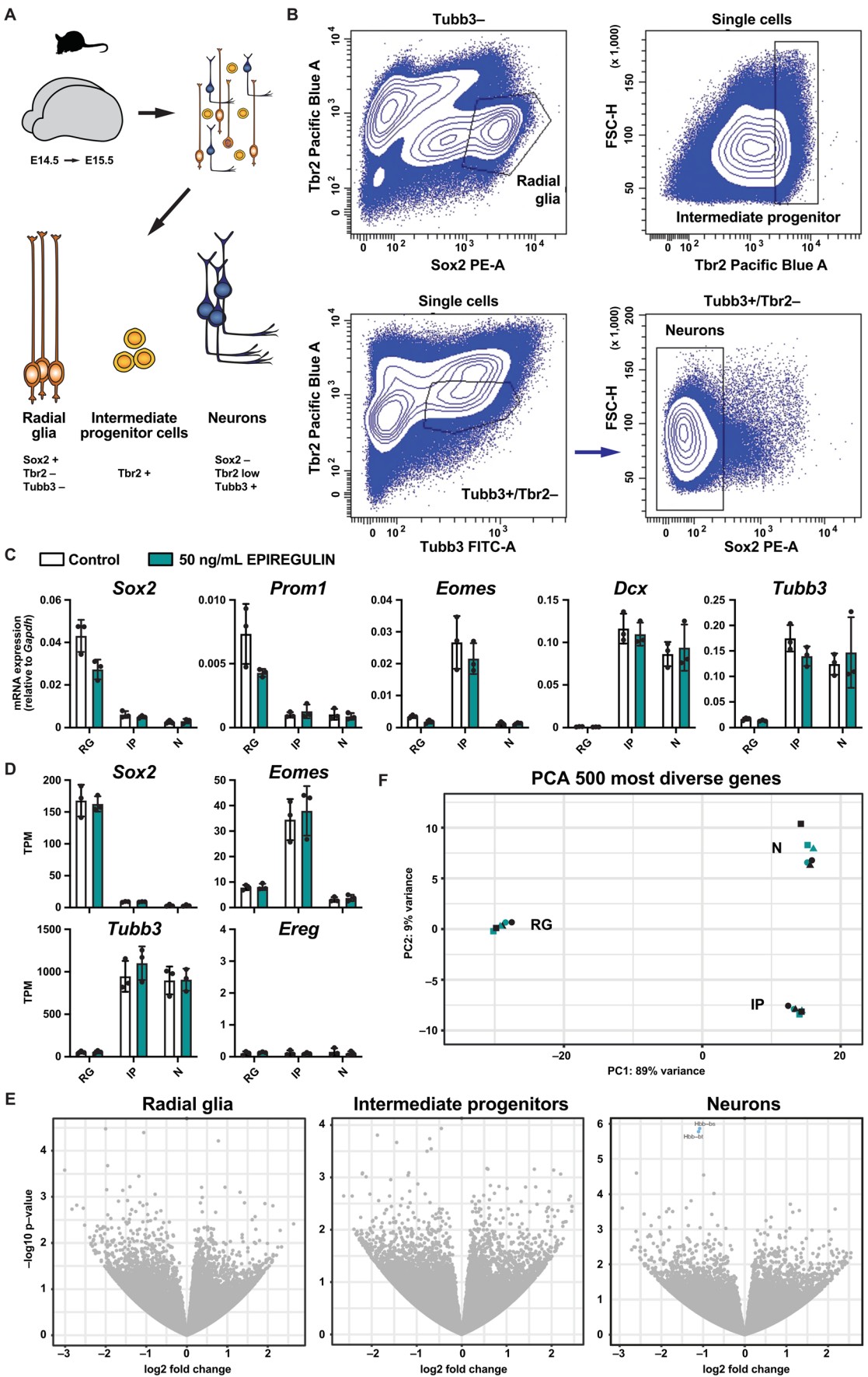

**Figure EV3.  Gene expression analysis upon addition of EPIREGULIN to the mouse neocortex.**

(A) Schematic illustration of the experimental workflow. Mouse brain hemispheres (E14.5) from the *Tubb3*::GPF line (Attardo et al, 2008) were isolated and cultured under rotation in the presence of 50 ng/mL of EPIREGULIN for 24 h, dissociated, stained for Sox2 and Tbr2, and cell populations isolated by immuno-FACS based on the indicated marker combinations. (B) Gating strategy of RG (top, left) based on high levels of Sox2 and low levels of Tbr2; IP (top, right) based on high levels of Tbr2, irrespective of other markers; and neurons (bottom) based on enrichment of GFP expressed from the *Tubb3* promoter and low level of Tbr2, followed by exclusion of Sox2-positive cells. (C) Confirmation of cell type identity by RT-qPCR expression analysis of marker genes characteristic of RG (*Sox2*, *Prom1*), IP (*Eomes*), and neurons (*Dcx*, *Tubb3*) for control and hemispheres treated with EPIREGULIN for 24 h relative to *Gapdh*. (D) Expression of *Sox2*, *Eomes*, *Tubb3*, and *Ereg* in RG, IP, and neurons analyzed by RNA-seq. (E) Volcano plots of log10 (*p* value) against log2 fold change representing the differences in gene expression in the indicated cell types analyzed by RNA-seq. Gray, non-significant; blue, downregulated. (F) Principal component analysis (PCA) based on the 500 most divergent genes. The percentage of variance covered by the first two components is indicated. Data information: Bar graphs represent mean values. Error bars represent SD of 3 mNcx samples from different litters. E, Wald test of DESeq2 was used and *P* Values were corrected for multiple testing with the Independent Hypothesis Weighting package (IHW 1.18.0).

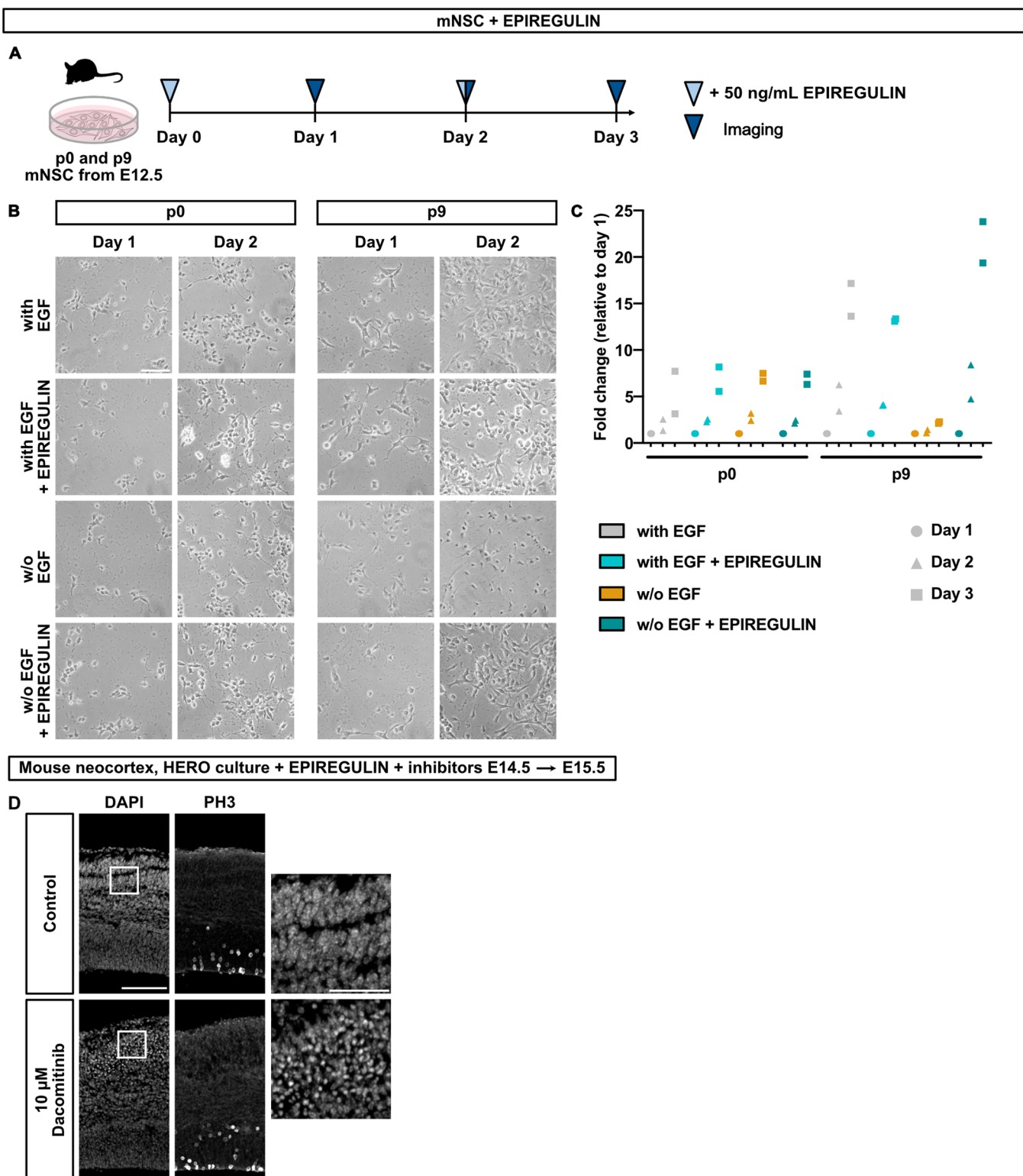

◀   **Figure EV4.   NSC proliferation upon exposure to different growth factors.**

(**A**) Schematic illustration of the experimental workflow. Early (p0) and late (p9) passage mouse NSC cultures were treated with different growth factors for 3 days and their proliferation was assessed. (**B**) Images of mNSC cultures following treatment with EPIREGULIN in culture medium containing FGF and with or without EGF. The control mNSCs were cultured in a medium with EGF and FGF. (**C**) Quantification of cells on days 1, 2, and 3 following EPIREGULIN treatment, either with or without EGF, shown as fold change relative to 1 h. (**D**) Staining for DAPI and immunofluorescence for PH3 of mNcx slices treated with 10 µM of the receptor inhibitor Dacomitinib for 24 h. Note the reduced tissue integrity and the apoptotic nuclei in the inset (right). Data information: Scale bars, 100 µm. **C**, Data points are from two different mNSC lines.

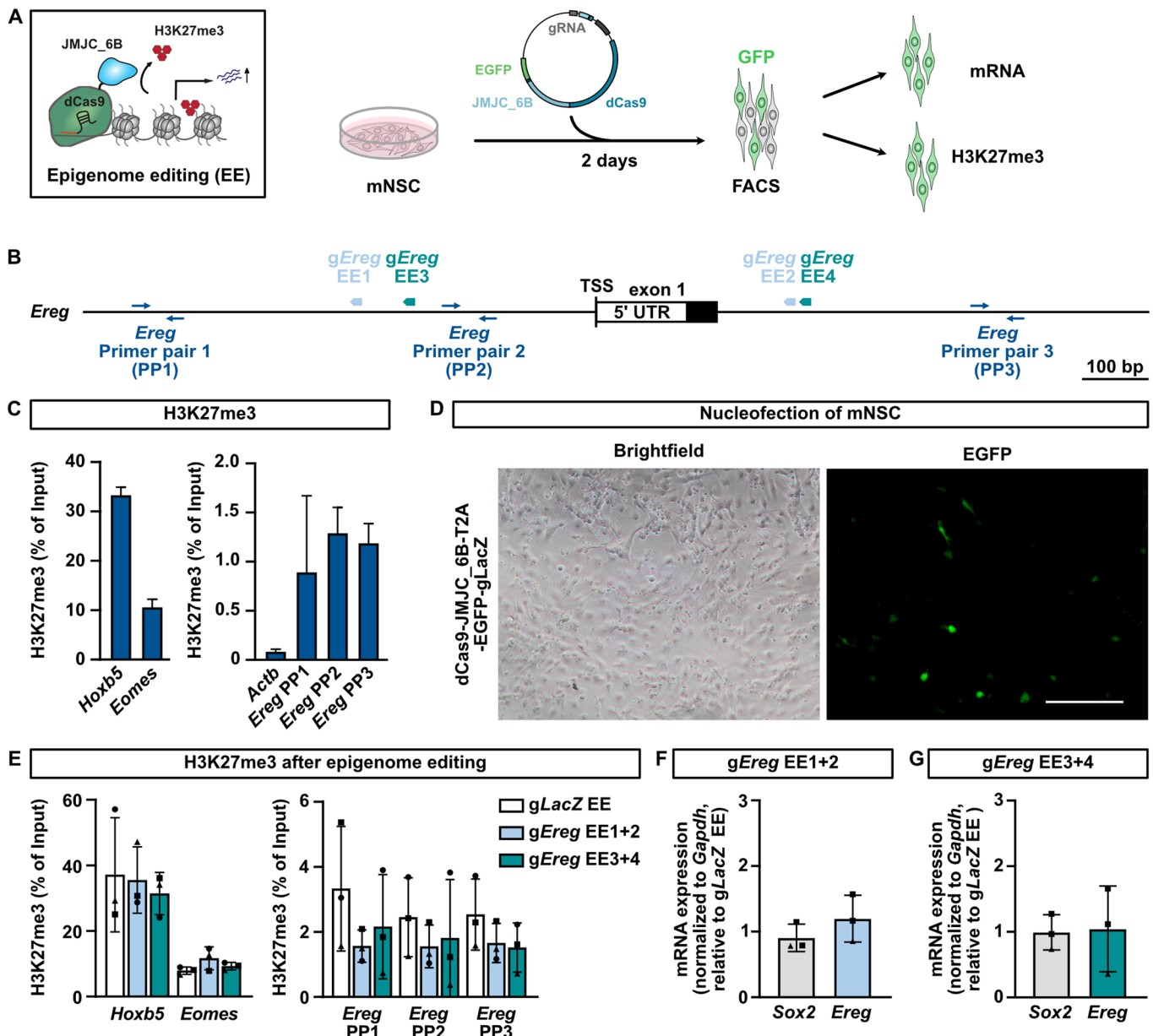

**Figure EV5. Editing of histone methylation at the *Ereg* locus in mNSCs.**

(A) Epigenome editing (EE) employing the catalytic domain of KDM6B (JMJC_6B) fused to nuclease deficient Cas9 (dCas9) in mNSCs. Histone methylation and gene expression were analyzed 2 days post-nucleofection following FACS isolation of GFP-positive cells. (B) The location of the gRNAs and primer binding sites (PP, primer pair) for ChIP-qPCR is shown for the *Ereg* locus. Guide RNAs g*Ereg* EE1 + 2 and g*Ereg* EE3 + 4 were co-expressed from one plasmid, respectively. (C) Level of H3K27me3 at *Hoxb*, *Eomes, Actb*, and *Ereg* (PP1 to PP3) as determined by ChIP-qPCR in mNSCs. (D) Bright-field and GFP fluorescence images of mNSCs 2 days post nucleofection with a dCas9-JMJC_6B-T2A-EGFP-gLacZ plasmid. (E) ChIP-qPCR analysis of H3K27me3 around the TSS of *Ereg* and two unrelated genes (*Hoxb5, Eomes*) after epigenome editing at the *Ereg* locus. (F, G) Expression of *Sox2* and *Ereg* as determined by RT-qPCR upon epigenome editing using g*Ereg* EE1 + 2 and g*Ereg* EE3 + 4. Expression normalized to *Gapdh* and relative to g*LacZ* EE. Data information: Scale bar, 100 μm. Bar graphs represent mean values. Error bars represent the SD of three replicates (from two to three independent experiments). One-way ANOVA with Dunnett post hoc test; no statistically significant changes were detected.

