## [Peer Review File · The EMBO Journal]

The growth factor EPIREGULIN promotes basal progenitor cell proliferation in the developing neocortex

Paula Cubillos, Nora Ditzer, Annika Kolodziejczyk, Gustav Schwenk, Janine Hoffmann, Theresa Schütze, Razvan Derihaci, Cahit Birdir, Johannes Köllner, Andreas Petzold, Mihail Sarov, Ulrich Martin, Katherine R Long, Pauline Wimberger, and Mareike Albert

Corresponding author(s): Mareike Albert (mareike.albert@tu-dresden.de)

Review Timeline:

Submission Date:	13th Sep 23
Editorial Decision:	12th Oct 23
Revision Received:	31st Jan 24
Editorial Decision:	16th Feb 24
Revision Received:	20th Feb 24
Accepted:	21st Feb 24

Editor: Ioannis Papaioannou

Transaction Report:

Dear Mareike,

Thank you for submitting your manuscript (EMBOJ-2023-115599) for consideration by the EMBO Journal. It has now been seen by three experts in the field, and we have received the full set of their comments, which are included below.

As you will see, all referees are positive about the study. They find the results very interesting and significant, the advance for the field considerable, the study mostly well-performed and the results clearly presented in the manuscript. However, they also identify a number of limitations, they point out several issues that require clarification, and they provide suggestions for the improvement of the study and the manuscript.

Given the referees' positive comments and recommendations, I would like to invite you to submit a revised version of your manuscript, addressing the comments of all three reviewers. I should add that it is EMBO Journal policy to allow only a single round of major revision, and acceptance of your manuscript will therefore depend on the completeness of your responses in this revised version. If you have any questions or comments, we can also discuss the revisions in a video chat, if you like.

We generally allow three months as standard revision time (11th January 2024). As a matter of policy, competing manuscripts published during this period will not negatively impact our assessment of the conceptual advance presented by your study. However, we request that you contact us as soon as possible upon publication of any related work, to discuss how to proceed. Should you foresee a problem in meeting this three-month deadline, please let us know in advance and we may be able to grant an extension.

Thank you for the opportunity to consider your work for publication in the EMBO Journal. I look forward to your revision.

Best regards,

Ioannis

Instructions for preparing your revised manuscript

1. When you are ready to submit the revision, please upload:

- A Word file of the manuscript text (including legends of main Figures, EV Figures and Tables). Please make sure that changes are highlighted (or "tracked") to be clearly visible.

- Individual production-quality figure files (one file per figure). When assembling your figures, please refer to our figure preparation guidelines in order to ensure proper formatting and readability in print as well as on screen:

If the data shown in a figure are obtained from n {less than or equal to} 2, please use scatter plots showing the individual data points.

- i. the name of the statistical test used to generate error bars and P values
- ii. the number (n) of independent experiments (please specify technical or biological replicates) underlying each data point (discussion of statistical methodology can be reported in the Materials and Methods section, but figure legends should contain a basic description of n , P , and the test applied)
- iii. the nature of the bars and error bars (s.d., s.e.m.).

- A point-by-point response to the referees' comments, with a detailed description of the changes made (as a word file). All referees' concerns must be fully addressed and their suggestions taken on board. When preparing your letter of response to the referees' comments, please bear in mind that this will form part of the Review Process File and will therefore be available online to the community. Please note that you have the possibility to opt out of the transparent process at any stage prior to publication by letting the editorial office know (contact@embojournal.org); if you do opt out, the Review Process File link will point to the following statement: "No Review Process File is available with this article, as the authors have chosen not to make the review

process public in this case.". For more details on our Transparent Editorial Process, please visit our website: <https://www.embopress.org/page/journal/14602075/authorguide#transparentprocess>

- Expanded View (EV) files (replacing Supplementary Information) that are collapsible/expandable online. A maximum of 5 EV Figures can be typeset. EV Figures should be cited as "Figure EV1, Figure EV2" etc. in the text, and their respective legends should be included in the manuscript file after the legends of regular figures. See detailed instructions regarding Expanded View files here:

- For the figures that you do NOT wish to display as Expanded View figures, they should be bundled together with their legends in a single PDF file called "Appendix", which should start with a short Table of Contents (including page numbers). Appendix figures should be referred to in the main text as: "Appendix Figure S1, Appendix Figure S2" etc. Please see detailed instructions here: <https://www.embopress.org/page/journal/14602075/authorguide#expandedview>

- A complete author checklist, which you can download from our author guidelines (<https://www.embopress.org/page/journal/14602075/authorguide>). Please note that the checklist will also be part of the Review Process File.

2. Please note that no statistics should be calculated if $n=2$.

3. Before submitting your revision, please make sure that all primary datasets (and computer code, where appropriate) produced in this study are deposited in appropriate public databases (see <https://www.embopress.org/page/journal/14602075/authorguide#dataavailability>).

*** Note: all links should resolve to a page where the data can be accessed. ***

*** Note: the Data Availability Section is restricted to new primary data that are part of this study. ***

4. Please check that the title and the abstract of the manuscript are brief, yet explicit, even to non-specialists. The length of the title should not exceed 100 characters (including spaces), and the abstract should be a single paragraph not exceeding 175 words.

5. Please also note our reference format: <https://www.embopress.org/page/journal/14602075/authorguide#referencesformat>.

7. Please remember: digital image enhancement is acceptable practice, as long as it accurately represents the original data and conforms to community standards. If a figure has been subjected to significant electronic manipulation, this must be noted in the figure legend or in the "Materials and Methods" section. The editors reserve the right to request original versions of figures and the original images that were used to assemble the figure.

8. Our journal encourages inclusion of data citations in the reference list to directly cite datasets that were obtained from public databases. Data citations in the article text are distinct from normal bibliographical citations and should directly link to the database records from which the data can be accessed. In the main text, data citations are formatted as follows: "Data ref: Smith et al, 2001" or "Data ref: NCBI Sequence Read Archive PRJNA342805, 2017". In the Reference list, data citations must be labeled with "[DATASET]". A data reference must provide the database name, accession number/identifiers, and a resolvable link to the landing page from which the data can be accessed at the end of the reference. Further instructions are available at: <https://www.embopress.org/page/journal/14602075/authorguide#referencesformat>.

9. We request authors to consider both actual and perceived competing interests. Please review our policy (<https://www.embopress.org/page/journal/14602075/authorguide#conflictsofinterest>) and update your competing interests statement if necessary. Please name this section 'Disclosure and competing interests statement' and place it after the Acknowledgements section.

10. Please note that all corresponding authors are required to provide an ORCID ID upon submission of a revised manuscript (<https://orcid.org/>). Please find instructions on how to link your ORCID ID to your account in our manuscript tracking system in our Author guidelines (<https://www.embopress.org/page/journal/14602075/authorguide#authorshipguidelines>).

11. We use CRediT to specify the contributions of each author in the journal submission system. CRediT replaces the author

contribution section, which should be removed from the manuscript. Please use the free text box to provide more detailed descriptions. See also guide to authors:

<https://www.embopress.org/page/journal/14602075/authorguide#authorshipguidelines>.

13. We would also welcome the submission of cover suggestions or motifs to be used by our Graphics Illustrator in designing a cover.

14. Please use the link below to submit your revision:

Referee #1:

This is a very nice study by Mareike Albert and colleagues studying genetic mechanisms responsible for the developmental expansion of the human neocortex as compared to mouse. They focus on Epiregulin, which after showing its high expression in human and gorilla but absent in mouse embryonic cortex, they use a variety of in vitro assays to show that it promotes proliferation of basal cortical progenitor cells in mouse and gorilla, but not apical progenitors. Interestingly, it also does not enhance proliferation of human progenitors in organoids, argued to be due to saturation of the signaling pathway in human cells. Then they show that Epiregulin competes for signaling with EGF, as both bind the same cell surface receptors. Finally, the authors propose that expression of Epiregulin in human but not mouse cells is due to a specific set of cis-regulatory elements, with a sequence highly conserved in primates but not in mouse. Intriguingly, they show that these enhancers, although not well conserved in both species, have similar gene expression-enhancing activity in both species. The study is very interesting, the rationale very clear and the experiments well conducted and appropriate to address the questions posed, and the conclusions are overall supported by the results. I think this is a good fit for EMBO Journal, and I only have a few comments for further improvement of the current findings.

In Results, the authors add Epiregulin to the developing mouse NCx in vitro to test its effect on the different germinal layers and progenitor cells. How were the germinal layers identified in these samples? How were Sox2+ cells assigned to SVZ and not VZ? This needs to be defined.

The authors find twice as many basal (but not apical) PH3 upon addition of Epiregulin to mouse NCx cultures, but in SVZ they find the same Tbr2 while more Sox2. Is the increased proliferation selective in basal Sox2?

Next, they go on to investigate if these are bRGs, but their definition is flawed. A study some years ago performed a detailed analysis of the cellular morphology of pVim+ cells in ferret, and demonstrated that presence/absence of pVim process does not correlate with bRG/bIP, contrary to what most labs commonly assume. Taking this into consideration, and given the results in Figs 1 and 2 here, the authors' conclusion that IPs increase is only based on increased abundance of Tbr2+ cells in VZ. However, they must demonstrate that these are dividing cells and hence that Tbr2+ mitoses increase, as well as Sox2+ mitoses (not currently shown).

The authors conclude that Epiregulin can induce the amplification of different BP types in the mNcx. These conclusions are based on the combination of results from slice and whole hemisphere culture, but the authors did not show changes in Sox2 and Tbr2 cell abundance in HERO cultures, leaving this as an assumption. This must be shown, or the entire set of analyses must be performed in a single type of assay.

When studying human cortical organoids, the authors conclude: "This indicates that EPIREGULIN is required for BP, but not AP, proliferation in human cortical organoids." - This is not necessarily like this. Abundance of Ki67+GFP+ cells in SVZ results from their self-renewal/amplification, but also from their genesis from VZ progenitors (vs postmitotic neurons). To discern between these two possibilities and reach some conclusion, the authors would need to perform proper cell lineage analyses. Are BPs (specifically) re-entering the cell cycle more with EREG than in control conditions?

Figure 5 should be combined with Figure 4 in a single main figure. In these experiments there is a very obvious accumulation of Sox2+ and Ki67+ cells in the apical surface, suggesting a possible cell cycle arrest in M phase. If this is reproducible. The authors should report on the abundance of these markers in VZ, currently absent.

The authors indicate that "RNA-seq analysis of sorted NPC populations did not point to major gene expression changes as mechanism underlying the EPIREGULIN-induced increase in cell proliferation" - It is very surprising that genes fundamental in proliferation, such as cell cycle genes, did not change in this paradigm of strong changes in progenitor proliferation. Even if it was only a change in progenitor cell fate, as suggested above, then even more transcriptomic changes would be expected.

In the last set of experiments (Figure 7), mScarlet signal should be shown in a different color better contrasting with the black background. Maybe white.

Regarding the role of human CREs in Epiregulin expression levels, one would conclude the opposite than the authors, but that the poor conservation of the CRE sequence demonstrates that these are not key to establish differences in expression levels between human and mouse. The proposed alternative, that differences are due to epigenetic marks, is not demonstrated in this study and thus just as speculative.

In Discussion, when indicating that "non-coding regions of the genome contribute to evolutionarily relevant inter-species differences in gene expression", the authors should also cite Espinos et al. 2022, *Dev Neurobiol*, a comprehensive recent review on this fundamental question in brain evolution.

The authors should discuss whether the CREs absent or very poorly conserved in mouse were a secondary loss during evolution, as shown to be the case for other genes (i.e. Chinnappa et al., 2022, *Sci Adv*), or rather a primate-specific innovation.

Referee #2:

Here the authors describe a novel regulator of specifically basal progenitors and basal radial glial cells. They identify epiregulin expression absent in the murine developing cortex, but expressed in primate and human cortex. The authors then demonstrate that addition to Epiregulin to murine cortex cultures increases specifically basal progenitors/RGCs, but not apical RGC proliferation and number. Conversely, deletion of EREG in cells of human iPSCs-derived cortical organoids reduced proliferation of SVZ, but not VZ cells and addition of Epiregulin showed effects in gorilla iPSC-derived organoids, but not human organoids, suggesting that the system is saturated in human cortex organoid, but not gorilla ones. The authors then proceed to sequence the mouse cortex after addition of Epiregulin and find no genes significantly regulated. They next searched for the receptor and identify EGFR as the main receptor based on selective pharmacological inhibitors reverting the effects of epiregulin addition in murine cultures. Lastly, they identify 2 regulatory sites in the human genome that may contribute to regulate Epiregulin expression in human, but not murine cortex development. These data are very interesting as expansion of basal progenitors and basal radial glia is highly relevant during ontogeny and phylogeny for expansion of specific brain regions.

Minor suggestions

- 1) The ferret also has an enlarged cortex and transcriptome data have been published e.g. from Victor Borrells lab where the SVZ and VZ had been dissected. Is Epiregulin also expressed in the SVZ of the ferret? Please incorporate in the first part of the results section.
- 2) I am surprised that no genes are changed in cortex cells upon the addition of epiregulin, as one would expect proliferation genes in basal progenitors to increase in expression. As the authors sorted even Tbr2+ cells, one would expect an increase in proliferation gene expression in this population. Please comment on this and possibly refine the sequencing data analysis.
- 3) The experiment on p.11 using growth factor expanded NSCs is very confusing. First because the data throughout the paper suggest that epiregulin affects basal progenitors, but not apical ones. So here they work with a neural stem cell population treated with FGF that also reacts. How do the authors align this with the remainder of their data.
- 4) Have the experiments in Figure 3 been performed with independent lines? I see different symbols in the data points, but they are not explained (or I did not find the explanation neither in the Figure legend nor Figure panels).
- 5) To explain the differential effects of Epiregulin addition to Gorilla versus human cortical organoids could the authors perform Western Blot or ELISA to measure endogenous levels of Epiregulin that may be lower in the Gorilla organoids?
- 6) I may have missed this, but are the CREs they identify in Figure 7 for human cells also present in Gorilla?

Referee #3:

In this manuscript, the authors analyze the role of Epiregulin (EREG), a member of the epidermal growth factor (EGF) family of ligands, in the regulation of basal progenitor proliferation in the developing human brain.

Using published RNAseq datasets from mouse and human fetal brains, combined with RTqPCR analyses of human/gorilla organoids, they first show that EREG is expressed in primate RGCs (at higher levels than in neurons) but is not expressed in the different cell types of the mouse developing cortex.

To explore the role of EREG in basal cortical progenitors, the authors used a large panel of approaches (Human and gorilla organoids, organotypic slices of mouse and human fetal brains, culture of mouse and human cortices (HERO), mouse NSCs, CrispR/Cas9 mediated KO, RNAseq analyses of FACS-selected cells from mouse fetal brain, and Epigenomic analyses). They show that addition of EREG to the culture medium enhances the proliferation of basal progenitors of the mouse neocortex and in gorilla organoids, but not in human organoids, which they attribute to the human cortex being EREG saturated. Inhibition of EREG expression in human organoids suggests that EREG is required to sustain basal progenitor proliferation. They further suggest that in the mouse, EREG competes with EGF to promote basal RG proliferation, and that its action is

mediated via the EGFR.

Seeking for the mechanisms involved in the differential regulation of EREG expression across species, they then studied putative EREG enhancer regions in the human genome. They show that the Ereg locus is enriched in repressive marks (H3K27me3) and devoid of active marks (H3K4me3 and H3K27ac) in the mouse neocortex, whereas the EREG locus is enriched in active marks (H3K27ac) in the human cortex. Removing the repressive mark H3K27me3 in the mouse neocortex did not result in an upregulation of Ereg gene expression, suggesting that the ereg locus is in a fully repressed state in the mouse. Using ATACseq and published H3K27ac ChIP-seq data sets of the human fetal cortex, they further suggest that cis-regulatory elements may be involved in inter-species differences in EREG expression.

General comments:

The ms is well-written and the results are clearly presented. The results are topical and of great interest and significance in the field of cortical development, with respect to human specificities. However, in numerous instances, the ms suffers from "shortcuts" where the authors make unsubstantiated claims (see below in specific comments).

Some conclusions are based on limited data sets -or on unknown numbers. Increasing the number of samples/observations will improve the significance of the results.

Altogether, these interesting results need to be strengthened.

As examples (non exhaustive):

Fig1D: RT-qPCR: 3 organoids were analysed. The authors should increase the number of biological replicates.

The number of slices & mice/organoids analysed should be provided.

Fig1G-P Fig 2 (SOX2; TBR2), Fig3 D,E,G,H...

The quantification of positive cells is not appropriate:

The authors quantified the numbers of PH3+, Ki67+ PCNA+ cells/ 400 or 200 m-wide field (Fig1; Fig6I). Idem for SOX2, TBR2 and pVimentin+ cells (Fig2). This rough analysis may vary depending on cell density in the ROI under consideration. The authors should compute the percentage of positive cells/ total cells (using Dapi-as in Fig4A), and indicate the number of slices and samples analysed.

Specific comments :

The authors conclude that EREG specifically acts on basal progenitors of the SVZ -but not apical progenitors). The results are presently unclear and this conclusion needs to be correctly substantiated.

- "addition of recombinant EPIREGULIN to mNcx cultures resulted in an increase in NPC proliferation, preferentially in the SVZ" (Page 8; Fig 1G-P; Fig2B): this statement is not fully substantiated as it is based on quantification of the number of cells/field (max 6 fields analysed per condition). The authors should properly document the increase in proliferation using cell-cycle and proliferation quantification with respect to the total number of cells and not just by computing numbers over ROI (200 or 400 microns). This will allow them to go beyond suggesting an effect as written in the last sentence of this section- which contrasts with the title of the subsection.

"Both major types of BPs, bIPs characterized by Tbr2 expression and the lack of a process as well as bRG marked by Sox2 and the presence of a process, were increased, suggesting that EPIREGULIN can induce the amplification of different BP types in the mNcx".

Again this conclusion (page 8) is not demonstrated by the analysis as the authors did not provide evidence that Tbr2+ bIPs lack processes nor that SOX2+ bRGs bear a process. The authors should quantify the percentage of cells with and without a process co-stained with p-Vimentin/SOX2 and TBR2 to substantiate their claim.

"EPIREGULIN ablation in human cortical organoids reduces basal progenitor proliferation"

"Both SOX2 and TBR2 were reduced upon EREG targeting (Fig 3G, H), suggesting that EPIREGULIN contributes to the proliferation of both BP types, bIPs and bRG, in 6-week cortical organoids, in which the SVZ-like area is in the process of expansion".

The results are shown on Fig3. In 3C, the limit between VZ and SVZ/CP is not convincing as the bar on the left appears to be located within the VZ. This is an important issue regarding which type of progenitor is affected. The authors should therefore check for neuronal markers for ascertaining the boundaries between the different cell compartments. Did the authors observe increased percentage of neurons in the KO condition following 7 days of in survival? The authors should also provide the % of SOX2+ in both VZ-like and SVZ-like as well as that of TBR2+ .

Fig3F: It would be useful to have merged images GFP/SOX2 and GFP/TBR2

"Unlike the human, gorilla BP proliferation, in particular the percentage of SOX2-positive cells likely representing bRG, could be further stimulated by the addition of EPIREGULIN" (Page 10):

The assay on human fetal tissue tests the effect of Epieregulin over a 24 hours period and shows a non-significant trend in increasing the % of Ki67+, Sox2+ and TBR2 + in the SVZ. This short duration may well not allow to detect an effect given the prolonged cell-cycle duration of human cortical progenitors. Hence, the human cortical organoid assay (10 days of culture). The percentage of SOX2+ cells should be provided in both the VZ and SVZ (Fig5C-E).

The limits of VZ, OSVZ must be shown on Fig 4B.

In the mouse, inhibition of EGFR results in a loss of the EREG-induced increase in abventricular mitosis.

"We then performed HERO cultures with only the solvent (ethanol; control), solvent and 50 ng/mL EPIREGULIN, or EPIREGULIN and different concentrations of the two inhibitors" (Page12):

Only one concentration of inhibitor has been tested (1uM) (Fig6I). How has this dosage determined?

The effects of the two inhibitors AG1478 and Dacomitinib on human/gorilla organoids or human fetal cortex should be examined, as they appear more relevant than mouse cortex. Do these inhibitors alter the proliferation rate of VZ apical progenitors and OSVZ bRGCs?

"These results suggest that EGFR is the key mediator of the EPIREGULIN-mediated increase in BP amplification, whereas Erbb4 appears to not play a major role". The authors should test an inhibitor specific to Erbb4 to better substantiate their claim.

"Epigenome editing in the mNSCs using dCas9-JMJC_6B resulted in a reduction of H3K27me3 at the Ereg locus, but not at the unrelated Hoxb5 and Eomes genes" (Page 13)

In contrast to author's claim, there does not seem to be a significant decrease in H3K27me3 (FigS6E). This may explain why there is no increase in ereg expression in FigS6F&G. The authors need to clarify this.

"Two additional genes are located within this genomic region, AREG and EPGN.... However, based on chromatin (Fig 7A) and RNA- seq (Fig EV5) data, they are likely not expressed in the fetal hNcx".

Note that as shown on Fig S5B, EPGN is expressed in RGs and Neurons at GW12-13. Given that this gene encodes a growth factor that binds to EGFR, it may also play a role at this stage. The authors need to comment upon this.

Minor comments:

Fig 6H : The limit between VZ and OSVZ must be shown.

"The Ereg locus is enriched in repressive marks (H3K27me3) in the mouse neocortex, whereas it is enriched in active marks (H3K27ac) in the human neocortex" (Page 6). What about repressive marks in humans (H3K27me3) (not shown in FigS1-correct the legend)?

"Both EGF and FGF were reported to elicit NSC proliferation in vitro (Tropepe et al, 1999), and upon acute withdrawal of EGF, mNSCs did not proliferate further" (Page 11).

This is somehow surprising that mNSCs stop proliferating 2-days after EGF removal (the medium still contains FGF2) (Fig6G) since it has been shown that mNSCs can proliferate in the absence of EGF (Tropepe et al).

Fig7A: please correct the drawing of ATAC and H3K27ac.

"the presence of different epidermal growth factors modulates the cellular outcome". Do they mean "different growth factors"?

"To test whether these putative enhancer regions display enhancer activity, we cloned two of the human CRE regions (216 bp)": the authors mention CRE6 and CRE9: why CRE9 was chosen?

"Comparing hCRE6 and mCRE6, higher levels of mScarlet were observed for hCRE6 compared to the orthologous mouse sequence (Fig 7E). Human CRE9 was also able to drive mScarlet expression at similar levels as hCRE6, yet, in this case the orthologous mouse sequence showed similar or even higher activity (Fig 7F)": the level of mScarlet or the percentage of Scarlet+ cells should be quantified (Fig7E,F).

Referee #1:

This is a very nice study by Mareike Albert and colleagues studying genetic mechanisms responsible for the developmental expansion of the human neocortex as compared to mouse. They focus on Epiregulin, which after showing its high expression in human and gorilla but absent in mouse embryonic cortex, they use a variety of in vitro assays to show that it promotes proliferation of basal cortical progenitor cells in mouse and gorilla, but not apical progenitors. Interestingly, it also does not enhance proliferation of human progenitors in organoids, argued to be due to saturation of the signaling pathway in human cells. Then they show that Epiregulin competes for signaling with EGF, as both bind the same cell surface receptors. Finally, the authors propose that expression of Epiregulin in human but not mouse cells is due to a specific set of cis-regulatory elements, with a sequence highly conserved in primates but not in mouse. Intriguingly, they show that these enhancers, although not well conserved in both species, have similar gene expression-enhancing activity in both species. The study is very interesting, the rationale very clear and the experiments well conducted and appropriate to address the questions posed, and the conclusions are overall supported by the results. I think this is a good fit for EMBO Journal, and I only have a few comments for further improvement of the current findings.

We are pleased that this reviewer finds our study interesting and well conducted, and would like to thank the reviewer for their overall very positive feedback on our study.

In Results, the authors add Epiregulin to the developing mouse NCx in vitro to test its effect on the different germinal layers and progenitor cells. How were the germinal layers identified in these samples? How were Sox2⁺ cells assigned to SVZ and not VZ? This needs to be defined.

The VZ was identified based on the alignment and pseudostratification of nuclei in the DAPI channel. This is now indicated in the manuscript. Additional markers, such as Sox2 and Tbr2, if present in the staining, were used to validate the zone boundaries. We have now outlined the boundaries of the germinal zones with dotted lines throughout the manuscript for easier recognition.

The authors find twice as many basal (but not apical) PH3 upon addition of Epiregulin to mouse NCx cultures, but in SVZ they find the same Tbr2 while more Sox2. Is the increased proliferation selective in basal Sox2?

In response to the request by reviewer 3, we now provide the quantification of Sox2⁺ and Tbr2⁺ cells in the mouse cortex as percent of cells (instead of “per 400 μ m-wide field”). With the new quantification, both Sox2⁺ and Tbr2⁺ cells are significantly increased in the SVZ/IZ (Figure 2 D, G). In addition, we have quantified PH3 and Sox2 double-positive cells in the mouse cortex (see new graphs in Figure 1J–L). This new data indicates that the majority of additional abventricular PH3-positive cells in EPIREGULIN-treated cortices are Sox2-positive. Taken together, this supports our conclusion that both Sox2⁺ and Tbr2⁺ cells contribute to the increase in basal progenitor cells.

Next, they go on to investigate if these are bRGs, but their definition is flawed. A study some years ago performed a detailed analysis of the cellular morphology of pVim⁺ cells in ferret, and demonstrated that presence/absence of pVim process does not correlate with bRG/bIP, contrary to what most labs commonly assume. Taking this into consideration, and given the results in Figs 1 and 2 here, the authors' conclusion that IPs increase is only based on increased abundance of Tbr2⁺ cells in VZ. However, they must demonstrate that these are dividing cells and hence that Tbr2⁺ mitoses increase, as well as Sox2⁺ mitoses (not currently shown).

Previously published detailed morphological analysis and interspecies comparison of basal progenitor cells by Kalebic et al, 2019 has revealed that less than 10% of basal progenitors in mouse are bRG, irrespective of using mGFP or pVim to identify processes. This is in line with earlier reports (Shitamukai et al., 2011; Wang et al, 2010; Wong et al, 2015) and in agreement with our pVim data in the SVZ (Figure 2K, L). In contrast to mouse, the ferret and human neocortex were reported to

contain much higher proportions of bRG and additional bRG morphotypes with two or more processes (Reillo et al., 2011; Reillo et al, 2017; Kalebic et al, 2019). Moreover, a seminal study in macaque showed that primate bRG morphology is highly dynamic and correlates with different proliferative capacities (Betiazeau et al, 2013). Additionally, interspecies differences have been reported with respect to the expression of cell type markers in basal progenitor cells with a basal process. While in ferret and human, the majority of basal progenitor cells with a basal process express SOX2/PAX6 (Fietz et al., 2010; Hansen et al., 2010; Reillo et al., 2011), we have previously shown that in the mouse lateral neocortex, the majority of basal progenitor cells with a basal process express TBR2 (Florio et al, 2015). In the mouse, bRG with human-like gene expression appear to be restricted to the medial neocortex at later developmental stages (Vaid et al, 2018).

Here, we have shown that abventricular mitoses increase upon EPIREGULIN addition in the lateral mouse neocortex and that the majority of these are Sox2⁺ (see previous point; new Figure 1L). In addition, we observed an increase in both abventricular pVim⁺ mitotic cells with and without a process. New co-staining of pVim with Sox2 revealed that among the abventricular pVim-positive cells with a process, both Sox2⁺ and Sox2⁻ cells are increased upon addition of EPIREGULIN compared to control (new Figure 2M, N). Thus, these new data confirm that “true” bRG, based on morphology (pVim⁺ process) and ferret/primate-like molecular identity (Sox2 expression), are increased upon addition of EPIREGULIN. In addition, Sox2⁻ process⁺ BPs and process⁻ bIPs are increased.

The authors conclude that Epiregulin can induce the amplification of different BP types in the mNcx. These conclusions are based on the combination of results from slice and whole hemisphere culture, but the authors did not show changes in Sox2 and Tbr2 cell abundance in HERO cultures, leaving this as an assumption. This must be shown, or the entire set of analyses must be performed in a single type of assay.

Since the slice culture does not allow to follow processes of progenitor cells, we performed this specific analysis using the whole hemisphere rotation (HERO) culture system that allows to cut sufficiently thick slices to track RG processes. To confirm that both types of culture systems provide comparable results, we have now included an additional figure (new Appendix Figure S3) showing that addition of EPIREGULIN leads to a significant increase in abventricular mitosis in both slice and HERO culture. In addition, the pVim staining confirms the increase in abventricular mitosis upon EPIREGULIN addition. Moreover, new Figure 2M, N provides information on Sox2 in basal progenitor cells with a process (see previous point).

When studying human cortical organoids, the authors conclude: "This indicates that EPIREGULIN is required for BP, but not AP, proliferation in human cortical organoids." - This is not necessarily like this. Abundance of Ki67+GFP⁺ cells in SVZ results from their self-renewal/amplification, but also from their genesis from VZ progenitors (vs postmitotic neurons). To discern between these two possibilities and reach some conclusion, the authors would need to perform proper cell lineage analyses. Are BPs (specifically) re-entering the cell cycle more with EREG than in control conditions?

Indeed, cell cycle re-entry analysis based on combination of BrdU labelling and KI67 staining is a well-established assay in animal models, such as mouse and ferret, to show self-renewal/amplification of neural progenitor cells. Therefore, we have attempted to perform the same assay in our human cortical organoid models, adjusting the time of BrdU administration to the longer cell cycle duration of human neural progenitor cells (roughly twice as long as the mouse). However, just like seen by others in the literature (Jong et al, Nat Comm 2021; Jabali et al, EMBO Rep 2022), we find a very low proportion of NPCs labelled with BrdU, much lower than theoretically expected. In combination with electroporation, we did not obtain meaningful numbers to quantify. Further optimization will be needed to setup this assay in human cortical organoids, which was not possible within the given time-frame. Alternative approaches, such a lineage tracing by live imaging, microinjection followed by

clonal analysis or viral barcoding, are not established in our lab and beyond the scope of this manuscript.

Figure 5 should be combined with Figure 4 in a single main figure. In these experiments there is a very obvious accumulation of Sox2+ and Ki67+ cells in the apical surface, suggesting a possible cell cycle arrest in M phase. If this is reproducible. The authors should report on the abundance of these markers in VZ, currently absent.

As requested by this reviewer and reviewer 3, we have added the quantification of KI67, SOX2 and TBR2 in the VZ for human and gorilla organoids (new Figure panels 4H–J and 5C–E). We have also carefully considered combining Figure 4 and 5, as suggested. But given the addition of the new data, the combined figure would not fit within the space restrictions and we therefore prefer to keep the two figures separate according to their logical separation by species. We thank the reviewer for pointing out the potential accumulation of SOX2 and KI67 at the apical surface. We have carefully checked all images, but did not find this to be a consistent observation and have therefore replaced the images in the Figure accordingly.

The authors indicate that "RNA-seq analysis of sorted NPC populations did not point to major gene expression changes as mechanism underlying the EPIREGULIN-induced increase in cell proliferation" - It is very surprising that genes fundamental in proliferation, such as cell cycle genes, did not change in this paradigm of strong changes in progenitor proliferation. Even if it was only a change in progenitor cell fate, as suggested above, then even more transcriptomic changes would be expected.

Indeed, the RNA-seq results may be unexpected, yet, similar results have been reported in the literature before. Knockout of *ASPM*, the most common recessive microcephaly gene, in ferret was shown to induce severe microcephaly associated with changes in cell type proportions (Johnson et al., Nature 2018). Yet, the authors report that "Single-cell RNA-sequencing [...] reinforced the conclusion that NPC proportions were altered in the *Aspm* knockout animals, although their transcriptional programs were mostly preserved". In the current study, we have used a targeted sorting approach for the isolation of specific neural cell populations (RG, IP, N) based on marker gene expression (Sox2, Tbr2, Tubb3). The approach applied a conservative sorting strategy and, thus, does not take into account the transitioning cell populations. In analogy to what has been reported by Johnson et al, Nature 2018, our data indicate that cell populations change, which may not necessarily require the expression of new genes in each of these populations. Bulk RNA-seq analysis of the entire cortex, where transcription from all cell types is averaged, may be expected to reveal increased proliferative signatures. In the design of our study, we have decided against such a bulk approach due to the difficulty in the interpretation of results, as changes in proliferative signatures may just reflect shifts in cell populations.

In the last set of experiments (Figure 7), mScarlet signal should be shown in a different color better contrasting with the black background. Maybe white.

We have revised Figure 7E/F, now showing mScarlet in white.

Regarding the role of human CREs in Epiregulin expression levels, one would conclude the opposite than the authors, but that the poor conservation of the CRE sequence demonstrates that these are not key to establish differences in expression levels between human and mouse. The proposed alternative, that differences are due to epigenetic marks, is not demonstrated in this study and thus just as speculative.

Previous data by Reilly et al, Science 2015, suggest that evolutionary changes in enhancer activity are implicated in the regulation of conserved developmental pathways during cortical evolution (now added to the discussion). Our data is consistent with this hypothesis, which is not mutually exclusive

to the contribution of reduced conservation. Accordingly, in the discussion we state that “local epigenetic modifications and chromatin compaction may further contribute...”.

In Discussion, when indicating that "non-coding regions of the genome contribute to evolutionarily relevant inter-species differences in gene expression", the authors should also cite Espinos et al. 2022, Dev Neurobiol, a comprehensive recent review on this fundamental question in brain evolution.

As suggested, we have added this reference, which provides an extensive review of cortical evolution beyond mammals, to the introduction and discussion.

The authors should discuss whether the CREs absent or very poorly conserved in mouse were a secondary loss during evolution, as shown to be the case for other genes (i.e. Chinnappa et al., 2022, Sci Adv), or rather a primate-specific innovation.

To address this question, we have plotted the conservation of the 11 human putative *EREG* CRE sequences in additional species, spanning primates (chimpanzee, gorilla, orang, rhesus, marmoset), rodents (mouse, rat) and carnivores (cat, dog, ferret) (new Appendix Figure S6). Overall, the CRE sequences are more conserved among the five primate species, including the near lissencephalic marmoset, than in other species with a gyrencephalic brain, such as the ferret, which has an expanded OSVZ and abundant bRG. Unlike suggested in Chinnappa et al, Sci Adv 2022, for the microRNA *miR-3607*, based on sequence conservation, there is no strong evidence for a secondary loss for most of the putative *EREG* CREs. For some of the putative *EREG* CREs, specifically CRE2, CRE7 and CRE11, parts on the sequences are not conserved in rodents, but are present in carnivores and primates. Overall, for most CREs, the conservation data is suggestive of primate-specific innovation rather than secondary loss. This point has been added to the discussion.

Referee #2:

Here the authors describe a novel regulator of specifically basal progenitors and basal radial glial cells. They identify epiregulin expression absent in the murine developing cortex, but expressed in primate and human cortex. The authors then demonstrate that addition to Epiregulin to murine cortex cultures increases specifically basal progenitors/RGCs, but not apical RGC proliferation and number. Conversely, deletion of *EREG* in cells of human iPSCs-derived cortical organoids reduced proliferation of SVZ, but not VZ cells and addition of Epiregulin showed effects in gorilla IPSC-derived organoids, but not human organoids, suggesting that the system is saturated in human cortex organoid, but not gorilla ones. The authors then proceed to sequence the mouse cortex after addition of Epiregulin and find no genes significantly regulated. They next searched for the receptor and identify EGFR as the main receptor based on selective pharmacological inhibitors reverting the effects of epiregulin addition in murine cultures. Lastly, they identify 2 regulatory sites in the human genome that may contribute to regulate Epiregulin expression in human, but not murine cortex development. These data are very interesting as expansion of basal progenitors and basal radial glia is highly relevant during ontogeny and phylogeny for expansion of specific brain regions.

We would like to thank the reviewer for their supportive comments and are pleased that the reviewer recognizes the relevance of our study in the context of brain evolution.

Minor suggestions

1) The ferret also has an enlarged cortex and transcriptome data have been published e.g. from Victor Borrells lab where the SVZ and VZ had been dissected. Is Epiregulin also expressed in the SVZ of the ferret? Please incorporate in the first part of the results section.

Thank you for raising this interesting question. We have mined the data from Victor Borrell's team on the microdissected ferret cortex (De Juan Romero et al, 2015). The microarray design includes one probe based on the human Epiregulin mRNA. Based on the microarray data, *EREG* may be expressed

at similar levels in the ferret VZ, ISVZ and OSVZ (see new Figure EV1H). Yet, inter-species comparison of expression levels of different studies is inherently difficult. Therefore, we have expressed ferret *EREG* levels relative to the house keeping gene *GAPDH*. Overall, the levels of *EREG* from this data set are much higher than expected (ca. 50-fold of the house keeping gene *GAPDH* and 750-fold of the growth factor gene *EGF*). In conclusion, published transcriptome data suggests that *EREG* may also be expressed in the gyrencephalic ferret cortex, yet additional confirmation using an independent approach would be required to draw a definitive conclusion.

2) I am surprised that no genes are changed in cortex cells upon the addition of epipegulin, as one would expect proliferation genes in basal progenitors to increase in expression. As the authors sorted even *Tbr2*⁺ cells, one would expect an increase in proliferation gene expression in this population. Please comment on this and possibly refine the sequencing data analysis.

As discussed in response to reviewer 1 (see above), the lack of changes in the transcriptional signatures of neural progenitor cells may be surprising, but has been reported before in other models with altered neural progenitor proportions. Specifically, in a ferret model of microcephaly (Johnson et al., Nature 2018), the authors reported that “Single-cell RNA-sequencing [...] reinforced the conclusion that NPC proportions were altered in the *Aspm* knockout animals, although their transcriptional programs were mostly preserved”. This is perfectly in line with our findings using a sorting approach to study specific neural cell populations. Bulk sequencing approaches that were regularly used before cell type-specific analysis became possible, often reported increased expression of proliferation genes. Yet, it is important to note that such bulk approaches cannot distinguish between actual increased expression of proliferation genes in a given cell type or a higher abundance of proliferative cells (as we suggest here). Only with novel approaches, such a cell sorting (used here) or single cell RNA-seq (Johnson et al., Nature 2018), these potential scenarios can be resolved.

Nevertheless, to corroborate our findings, we performed additional analysis of the RNA-seq data. In addition to analysing all the 18 data sets together (3 cell types x 3 replicates x 2 conditions), we have done pair-wise comparisons of each cell types with the respective treatment. Still, we do not find any differences in expression upon Epipegulin treatment.

3) The experiment on p.11 using growth factor expanded NSCs is very confusing. First because the data throughout the paper suggest that epipegulin affects basal progenitors, but not apical ones. So here they work with a neural stem cell population treated with FGF that also reacts. How do the authors align this with the remainder of their data.

In the developing cortex *in vivo*, the cerebrospinal fluid provides pro-proliferative signals to apical progenitors in the cortex (Lehtinen et al, Neuron 2011; Chau et al, Dev Cell 2015; and others). The cerebrospinal fluid stimulates the self-renewal of Sox2-positive progenitors, for example, by promoting *Igf2* and *LIFR* signalling (Lehtinen et al, Neuron 2011; Chau et al, Dev Cell 2015). For the *in vitro* proliferation of neural progenitor cells isolated from the embryonic cortex, *EGF* and *FGF* have been reported to be critical (Tropepe et al, Dev Biol 1999; and others). Upon initial isolation of NSCs from the developing cortex, these cells can undergo a few cell divisions even in the absence of growth factors. In culture, the cells then quickly become dependent of *EGF* and *FGF* and cease to proliferate in the absence of these two growth factors (Figure 6G). In fact, if *EGF* is replaced with *BMP4*, NSCs become quiescent, a model which is used to study qNSCs (Martynoga et al, G&D 2013).

To further explore the potential of Epipegulin to stimulate proliferation of NSC *in vitro*, we have repeated the NSC proliferation assay with both freshly isolated cells (passage 0) and NSCs already adapted to culture conditions (p9) (see new Figure EV4A–C). Interestingly, freshly isolated NSCs (p0) continue to proliferate for three days, independent of the presence of *EGF* and *FGF*, and can also not be stimulated to proliferate further by the addition of Epipegulin. This suggests that the freshly isolated NSCs from E12.5, which mostly represent Sox2-positive apical progenitor cells at this stage of development, may be in a fully activated proliferative state due to their recent exposure to the pro-

proliferative signals of the cerebrospinal fluid. NSCs (p9) adapted to culture in EGF and FGF completely stop to proliferate upon withdrawal of EGF (see new Figure EV4A–C). This can be rescued by addition of Epiregulin. In the presence of EGF, Epiregulin is not able to further promote proliferation, supporting the competition of the two growth factors that is discussed in the manuscript.

Basal progenitor cells in the subventricular zone reside away from the ventricle and the pro-proliferative signals of the cerebrospinal fluid. Thus, the increased proliferative capacity of basal progenitors in species with an expanded cortex is thought to result from a proliferative niche in the OSVZ (Fietz et al, 2012; Florio & Huttner, 2014; Libe-Philippot & Vanderhaeghen, 2021; Pollen et al, 2015), to which Epiregulin may contribute.

4) Have the experiments in Figure 3 been performed with independent lines? I see different symbols in the data points, but they are not explained (or I did not find the explanation neither in the Figure legend nor Figure panels).

We apologize for not specifying the nature of the different replicates. This information has now been added to the figure legends throughout the entire manuscript. In the *EREG* KO experiment in Figure 3, 12–18 organoids from 3 independent batches (indicated by different symbols) have been analysed.

Moreover, we acknowledge the importance of replicating the results in an independent cell line. The first set of KO experiments was performed in the CRTDi004-A iPSC line (Völkner et al, Nat Comm 2022). In addition, we have now repeated the *EREG* KO in the HPSI0114i-kolf_2 iPSC line from the HipSci initiative (see new Figure EV2C–G). The new data confirm that *EREG* KO leads to reduced KI67 and SOX2 in the SVZ in human cortical organoids.

5) To explain the differential effects of Epiregulin addition to Gorilla versus human cortical organoids could the authors perform Western Blot or ELISA to measure endogenous levels of Epiregulin that may be lower in the Gorilla organoids?

Indeed, it would be highly interesting to assess Epiregulin protein levels in different species. We have searched for antibodies targeting Epiregulin and acquired two that looked promising according to supplier information. To confirm that the antibodies are indeed able to recognize human Epiregulin protein, we have overexpressed human *EREG* in mouse N2a cells in culture and prepared protein extracts (new Appendix Figure S1A). Epiregulin has a predicted molecular weight of 19 kDa, whereas membrane-bound glycosylated proepiregulin has been observed to run at 25–30 kDa. The first antibody (rabbit anti-Epiregulin, ab233512, Abcam) did not detect any additional bands in the sample overexpressing Epiregulin. The second antibody (rabbit anti-Epiregulin, D4O5I, #12048, Cell Signaling) did show new bands at 25 and 35 kDa that may represent Epiregulin protein. However, we could not detect any signal related to endogenous Epiregulin in protein lysates from gorilla and human organoids (new Appendix Figure S1B), despite proper loading as confirmed using an anti-Vinculin antibody. Thus, unfortunately, we were not able to resolve endogenous Epiregulin levels in different species due to a lack of sufficiently sensitive antibodies.

6) I may have missed this, but are the CREs they identify in Figure 7 for human cells also present in Gorilla?

We have added a new figure (Appendix Figure S6, see also response to reviewer 1) that displays the conservation of the putative *EREG* CREs in additional species, including the gorilla. All of the 11 elements are largely conserved in gorilla and do not show any large deletions as seen in mouse.

Referee #3:

In this manuscript, the authors analyze the role of Epiregulin (*EREG*), a member of the epidermal

growth factor (EGF) family of ligands, in the regulation of basal progenitor proliferation in the developing human brain.

Using published RNAseq datasets from mouse and human fetal brains, combined with RTqPCR analyses of human/gorilla organoids, they first show that EREG is expressed in primate RGCs (at higher levels than in neurons) but is not expressed in the different cell types of the mouse developing cortex.

To explore the role of EREG in basal cortical progenitors, the authors used a large panel of approaches (Human and gorilla organoids, organotypic slices of mouse and human fetal brains, culture of mouse and human cortices (HERO), mouse NSCs, CrispR/Cas9 mediated KO, RNAseq analyses of FACS-selected cells from mouse fetal brain, and Epigenomic analyses).

They show that addition of EREG to the culture medium enhances the proliferation of basal progenitors of the mouse neocortex and in gorilla organoids, but not in human organoids, which they attribute to the human cortex being EREG saturated. Inhibition of EREG expression in human organoids suggests that EREG is required to sustain basal progenitor proliferation.

They further suggest that in the mouse, EREG competes with EGF to promote basal RG proliferation, and that its action is mediated via the EGFR.

Seeking for the mechanisms involved in the differential regulation of EREG expression across species, they then studied putative EREG enhancer regions in the human genome. They show that the *Ereg* locus is enriched in repressive marks (H3K27me3) and devoid of active marks (H3K4me3 and H3K27ac) in the mouse neocortex, whereas the EREG locus is enriched in active marks (H3K27ac) in the human cortex. Removing the repressive mark H3K27me3 in the mouse neocortex did not result in an upregulation of *Ereg* gene expression, suggesting that the *ereg* locus is in a fully repressed state in the mouse. Using ATACseq and published H3K27ac ChIP-seq data sets of the human fetal cortex, they further suggest that cis-regulatory elements may be involved in inter-species differences in EREG expression.

General comments:

The ms is well-written and the results are clearly presented. The results are topical and of great interest and significance in the field of cortical development, with respect to human specificities. However, in numerous instances, the ms suffers from "shortcuts" where the authors make unsubstantiated claims (see below in specific comments).

Some conclusions are based on limited data sets -or on unknown numbers. Increasing the number of samples/observations will improve the significance of the results.

Altogether, these interesting results need to be strengthened.

We are pleased that this reviewer acknowledges the wide range of approaches used, and the interest and significance of our work for cortical development and evolution. The reviewer also highlights a number of short comings that we address in the following.

As examples (non exhaustive):

Fig1D: RT-qPCR: 3 organoids were analysed. The authors should increase the number of biological replicates.

As requested by the reviewer, more replicates were added to the RT-qPCR analysis of *EREG* expression in gorilla and human organoids (Figure 1D) to account for the higher variability of organoid structures compared to *in vivo* development. Each data point represents one organoid, and organoids from different batches are marked by different symbols, which is now explained in the figure legend. *EREG* expression is robustly detected in gorilla and human organoids, with somewhat lower levels than in human foetal tissue (difference is not significant).

The number of slices & mice/organoids analysed should be provided.

Fig1G-P Fig 2 (SOX2; TBR2), Fig3 D,E,G,H...

We previously showed individual replicates, and thus numbers, in all graphs. We apologize that we have neglected to specify the nature of the replicates (mice, individuals, organoids, batches etc.). This information has now been added to all figure legends.

The quantification of positive cells is not appropriate:

The authors quantified the numbers of PH3+, Ki67+ PCNA+ cells/ 400 or 200µm-wide field (Fig1; Fig6I). Idem for SOX2, TBR2 and pVimentin+ cells (Fig2). This rough analysis may vary depending on cell density in the ROI under consideration. The authors should compute the percentage of positive cells/ total cells (using Dapi-as in Fig4A), and indicate the number of slices and samples analysed.

According to the reviewer's suggestion, we have re-quantified all data related to Ki67, PCNA (new graphs in Figure 1N–S), Sox2 and Tbr2 (new graphs in Figure 2B–G). This data is now displayed as percent of cells (based on DAPI). While the individual numbers and statistical analysis change slightly, the overall conclusions remain valid. The number of mitotic cells (PH3, pVim) is commonly presented per 100/200/400 µm-wide field in the literature (see for example, Chinnappa, Borrell et al, *Sci Adv* 2022; Di Matteo, Cappello et al, *EMBO Mol Med* 2020; Johnson, Walsh et al, *Nature* 2018; Kalebic, Huttner et al, *Cell St Cell* 2019). This number is less affected by cell number and rather depends on the length of the apical surface, in particular in the mouse. We therefore feel that for mitotic cells, it is a more appropriate representation to normalize by apical surface/field. The number of slices and samples is indicated in the graphs, as individual replicates are shown. As discussed above, the replicates are now better described in the figure legends.

Specific comments :

The authors conclude that EREG specifically acts on basal progenitors of the SVZ -but not apical progenitors). The results are presently unclear and this conclusion needs to be correctly substantiated.

- "addition of recombinant EPIREGULIN to mNcx cultures resulted in an increase in NPC proliferation, preferentially in the SVZ" (Page 8; Fig 1G-P; Fig2B): this statement is not fully substantiated as it is based on quantification of the number of cells/field (max 6 fields analysed per condition). The authors should properly document the increase in proliferation using cell-cycle and proliferation quantification with respect to the total number of cells and not just by computing numbers over ROI (200 or 400 microns). This will allow them to go beyond suggesting an effect as written in the last sentence of this section- which contrasts with the title of the subsection.

As discussed in response to the previous points, the cell cycle markers Ki67 and PCNA as well as the cell type markers Sox2 and Tbr2 are now re-quantified as percent of cells (new graphs in Figures 1N–S and 2B–G). The overall conclusions remain valid.

"Both major types of BPs, bIPs characterized by Tbr2 expression and the lack of a process as well as bRG marked by Sox2 and the presence of a process, were increased, suggesting that EPIREGULIN can induce the amplification of different BP types in the mNcx".

Again this conclusion (page 8) is not demonstrated by the analysis as the authors did not provide evidence that Tbr2+ bIPs lack processes nor that SOX2+ bRGs bear a process. The authors should quantify the percentage of cells with and without a process co-stained with p-Vimentin/SOX2 and TBR2 to substantiate their claim.

To address this point, we have now co-stained pVim and Sox2 (see revised Figure 2I and new graphs in Figure 2M, N, and response to reviewer 1). Upon Epiregulin treatment, Sox2 is increased in basal progenitor cells with and without a process. This further corroborates our conclusion that Epiregulin promotes proliferation of both major types of basal progenitors, bRG and bIPs.

"EPIREGULIN ablation in human cortical organoids reduces basal progenitor proliferation"
"Both SOX2 and TBR2 were reduced upon EREG targeting (Fig 3G, H), suggesting that EPIREGULIN contributes to the proliferation of both BP types, bIPs and bRG, in 6-week cortical

organoids, in which the SVZ-like area is in the process of expansion".

The results are shown on Fig3. In 3C, the limit between VZ and SVZ/CP is not convincing as the bar on the left appears to be located within the VZ. This is an important issue regarding which type of progenitor is affected. The authors should therefore check for neuronal markers for ascertaining the boundaries between the different cell compartments.

Indeed, the marking of the zone boundaries was not clear in the previous figure 3. We thank the reviewer for pointing this out. In the revised figures, we now outline the VZ boundary along the entire width of the tissue in all mouse embryonic cortices, human foetal tissue and cortical organoid images. As discussed in response to reviewer 1 and now stated in the revised manuscript, the VZ boundary is deduced based on the alignment of nuclei in DAPI, and confirmed by additional markers.

Did the authors observe increased percentage of neurons in the KO condition following 7 days of in survival? The authors should also provide the % of SOX2+ in both VZ-like and SVZ-like as well as that of TBR2+.

As requested, we have analysed neuronal markers following *EREG* KO in the human cortical organoids. Staining for CTIP2 revealed a significant decrease in the percentage of neurons among electroporated cells in the VZ and no change in the SVZ/CP (see new Figure EV2C–E). We confirmed these results with staining for two additional neuronal markers (HuC/D and Tbr1; not shown). Thus, a reduction in basal progenitors did not result in more neurons, as suggested by this reviewer. In fact, one may have expected the opposite as basal progenitors are a major source of neurons. Moreover, a recent preprint (Wang, Kriegstein, et al, BioRxiv 2024) reports the identification of tripotential intermediate progenitor cells (termed “tri-IPC”) in the human developing neocortex that can give rise to astrocytes, OPCs and interneurons. While it remains to be shown whether these also exist in human cortical organoids, it is interesting to note that these tri-IPCs are characterized by high EGFR expression.

The data for KI67, SOX2 and TBR2 in the VZ has now been included, in addition to the SVZ/CP, for human cortical organoids (new panels D, G, I in revised Figures 3).

Fig3F: It would be useful to have merged images GFP/SOX2 and GFP/TBR2

These panels have been added to revised Figure 3F, as requested.

"Unlike the human, gorilla BP proliferation, in particular the percentage of SOX2-positive cells likely representing bRG, could be further stimulated by the addition of EPIREGULIN" (Page 10): The assay on human fetal tissue tests the effect of Epieregulin over a 24 hours period and shows a non-significant trend in increasing the % of Ki67+, Sox2+ and TBR2 + in the SVZ. This short duration may well not allow to detect an effect given the prolonged cell-cycle duration of human cortical progenitors. Hence, the human cortical organoid assay (10 days of culture). The percentage of SOX2+ cells should be provided in both the VZ and SVZ (Fig5C-E).

The data for KI67, SOX2 and TBR2 in the VZ has now been included, in addition to the SVZ/CP, for both human cortical organoids (new panels H–J in revised Figures 4) and gorilla cortical organoids (new panels C–E in revised Figure 5).

The limits of VZ, OSVZ must be shown on Fig 4B.

The zone boundaries are now outlined in the revised figure.

In the mouse, inhibition of EGFR results in a loss of the EREG-induced increase in abventricular mitosis.

"We then performed HERO cultures with only the solvent (ethanol; control), solvent and 50 ng/mL

EPIREGULIN, or EPIREGULIN and different concentrations of the two inhibitors" (Page12): Only one concentration of inhibitor has been tested (1 μ M) (Fig6I). How has this dosage determined?

The concentrations of inhibitors were selected based on previous literature (Martin *et al*, 2017; Xu *et al*, 2017). Initially, we tested higher concentrations (10 μ M instead of 1 μ M), but found that this higher concentration affected the integrity of the mouse cortical tissue and resulted in many DAPI-dense small nuclei that are characteristic of apoptotic cells. An example image of the treatment with 10 μ M of Dacomitinib is now included in new Figure EV4D, including a high magnification view of the cortical plate.

The effects of the two inhibitors AG1478 and Dacomitinib on human/gorilla organoids or human fetal cortex should be examined, as they appear more relevant than mouse cortex. Do these inhibitors alter the proliferation rate of VZ apical progenitors and OSVZ bRGCs?

As requested by this reviewer, we have treated human cortical organoids with the two inhibitors AG1478 and Dacomitinib to interrogate whether EGFR- and/or Erbb4-signalling is globally required for apical and basal progenitor proliferation in human (new Appendix Figure S5). We have treated 6-week human cortical organoids for 7 days using each of the two inhibitors at a 1 μ M and 10 μ M concentration. Inhibitors were replenished with each media change every 2-3 days. The 10 μ M Dacomitinib concentration resulted in disassembly of the organoid structures, suggesting that the concentration was too high, as discussed above for the mouse. For all other conditions, the organoid structures were overall well preserved. We did not observe any significant changes in KI67, SOX2 and TBR2, neither in the VZ nor SVZ. While this may be unexpected, it is worth to note that targeted disruption of *Egfr* in mice does not lead to any apparent neural phenotypes at early stages of development and some mice can survive several weeks postnatally, while others display mild to severe forebrain defects (Sibilia and Wagner, Science 1995; Sibilia *et al.*, EMBO J 1998; Threadgill *et al.*, Science 1995; Tropepe *et al.*, Dev Biol 1999).

"These results suggest that EGFR is the key mediator of the EPIREGULIN-mediated increase in BP amplification, whereas Erbb4 appears to not play a major role". The authors should test an inhibitor specific to Erbb4 to better substantiate their claim.

Indeed, we were also interested to interrogate ErbB4 function directly, but could not find an inhibitor that is specific only to that receptor. To still address the role of the different receptors we have therefore selected two different inhibitors, one that specifically targets EGFR and one pan-ErbB receptor inhibitor (targeting EGFR, ErbB4 and other ErbB receptors). Since both inhibitors completely block the EPIREGULIN-mediated increase in abventricular mitosis, we conclude that EGFR is the main receptor mediating the downstream effect and that ErrB4, despite its high expression in the mouse, does not play a major role.

"Epigenome editing in the mNSCs using dCas9-JMJC_6B resulted in a reduction of H3K27me3 at the *Ereg* locus, but not at the unrelated *Hoxb5* and *Eomes* genes" (Page 13)

In contrast to author's claim, there does not seem to be a significant decrease in H3K27me3 (FigS6E). This may explain why there is no increase in *ereg* expression in FigS6F&G. The authors need to clarify this.

We have previously stated that "Epigenome editing in the mNSCs using dCas9-JMJC_6B resulted in a reduction of H3K27me3 at the *Ereg* locus, but not at the unrelated *Hoxb5* and *Eomes* genes". For another, unrelated gene, we have observed that a similar level of reduction lead to an upregulation of expression. However, unlike for *Ereg*, in that case the locus was initially in a bivalent state and therefore like more susceptible to upregulation (unpublished data). We have rephrased the text in the manuscript for clarity, from previously stating "loss of repressive epigenetic modifications" to "reduction in repressive epigenetic modifications".

"Two additional genes are located within this genomic region, AREG and EPGN.... However, based on chromatin (Fig 7A) and RNA- seq (Fig EV5) data, they are likely not expressed in the fetal hNcx".

Note that as shown on Fig S5B, EPGN is expressed in RGs and Neurons at GW12-13. Given that this gene encodes a growth factor that binds to EGFR, it may also play a role at this stage. The authors need to comment upon this.

Whereas *EREG* expression was consistently seen in several data sets from human foetal tissue and organoids (Camp et al, PNAS 2015; Florio et al, Science 2015; Johnson et al, Nature 2015), *AREG* expression was not seen in transcriptome data from Florio et al. and Camp et al., and *EPGN* was not seen in data sets from Johnson et al. and Camp et al. All these references have now been added to the manuscript.

Minor comments:

Fig 6H : The limit between VZ and OSVZ must be shown.

Lines have been added to delineate the zone boundaries.

"The Ereg locus is enriched in repressive marks (H3K27me3) in the mouse neocortex, whereas it is enriched in active marks (H3K27ac) in the human neocortex" (Page 6). What about repressive marks in humans (H3K27me3) (not shown in FigS1-correct the legend)?

To our knowledge, H3K27me3 has not been mapped in a cell type-specific manner in the human foetal cortex. Single cell profiling of H3K27me3 by CUT&Tag in human cerebral organoids has recently been reported in a preprint (Zenk, Treutlein et al., bioRxiv 2023), but the data has not yet been released for mining. A data set of H3K27me3 ChIP-seq (GSE63634; Yan, Tang, J Biol Chem 2016) in total foetal brain reveals H3K27me3 peaks at the *EREG* locus, yet the bulk data is difficult to interpret as *EREG* is not expressed in neurons, which may contribute to the observed signal.

The presence of H3K27ac at the *EREG* locus in the hNcx (Figure EV1E) suggests that H3K27me3 is low in human NPCs, where *EREG* is expressed, as these two histone modifications occur on the same amino acid residue and are thus mutually exclusive.

The legend has been revised to now specifically state that in human, only H3K27ac is shown.

"Both EGF and FGF were reported to elicit NSC proliferation in vitro (Tropepe et al, 1999), and upon acute withdrawal of EGF, mNSCs did not proliferate further" (Page 11).

This is somehow surprising that mNSCs stop proliferating 2-days after EGF removal (the medium still contains FGF2) (Fig6G) since it has been shown that mNSCs can proliferate in the absence of EGF (Tropepe et al).

As discussed in response to reviewer 2 (see above), freshly isolated NSCs (Tropepe et al, Dev Biol 1999) and NSCs adapted to culture (this manuscript) respond differently to withdrawal of growth factors. We have therefore repeated the NSC proliferation assay with both freshly isolated mouse NSCs (p0) and NSCs that were adapted to culture (p9) (see new Figure EV4A–C). In agreement with the literature (Tropepe et al, Dev Biol 1999), freshly isolated NSCs did continue to divide in the absence of EGF and FGF and could not further be stimulated by addition of Epirigulin. The proliferation assay with the later passage NSCs confirmed our previous results, showing that Epirigulin promotes proliferation in the absence of EGF. As discussed in response to reviewer 2, the recent exposure of freshly isolated NSCs to the pro-proliferative signals of the cerebrospinal fluid may contribute to the differences between the two NSC states.

Fig7A: please correct the drawing of ATAC and H3K27ac.

We would like to apologize, but we cannot identify the error to be corrected. We would be happy to address this comment if the reviewer could provide more specific instructions.

"the presence of different epidermal growth factors modulates the cellular outcome". Do they mean "different growth factors"?

This sentence refers to the different members of the epidermal growth factor family.

"To test whether these putative enhancer regions display enhancer activity, we cloned two of the human CRE regions (216 bp)": the authors mention CRE6 and CRE9: why CRE9 was chosen?

We selected two CREs upstream of the *EREG* TSS (CRE5 and CRE6) and one downstream of the TSS (CRE9). One of them dropped out since we were not able to clone it, and therefore we proceeded with CRE6 and CRE9.

"Comparing hCRE6 and mCRE6, higher levels of mScarlet were observed for hCRE6 compared to the orthologous mouse sequence (Fig 7E). Human CRE9 was also able to drive mScarlet expression at similar levels as hCRE6, yet, in this case the orthologous mouse sequence showed similar or even higher activity (Fig 7F)": the level of mScarlet or the percentage of Scarlet+ cells should be quantified (Fig7E,F).

Indeed, this is an important comment and we now provide quantifications of the level of mScarlet in the VZ + SVZ and the CP for all constructs (new graphs in Figure 7 F/G). The elements hCRE6 and mCRE9 show a significantly higher level of expression compared to the control, supporting the conclusion that they may represent putative enhancer elements.

Dear Mareike,

Thank you for the submission of your revised manuscript to The EMBO Journal. We have now received the comments of the three referees that were asked to re-assess your study (included below). As you will see, all referees are satisfied with the revision, and they acknowledge that the revised manuscript is significantly improved, and that most of their previous concerns have been satisfactorily addressed. However, referees #1 and #3 raise a number of remaining concerns which we need you to address with appropriate corrections and/or clarification in a minor revision before we can proceed with acceptance of your manuscript for publication.

From the editorial side, there are also a few minor changes and corrections that we need from you:

- Please note that, as per our journal's policy, middle authors cannot be designated as having contributed equally to the study; this is possible only for first and last authors. Each author's contributions can be specified in detail using the CRediT system during re-submission (please see my relevant comment below for more information).
- Please enter all relevant funding information in our online manuscript handling system (eJP). It should match exactly the information provided in the Acknowledgements section of your manuscript. Currently, information on ERA-NET Neuron (MEP|cephaly) is missing in eJP.
- Please make sure that the deposited RNA sequencing data are publicly available at the time of publication, and add the specific URL of the dataset to your Data availability section. The reviewer access token can now be removed.
- Please change the heading of your conflict-of-interest statement to "Disclosure and competing interests statement".
- The author contributions statement should be removed from the manuscript file. Instead, we use CRediT to specify the contributions of each author in the journal submission system. Please use the free text box to provide more detailed descriptions during submission. See also our guide to authors for more information:
<https://www.embopress.org/page/journal/14602075/authorguide#authorshipguidelines>.
- We noticed that there are no callouts for Figure panels 5A-E; please make sure that all Figure panels are called out (in alphabetical order) in your revised manuscript.
- Please add a brief Table of Contents including page numbers on the first page of your Appendix.
- The nomenclature of the Appendix Figures should be "Appendix Figure S1" to "Appendix Figure S6". Please update accordingly all Appendix Figure legends and their respective callouts throughout the Appendix and the main manuscript file.
- Please move the References of the Appendix to the main list of References in the manuscript file.
- Please remove "Graphical abstract. Model of EPIREGULIN-mediated regulation of BP proliferation in the neocortex of different species." from the first page of your Appendix.
- Please note that the uploaded numerical source data should be included in the respective Figure-specific folders. All source data files need to be saved in one folder per Figure and then uploaded as a .zip item. For example, all source data files for Figure 1 need to be saved in a single folder, which then needs to be zipped and uploaded as "SD Figure 1.zip".
- The EMBO Journal papers are accompanied online by:
A) a short (2 sentences-long) summary of the findings and their significance, and
B) 2-5 short bullet points highlighting the key results.
Could you please upload this information in a separate Word file along with your revised manuscript?
- Please note that information related to "n" is missing in the legends of Figures 1c; 6b-c, EV 1g-h.
- Please note that no statistics should be calculated and presented when n=2. This seems to be the case in Figure EV 4c, and I would thus suggest showing the individual data points instead of including SD error bars since n=2.
- Please note that the error bars are not defined in the legends of Figures EV 1g-h.
- Please note that the measure of center for the error bars needs to be defined in the legends of Figures 1c-d, g-l, n-s; 2b-g, j-n; 3d-e, g-j; 4c-e, h-m; 5c-h; 6b-c, f-g, i; 7f-g; EV 2d-g, i-j; EV 3c-d; EV 4c; EV 5c, e-f.
- Please indicate the statistical test used for data analysis in the legend of Figure EV 3e.

- Tables S1-S3 should be renamed Table EV1-EV3 (the callouts are correct).

Please also note that as part of the EMBO publications' Transparent Editorial Process, The EMBO Journal publishes online a Peer Review File along with each accepted manuscript. This File will be published in conjunction with your paper and will include the referee reports, your point-by-point response and all pertinent correspondence relating to the manuscript. You can opt out of this by letting the editorial office know (contact@embojournal.org). If you do opt out, the Peer Review File link will point to the following statement: "No Peer Review File is available with this article, as the authors have chosen not to make the review process public in this case."

We look forward to seeing a final version of your manuscript as soon as possible. Please use this link to submit your revision: <https://emboj.msubmit.net/cgi-bin/main.plex>

Best regards,

Ioannis

Referee #1:

I congratulate the authors for having responded satisfactorily to my concerns. However, there are some points that still remain for final correction, which the editor should be able to judge without the need for another round of peer review.

Related to my second point, the legend and/or Figure 2B-E must indicate explicitly what the percentage refers to. Is it % of DAPI cells? % of cells expressing that marker across the cortex? Regarding Tbr2, the new quantifications do not demonstrate that Tbr2+ cells increase, but rather disguise that they do not increase. In light of the previous quantifications "per 400 um-wide field", the new "%" quantification indicates that in Epiregulin treatment there are less total cells (DAPI?) in SVZ, a reduction not affecting Tbr2+ cells as demonstrated by the "per 400 um..." measurements. Taken together, the data indicates that Epiregulin treatment reduces cell numbers in SVZ but sparing Tbr2+ cells, which do not change, while Sox2+ cells and Sox2+ mitoses increase. This must be openly and clearly presented in the manuscript.

Referee #2:

The authors have considerably improved their manuscript including addressing the most important issues I had raised, e.g. repeating the organoid analysis with a distinct line. The interesting new findings of this work are now all substantiated well by the data and experimental approaches. This work is now ready to be published in EMBO Journal.

Referee #3:

The authors addressed most of our concerns. They have adequately requantified the data. Specifically, they express the results as percentage of cells-instead of numbers- in the majority of graphs. They also show all the numbers and types of replicates for each experiment, as required. These changes substantiate and fully validate their conclusions.

Major comment

There is one remaining concern regarding the identity of the progenitors whose amplification is regulated by EREG (HERO culture/mouse Ncx) (page 8/fig 2).

In their response, the authors stated that "Upon Epiregulin treatment, Sox2 is increased in basal progenitor cells with and without a process. This further corroborates our conclusion that Epiregulin promotes proliferation of both major types of basal progenitors, bRG and bIPs". However, this is still not clear since the authors did not quantify SOX2 in "basal progenitor cells with and without process", but only in progenitors "with process" as shown in Fig. 2M,N. We guess that the graphs display the

percentage of "pVim+ cells" and not "pVim+ Sox2+cells" as indicated ? This needs to be corrected.

Minor comments:

In several instances, the authors have not corrected the text according to the revised quantifications.

For example, page 8 :

Since the authors now show the percentage of Ki67, PCNA and SOX2 positive cells, they should replace "number of Ki67/PCNA/SOX2 positive cells " by "the percentage of Ki67/PCNA/SOX2 positive cells".

"Moreover, there was no change in Ki67 and PCNA in the VZ...":needs to be reworted "there was no change in the percentage of..."

"both Ki67 and PCNA were significantly increased in the SVZ/IZ..."

"Likewise, SOX2 was unchanged in the VZ": should be: "the percentage of SOX2 positive cells...".

This list is non-exhaustive; the authors need to correct all relevant sentences.

-In the numerical source data/Fig2: the authors show the results for"pVim SOX2- with process" and "pVim SOX2+ with process": for "pVim SOX2- with process">>> "Fig2"M" should be "N".

REVISION II**Referee #1:**

I congratulate the authors for having responded satisfactorily to my concerns. However, there are some points that still remain for final correction, which the editor should be able to judge without the need for another round of peer review.

We would like to thank the reviewer for their constructive criticism. We are pleased that our response to the concerns is satisfactory. The remaining points are clarified below.

Related to my second point, the legend and/or Figure 2B-E must indicate explicitly what the percentage refers to. Is it % of DAPI cells? % of cells expressing that marker across the cortex? Regarding Tbr2, the new quantifications do not demonstrate that Tbr2+ cells increase, but rather disguise that they do not increase. In light of the previous quantifications "per 400 um-wide field", the new "%" quantification indicates that in Epiregulin treatment there are less total cells (DAPI?) in SVZ, a reduction not affecting Tbr2+ cells as demonstrated by the "per 400 um..." measurements. Taken together, the data indicates that Epiregulin treatment reduces cell numbers in SVZ but sparing Tbr2+ cells, which do not change, while Sox2+ cells and Sox2+ mitoses increase. This must be openly and clearly presented in the manuscript.

We have added the specific information on the percentage (percent of DAPI positive cells) to the legend of Figures 1N-S and 2B-G.

To address the second point, we have included an additional quantification of the total number of cells (based on DAPI, see new Appendix Figure S3A-C). This quantification shows that the overall number of cells is not decreased in the SVZ/IZ upon EPIREGULIN treatment. Thus, we conclude that the increased percentage of Tbr2-positive cells in the SVZ/IZ (Figure 2G) represents a real increase.

Referee #2:

The authors have considerably improved their manuscript including addressing the most important issues I had raised, e.g. repeating the organoid analysis with a distinct line. The interesting new findings of this work are now all substantiated well by the data and experimental approaches. This work is now ready to be published in EMBO Journal.

We are pleased that the reviewer is satisfied with the revised manuscript. We would like to thank the reviewer for their helpful comments that have contributed to strengthening our findings.

Referee #3:

The authors addressed most of our concerns. They have adequately requantified the data. Specifically, they express the results as percentage of cells-instead of numbers- in the majority of graphs. They also show all the numbers and types of replicates for each experiment, as required. These changes substantiate and fully validate their conclusions.

We appreciate the reviewer's constructive criticism and would like to thank them for their helpful comments. We are pleased that the reviewer acknowledges that validity of our conclusions.

Major comment

There is one remaining concern regarding the identity of the progenitors whose amplification is regulated by EREG (HERO culture/mouse Ncx) (page 8/fig 2).

In their response, the authors stated that "Upon Epiregulin treatment, Sox2 is increased in basal

progenitor cells with and without a process. This further corroborates our conclusion that Epiregulin promotes proliferation of both major types of basal progenitors, bRG and bIPs". However, this is still not clear since the authors did not quantify SOX2 in "basal progenitor cells with and without process", but only in progenitors "with process" as shown in Fig. 2M,N. We guess that the graphs display the percentage of "pVim+ cells" and not "pVim+ Sox2+cells" as indicated ? This needs to be corrected.

Indeed, the graph was incorrectly labelled, which we have now corrected. In Figure 2M/N, abventricular cells "with a process" are shown. The label on the y-axis was corrected to refer to "pVim+ cells" and the Sox2 state is indicated above each graph.

Minor comments:

In several instances, the authors have not corrected the text according to the revised quantifications.

For example, page 8 :

Since the authors now show the percentage of Ki67, PCNA and SOX2 positive cells, they should replace "number of Ki67/PCNA/SOX2 positive cells " by "the percentage of Ki67/PCNA/SOX2 positive cells".

"Moreover, there was no change in Ki67 and PCNA in the VZ...":needs to be reworded "there was no change in the percentage of..."

"both Ki67 and PCNA were significantly increased in the SVZ/IZ..."

"Likewise, SOX2 was unchanged in the VZ": should be: "the percentage of SOX2 positive cells..."

This list is non-exhaustive; the authors need to correct all relevant sentences.

All sentences have been corrected in the text to properly represent the re-quantified data.

-In the numerical source data/fig2: the authors show the results for "pVim SOX2- with process" and "pVim SOX2+ with process": for "pVim SOX2- with process">>> "Fig2" M" should be "N".

For clarity, we now refer to each panel (Sox+ and Sox2-) separately in the text.

Dear Mareike,

I am pleased to inform you that your manuscript has been accepted for publication in The EMBO Journal.

Best regards,

Ioannis
